# Representing Part-Whole Hierarchy with Nested Neuronal Coherence

## Abstract

Human vision flexibly extracts part-whole hierarchies from visual scenes. However, representing such hierarchical structures poses a key challenge for neural networks. Most machine learning efforts addressing this issue have focused on slot-based methods, which can be limiting due to their discrete nature and inability to express uncertainty. Inspired by how neural syntax is organized in the brain, this paper presents a framework for representing hierarchical part-whole relationships through hierarchically nested neuronal coherence, characterized by its continuous and distributed nature. At the implementation level, we further developed a cortical-inspired hybrid model, Composer, which dynamically achieves emergent nestedness when given images. To evaluate the emergent hierarchical structure, we created four synthetic datasets and three quantitative metrics, demonstrating its ability to parse a range of scenes of varying complexity. We believe this work, from representation and implementation to evaluation, advances a new paradigm for developing human-like vision in neural network models.

## 1 Introduction

The representation of hierarchical structures is a critical challenge for neural networks (Riesenhuber & Dayan, 1996). While there is strong evidence from psychology that humans naturally parse objects into part-whole hierarchies (Hinton, 1979; Kahneman et al., 1992; Thompson, 1980), representing these relationships in artificial neural networks remains a profound challenge (Hinton, 2021).

The challenge is illustrated by an example. Given a visual scene, as shown in Fig.1a, the mental interpretation of the parts and wholes can be uncertain—such as the number and grouping of objects, and so on. Therefore, the structure is dynamically inferred based on the "content" and is continuously constrained by the self-consistency of multiple sources of inconclusive information (Rogers & McClelland, 2014). The uncertain and content-sensitive nature of part-whole hierarchies challenges solutions directly inspired by symbolic processing, which is discrete and lacks content sensitivity.

Most efforts to address this challenge in machine learning focus on slot-based methods (Greff et al., 2020), which can be viewed as a symbolic-inspired solution. These approaches divide their latent representations into predefined "slots," creating a discrete separation of object representations (Greff et al., 2020). These slots are easier to organize into compositional structures, such as graph neural networks (Han et al., 2022; Xu et al., 2017), generative models (Deng et al., 2021), attention-based models (Sun et al., 2021; Fisher & Rao, 2022), or capsule networks (Hinton et al., 2018; Garau et al., 2022). However, the predefined and discrete nature of slots conceptually limits their ability to continuously infer content-sensitive structures from uncertainty. For instance, the existence and maximum number of objects are predefined discretely when specifying the slots, independent of the content.

By contrast, discrete compositional structures, such as mental entities or their structural organization, are hypothesized to be represented by continuously and dynamically correlated neuronal firings (Buzsáki, 2010) (Fig.1d). This overall state is known as neuronal coherence, and each correlated group of neurons is referred to as a cell assembly (Fig.1d), which can be organized into sequences and hierarchical structures to form neural syntax (Buzsáki, 2010). Inspired by neuroscience, this paper aims to develop a framework that represents part-whole hierarchies not with discrete slots, but through continuous neuronal coherence in neural networks, thus bridging a key feature of human-like vision with machine learning models.

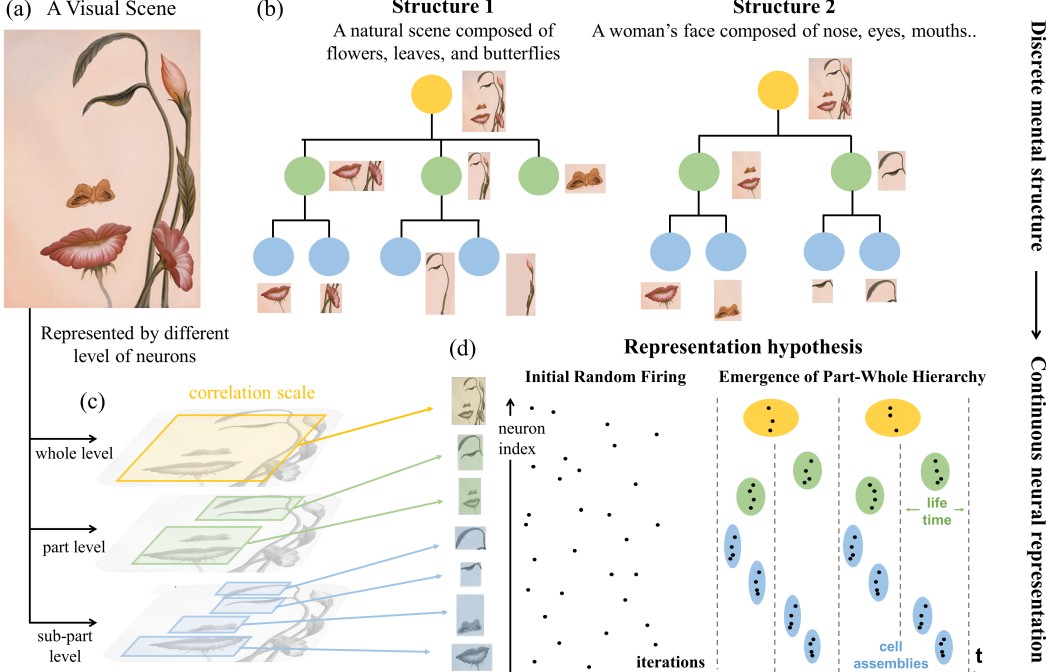

Figure 1: (a) A visual scene. (b) Different plausible mental parsing structures of the same visual scene. (c) Feature neurons are divided into different levels, each associated with its own spatial scale, as indicated by colored boxes of varying sizes. Each greyscale image illustrate the "pixel-level" feature maps of each population of neurons. Here, for simplicity, levels have shared feature map, but are distinguished by their correlation scales. (d) Spike raster plot: represent the mental structure 2 in (b) as nested neuronal coherence (right), emerged from initial random firing (left). The y-axis of the spike raster plot are neuron id (indicated by arrows). Cell assemblies (right) are indicated by colored shadows. Color stands for level in all cases.

How does neuronal coherence emerges as a reflection of the parsing structure of various visual scenes? At the implementation level, we adopt a hybrid approach that combines biological constraints and machine learning, to develop a cortical-column-inspired neural network model, **Composer** (short for **CO**rtical-inspired e**M**ergence of **P**art-wh**O**le relation**S**hip through n**E**uronal cohe**R**ence). Through the iterative bottom-up and top-down processing within and across columns, nestedness gradually emerges from randomness.

How can we explicitly and quantitatively evaluate the discrete hierarchical structure that is implicitly represented as the continuous and uncertain correlation pattern among neuronal firings? To this end, we developed four synthetic datasets of varying complexities and three quantitative metrics to measure different aspects of part-whole representation. Both quantitative results and qualitative visualizations confirm the validity of Composer and the plausibility of the framework. Finally, for comparison, we show that Composer outperforms the state-of-the-art method, the Agglomerator (Garau et al., 2022). Our contributions are summarized as follows:

(1) We developed a brain-inspired representation framework to address the part-whole hierarchy in neural networks through continuous neuronal coherence instead of discrete slots.

(2) We introduced Composer, a brain-inspired hybrid model that integrates machine learning and neuroscience, to demonstrate how the coherent state emerges to represent the part-whole relationship.

(3) We created datasets and quantitative metrics to explicitly interpret the hierarchical structure from uncertain and distributed representations in neural networks. Quantitative evaluation of hierarchical relationship is often missing in related works, yet it is essential for verifying or falsifying models.

Taken together, we are pioneering a systematic framework to represent the part-whole relationship in neural networks, from representation, implementation, to evaluation, and we hope that the resulting insights will spark further innovations.

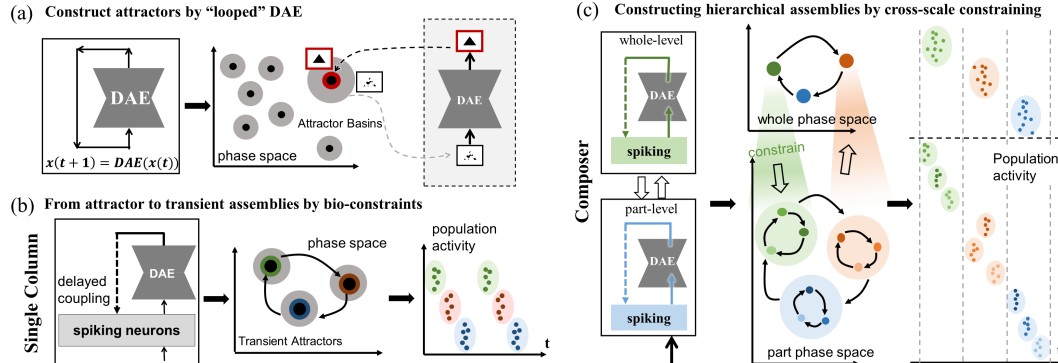

Figure 2: How does nested neuronal coherence emerge: an intuitive explanation. (a) Use denoising auto-encoders (DAE, right) to construct many attractors (middle) by connecting its input and output layers (left); (b) Single-level column model: Disrupt stable attractors to create sequences of transient attractors (middle), forming the dynamical basis underlining the neuronal coherence and rhythmic activity (right) by incorporating coupling delay and spiking neurons with a refractory period (left); (c) Multi-level Composer: Hierarchically nested neuronal sequences (middle), which underpin nested neuronal coherence (right), are facilitated by cross-level constraining between columns (left, middle) that are associated with different time scales.

## 2 NEUROSCIENCE-MOTIVATED REPRESENTATION HYPOTHESIS

A widely discussed hypothesis in neuroscience is that transiently active ensembles of neurons, known as "cell assemblies," represent distinct cognitive entities (Hebb, 2005). These cell assemblies are dynamically correlated and organized into sequences (Hebb, 2005) (Fig.1d), accompanied by gamma-band rhythms (Buzsáki & Draguhn, 2004). It posits that the oscillatory firing patterns of neurons give rise to two types of messages: (1) discharge frequency encodes the presence of features, and (2) the timing of spikes encodes feature "binding" (Singer, 2009; Malsburg, 1994). Consequently, temporally correlated spike firings "bind" distributed features (contents) into entities or neural words (structures). Furthermore, the hierarchical organization of assembly sequences may be regarded as a neural syntax, with the "nestedness" in hierarchical relationships represented as temporally nested structures of cell assembly trajectories, accompanied by nested neural rhythms (Buzsáki, 2010) (Fig.1d). Such nested states are of great interest in neuroscience and are argued to be prerequisites of consciousness (Northoff & Zilio, 2022). Together, these hypotheses motivate a coherence-based representation paradigm for part-whole hierarchy, introduced as follows.

**Spatial separation of levels**: We firstly assume that given a visual scene (Fig.1a), the contents or features of visual objects are concurrently represented at different "part-whole levels" (Fig.1c). Each level is supported by a set of "spiking neurons", whose receptive fields are the corresponding features.

**Correlation at different scales**: Secondly, we assume that feature neurons at different levels tend to form correlation patterns of different spatial-temporal scales, or synchronized cell assemblies of different sizes, to "bind" features into parts or wholes. It is indicated by the colored box from large to small in Fig.1c.

**Part-whole hierarchy as assembly sequence hierarchy**: Thirdly, we assume that the longer life time of each whole-level cell assembly covers the sequence of its part-level assemblies. As a result, the nested structure in a part-whole relationship is represented as the nested temporal structure of cell assembly sequences (Fig.1d, right).

## 3 INTUITION OF THE MECHANISM

How does a hierarchically structured assembly sequence emerge from randomness, with parts temporally nested within wholes (Fig.1d)? This challenge comprises two subproblems: (1) how assembly sequences at each level create neural words (parts/wholes), and (2) how assemblies at different levels nest to form neural syntax (hierarchical relationships).

To form assembly sequences, we need to balance positive and negative feedback: First, a group of neurons (cell assembly) is activated by positive feedback and then quickly inactivated by negative feedback, allowing the next assembly to compete for emergence.

**From DAE to attractor network**: To construct positive feedback, we explore the close relationship between denoising auto-encoder (DAE, (Vincent et al., 2008)) and attractor networks (Fig.2a), where denoising acts as a shared feature and a form of positive feedback. Specifically, by connecting the input and output layers of a DAE, we form a dynamical system: $x_{t+1} = DAE(x_t)$, where the DAE is trained on a large dataset. In this setup, each data sample serves as an attractor in the "bottom-up / top-down" dynamics (Fig.2a), since the decoder output of the DAE recovers the noiseless data sample from a perturbed input (Fig.2a middle). This denoising feedback from the DAE provides positive feedback to stably activate a group of neurons based on "associative memories" (Krotov, 2023) parameterized within the DAE.

**From attractors to rhythmic activities**: To establish negative feedback, we consider neuronal adaptation, a universal biological constraint intrinsic to each neuron. Specifically, as shown in Fig.2b left, a layer of spiking neurons with refractory periods connects the output and input of the DAE (also receiving external input). The constraints imposed by the refractory period disrupt the stability of the attractors in Fig.2a, forcing them to become transient. Additionally, to provide a time window for switching between alternative transient attractors (Fig.2b middle), we explore another biological constraint: coupling delay, which allows positive feedback to wait briefly. These biological constraints transform the stable attractors in Fig.2a into a cyclic sequence of transient attractors (Fig.2b middle), underpinning the rhythmic activity in the neuronal coherence state. See Zheng et al. (2022) for a theoretical proof.

**From assembly sequence to assembly hierarchy**: Inspired by neural syntax theory, we investigate a third biological constraint: integration time window. Buzsáki (2010) proposed that a hierarchy of time windows serves as the "syntax" for organizing cell assemblies, aligning with Mahjoory et al. (2019)'s recent findings of a gradient of time scales along the cortical hierarchy. Specifically, several columns in Fig.2b are stacked and interconnected (Fig.2c left). Higher-level columns have longer integration time windows to accommodate slower switching dynamics, while lower-level columns have shorter windows. To coordinate activities across levels, activated long-lived assemblies in higher-level columns constrain the dynamics of lower-level columns through gating, while the temporal integration of lower-level activities drives the dynamics of higher-level columns (Fig.2c middle). These biological constraints intuitively promote the emergence of nested neuronal coherence (Fig.2c middle). In the following section, we provide more details about the Composer.

# 4 MODEL

In this work, we focus on a whole level and a part level, examining the neuronal coherence of pixel-level features to simplify the problem. This framework can be generalized in future research (see A.1). Overall, the Composer comprises two levels of columns interconnected by top-down modulation and bottom-up integration (Fig.2c). Each column consists of a pixel-level spiking layer, referred to as the spike coding space (SCS, Fig.3c), and a denoising autoencoder (DAE). The term "pixel-level" indicates that the SCS at both levels matches the image dimensions ($d$). Within each column, the SCS and DAE are delay-coupled: the delayed feedback from the DAE modulates the firing rates of the spiking neurons through gating, consistent with dendritic computations of pyramidal cells in the cortex (Sherman & Guillery, 1998). Additionally, spikes in the SCS are integrated within a narrow time window before being fed to the DAE, acting as coincidence detectors (König et al., 1996). In the following sections, we first introduce each column individually and then explain how the columns are connected across levels.

## 4.1 PART-LEVEL COLUMN

Spiking neurons in $SCS_1$ of the part column receive inputs from three sources (Fig.3b): the external input image $x \in \{0, 1\}^d$, inner-column feedback $\gamma_1(t) \in \mathbb{R}^d$ and inter-column feedback $\Gamma(t) \in \mathbb{R}^d$ at each time step $t \in [0, T]$ (where $d$ is the number of image pixels and neurons in $SCS_1$):

$$\rho_1(t) = x \cdot \gamma_1(t) \cdot \Gamma(t) \tag{1}$$

Here, '·' denotes pixel-wise product and firing rate $\rho_1(t)$ determines the firing activity $s_1(t) \in \{0,1\}^d$:

$$P(s_1(t) = 1) = \rho_1(t) \cdot g_1(t - \hat{t}), \quad t \in [0, T]. \tag{2}$$

Here, $P()$ denotes the probability and $g_1(t - \hat{t})$ represents the relative refractory function of part-level neurons (Fig.3e), where $\hat{t}$ indicates the timing of the last spike firing event (for each neuron). After firing, the neuron enters an absolute refractory period of duration $\tau_{r1}$, followed by a relative refractory period lasting $\tau_{\delta 1} - \tau_{r1}$, during which its firing probability is suppressed by a factor $g < 1$ (Fig.3e). The inner-column feedback $\gamma_1(t)$ in Eq.1 is the denoised output of $\mathrm{DAE}_1$, delayed by $\tau_d$:

$$\gamma_1(t) = \mathrm{DAE}_1([I_1 * s_1](t - \tau_d)). \tag{3}$$

where $*$ is the convolution operator and $I_1$ is the integrative function for $s_1(t)$ with a time window of $\tau_1$ (Fig.3f): $[I_1 * s_1](t) = \sum_{\tau=0}^{\tau_1}(I_1(\tau) \cdot s_1(t - \tau))$. To clarify, the spiking activity $s_1(t)$ in the SCS is integrated over a short time window $\tau_1$ before being fed into $\mathrm{DAE}_1$, and the feedback from $\mathrm{DAE}_1$ to $\mathrm{SCS}_1$ is delayed by $\tau_d$.

## 4.2 WHOLE-LEVEL COLUMN

As the top level in the two-level Composer, the whole-level column does not receive top-down modulation from higher levels. The firing rate of spiking neurons in $\mathrm{SCS}_2$ is determined by the product of a driving term and a modulation term:

$$\rho_2(t) = (\lambda \cdot x + \overline{\lambda} \cdot D(t)) \cdot \gamma_2(t) \tag{4}$$

where $\lambda < 1$ is the factor of partial influence from skip connection (Fig.3c) and $\overline{\lambda} = 1 - \lambda$. $D(t)$ is the integrated spikes from the part-level $\mathrm{SCS}_1$, serving as the main driving input for the whole-level column. $\rho_2$ similarly determines the spike firing in $\mathrm{SCS}_2$ by:

$$P(s_2(t) = 1) = \rho_2(t) \cdot g_2(t - \hat{t}) \tag{5}$$

Besides, $\gamma_2$ in eq.4 is the delayed feedback from $DAE_2$ (similar to eq.3):

$$\gamma_2(t) = \mathrm{DAE}_2([I_2 * s_2](t - \tau_d)) \tag{6}$$

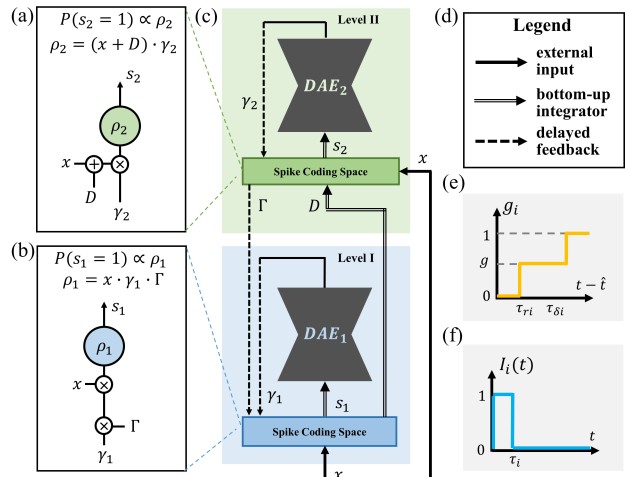

Figure 3: (a)(b) Pyramidal neuron models in the spike coding space of part level and whole level. $\otimes$ denotes multiplication and $\oplus$ indicates addition at the dendrites. (c) Detailed information flow. $x$ is the external input. Note that delayed coupling exists both within each column and between different columns. Levels are color-coded. (d) Legend for (c). (e) Relative refractory function $g_i$. (f) Integration function $I_i(t)$ with timescale $\tau_i$.

## 4.3 LINKING THE LEVELS

Up to now, we have introduced operations within each column of the Composer except for two variables: $\Gamma(t)$ in eq.1 and $D(t)$ in eq.4, which describe interactions between levels:

$$\Gamma(t) = [I_\Gamma * s_2](t - \tau_{d'}) \quad \text{and} \quad D(t) = [I_D * s_1](t). \tag{7}$$

where $I_\Gamma, I_D$ is the corresponding integration function. $\tau_{d'}$ is the cross-level feedback delay, which is set to be $\tau_{d'} = \tau_d$ for simplicity.

Each integration function $I_i$ has its corresponding time window parameter $\tau_i$ ($i \in 1, 2, \Gamma, D$), and each refractory function $g_i$ is characterized by an absolute refractory period $\tau_{ri}$ within the total refractory period $\tau_{\delta i}$ ($i \in 1, 2$). The general relationship among these parameters is: $\tau_1 < \tau_2 < \tau_D \sim \tau_\Gamma \ll \tau_d$ and $\tau_{\delta 1} \sim \tau_{\delta 2} \ll \tau_d$.

As suggested by neuroscience, the hierarchical organization of integration time windows and delay coupling are key mechanisms for cell assemblies (Buzsáki, 2010). These time-scale parameters are treated as hyperparameters. For further details and ablation studies, refer to A.5.2. Generally, the model's performance is robust to variations in these timescale parameters within a reasonable range. Lastly, $DAE_1$ and $DAE_2$ are pre-trained on part-object and whole-object datasets to perform denoising, using a BCE loss function. These pre-trained DAEs function like pre-configured wiring in the cortical columns (Buzsáki, 2019). Importantly, this pre-training only introduces priors for content (e.g., what an object looks like), not structures (e.g., how the hierarchy is organized or how parts or wholes are grouped). Therefore, the inference of part-whole structures is not guided by pre-training, but is achieved by the dynamics of the Composer in a content-sensitive manner. For more details on the training process, see A.6.

## 5 EVALUATION

Quantitative evaluation of the hierarchical structure is essential for verifying or falsifying models; however, it is often lacking in most related works (Hinton et al., 2018; Garau et al., 2022; Sun et al., 2021). The challenges arise from two main aspects: (1) real-world images typically lack explicit parsing structures, and (2) measuring distributed hierarchical structures is non-trivial. These challenges motivate us to develop synthetic datasets and metrics for explicitly evaluating the Composer.

### 5.1 DATASET

We created four synthetic part-whole datasets of varying complexities (Fig.4), each containing 60,000 samples. Each image comprises multiple whole objects, with each whole object consisting of clearly defined parts.

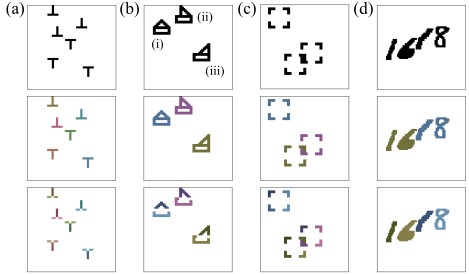

**Ts** dataset (Fig.4a) consists of 3 letter ⊤ and 3 reversed letter ⊥ as whole-level objects. Each ⊤ or ⊥ is composed of a horizontal bar and a vertical bar as parts. The $T$s dataset contains a greater number of whole objects in each scene. Similar stimuli are employed as target templates in perceptual tasks in neuroscience (Wolfe, 2021).

Figure 4: Exemplified samples in datasets: (a) Ts (b) SHOPs (c) Squares (d) Double-Digit MNIST. Top, input. Middle / Bottom, ground truth of Wholes / Parts.

**Squares** dataset (Fig.4c) consists of 3 randomly located squares as whole objects, each composed of 4 corners. The images in this dataset contain more parts compared to the previous one. Gestalt psychology uses similar stimuli to study illusory contours (Lee & Nguyen, 2001).

**SHOPs** (short for **S**hoes (ii), **H**ouse (i), **OP**era (iii) in Fig.4b) consists of 3 types of whole objects, each further composed of elementary rectangles and triangles. This dataset captures the complexity arising from overlapping parts that construct a whole, where overlapped pixels are not assigned to either part in the ground truth (Fig.4b, bottom) due to their ambiguity of belonging.

**Double-Digit MNIST** (Fig.4d) contains 2 randomly located double digits, with each consisting of 2 closely placed MNIST digits. This dataset exhibits objects of higher complexity and diversity.

### 5.2 SCORES

How can we evaluate the nestedness in Fig.1d (right)? The underlying idea is that correlated cell assemblies can be conceptualized as clusters of spike trains. Therefore, coherence measures for clustering methods, such as the Silhouette Score (Rousseeuw, 1987), are appropriate candidates. Accordingly, we introduce the Part Score and Whole Score to measure neuronal coherence within each level:

$$\text{Part Score} = \text{Silhouette}(s_1, s_1, \text{GT}_1, \text{D}_{vp}) \tag{8}$$
$$\text{Whole Score} = \text{Silhouette}(s_2, s_2, \text{GT}_2, \text{D}_{vp}) \tag{9}$$

Here, $s_1, s_2$ represent collections of spike trains, while $\text{GT}_1, \text{GT}_2$ indicate the part / whole-level ground truth for images (Fig.5a). Due to the one-to-one correspondence between image pixels and

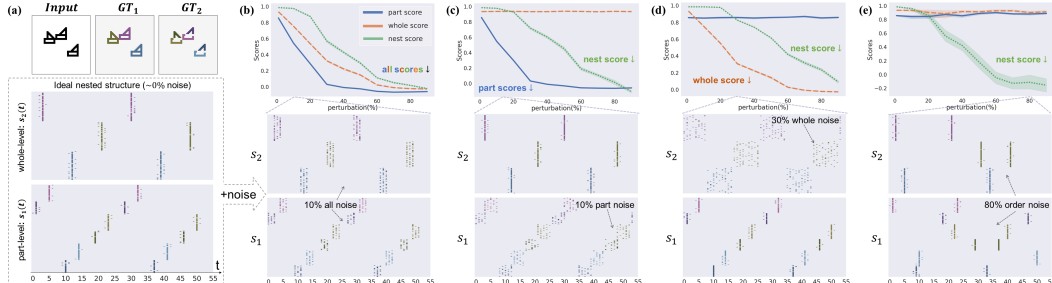

Figure 5: Intuitions of metrics. (a) Ideal nested structure (spike raster plot of two levels) given the input (top). If random noises are added to (c) part-level, (d) whole-level, or (b) both levels, corresponding scores degrades gradually from 1 to 0. If correlated noise are added to perturb the hierarchical relation among part assemblies and whole assemblies, only nest score degrades. (b)∼(e): Top, scores vs perturbation degree in each case; Bottom, exemplified perturbed spike pattern of certain degree. See Fig.13∼Fig.16 for details.

spiking neurons in SCS, these ground truths can also be applied to spike trains. They are used exclusively for evaluating representations, not for training purposes. The non-Euclidean Victor-Purpura metric, $D_{vp}$ (Victor & Purpura, 1996), evaluates distances among spike trains. $\text{Silhouette}(a, b, L, D)$ is the well-known Silhouette score (Rousseeuw, 1987) that quantifies inner-cluster coherence for sets $a$ and $b$ (with $a = b$ by default) based on cluster assignments $L$ (each sample in $a$ or $b$ has been assigned a cluster label $l \in L$) and metric $D$ (measuring distances between samples in $a$ and $b$). A higher Silhouette score signifies that data samples within the same cluster are closer together and further apart from those in different clusters, reflecting more synchronous spike firings that encode either parts or whole objects. This synchronization characterizes the degree of neuronal coherence "within" each level.

To assess the nested structure, we conceptualize nestedness as the average proximity of whole-level assemblies to all their constituent parts. It is crucial to consider $s_1$ and $s_2$ in conjunction to reflect cross-level nestedness. Accordingly, we generalize the Silhouette to capture cross-level coherence, defining $\text{Nest Score} \propto \text{Silhouette}(s_1, s_2, GT_2, D_{vp})$, which quantifies the coherence between part-level spike trains $s_1$ and whole-level spike trains $s_2$ based on the whole-level cluster assignment $GT_2$ (refer to A.4 for a comprehensive explanation). Notably, the pixel-wise correspondence among $s_1, s_2, GT_1, GT_2$ ensures comparability.

To elucidate the scores' intuition, we systematically introduce noise to a manually crafted ideal nested structure (Fig.5a) and monitor score changes. Perturbing only the part level leaves the Whole Score unchanged, while the Part Score and Nest Score decline from 1 to 0 (Fig.5c); the reverse is true when the whole level is perturbed (Fig.5d). Perturbing the relative order of part and whole assemblies to disrupt nestedness significantly reduces the Nest Score, with the Part and Whole Scores remaining stable (Fig.5e). Collectively, these three scores comprehensively reflect the structure of nested neuronal coherence. More details are provided in A.4.5. Similar to the Silhouette Score, the optimal coherence score is 1, with -1 indicating incoherence, and scores near 0 suggesting randomness.

## 6 EXPERIMENTS

### 6.1 QUALITATIVE RESULTS AND VISUALIZATION

**Emergence of the "Neural Syntax" in SCS**. Fig.6 illustrates a simulation on a random sample from the SHOPs dataset. The convergence of the Part Score, Whole Score, and Nest Score in Fig.6b indicates that the Composer progressively establishes neuronal coherence, where parts and wholes are represented as synchronized cell assemblies. This is further depicted in Fig.6c (right) and Fig.6d. In the final phase III, shown in Fig.6c (right), a periodic emergence of three two-level binary trees is observed, corresponding to the three SHOPs objects in Fig.6a. One of these trees is delineated by deep blue (for the whole object) and light blue (for part objects) boxes. The spikes in Fig.6c are sorted and colored according to ground truth (Fig.6a) to enhance visualization. Notably, the representation

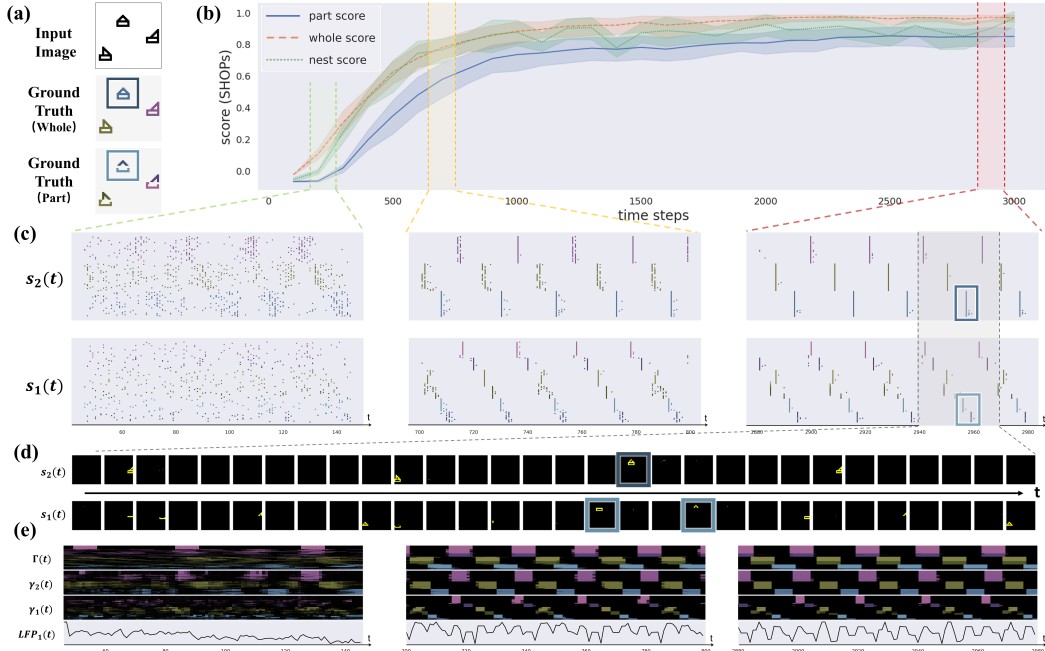

Figure 6: Emergence of the part-whole hierarchy with nested neuronal coherence. Exemplified by one SHOPs sample. (a) Input image and ground truth; (b) Evolution of the Scores. (c) The spike raster plot of three selected phases in (b): phase I (initial, green box), phase II (middle, yellow box), phase III (final, red box). $s_2(t)$, $s_1(t)$ stand for spiking representations in SCSs of whole/part levels. (d) zoomed in spiking pattern during the period marked by black box in (c), to visualize what each cell assembly represents (e.g. blue boxes in (c),(d)and(a)). (e) Evolution of the top-down attention maps and the local field potential (LFP) during the three phases in (b).

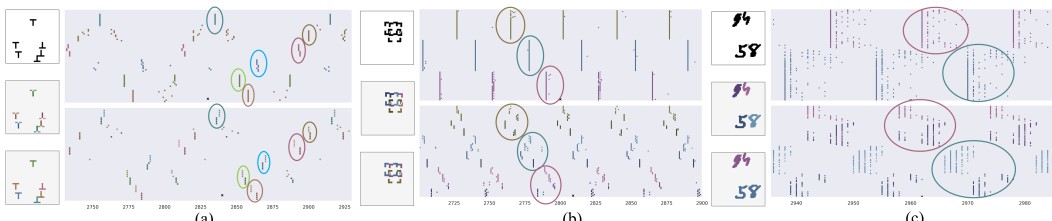

Figure 7: Visualization of the emergent part-whole hierarchy on other datasets: (a) Ts (b) Squares (c) Double-Digit MNIST. (In each sub-figure) Left: input image, part-level ground truth and whole-level ground truth. Right: spike raster plot in final phase III, top for whole level and bottom for part level. Cell assemblies are circled for clarification. See Fig.22~Fig.29 for complete visualizations

structure (Fig.6c, right) is continuously inferred from initial uncertainty (Fig.6c, left), based on the self-consistency among contents. For further visualization results, refer to A.9.3.

**Emergence of the Hierarchical Attention Sequence** is observed in Fig.6e: Initially starting from a state of randomness (left), the cross-level feedback $\Gamma$ and the inner-level feedback from DAEs $\gamma_1, \gamma_2$ progressively converge towards a sequential trajectory (Fig.6e). The high degree of similarity between the assembly sequence (Fig.6c) and the attention sequence (Fig.6e) suggests a strong connection between predictive top-down feedback and oscillatory cell assemblies, in alignment with the findings from neuroscience research (Engel et al., 2001). Consequently, the top-down attention mechanisms of the DAE, in conjunction with the bottom-up integration of spike activity within SCS, facilitate the concurrent emergence of both the attention sequence and neuronal coherence, as well as the rhythmic population activity denoted as $LFP_1$ (Fig.6e).

**Visualization Results on Other Datasets** are shown in Fig.7. Interestingly, the emergent coherence structure varies across different datasets. In the Ts dataset (comprising 6 Ts), 6 binary trees emerge periodically, while in the Squares dataset (comprising 3 Squares), 3 quadtrees emerge. In the Double-Digit MNIST (comprising 2 Double-MNISTs), 2 binary trees emerge. Overall, the Composer effectively and flexibly represents the part-whole hierarchy of scenes with diverse complexities.

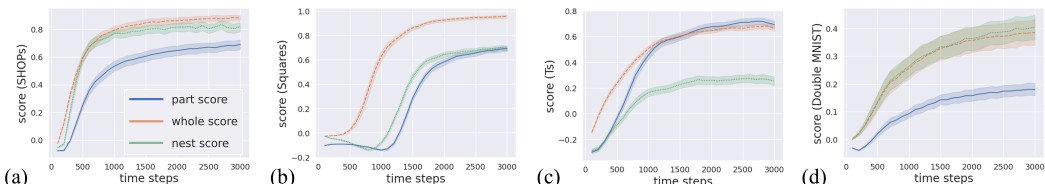

Figure 8: Convergence of Scores. (a) SHOPs (b) Squares (c) Ts (d) Double-Digit MNIST.

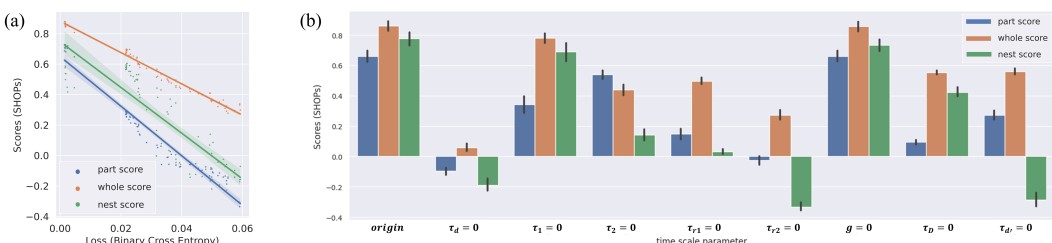

Figure 9: Ablation study on SHOPs. (a) Loss vs Scores (b) Ablation of time scale parameters. See Fig.17, Fig.20 for more results.

## 6.2 QUANTITATIVE ANALYSIS

**Score Convergence** across iterations is assessed using 100 random samples per dataset, as depicted in Fig.8. Notably, scores converge consistently across all datasets with negligible error bars, though the process varies. In Squares (Fig.8b), whole objects group more rapidly than parts, mirroring human visual processing (Lee & Nguyen, 2001). In Ts (Fig.8c), the large object number places high demands on cross-level coordination, resulting in a lagging Nest Score. In Double-Digit MNIST (Fig.8d), the Composer faces greater challenges in distinguishing part-level MNIST digits, which is partially attributable to the dataset's diversity. See A.6.5 for details.

**Benchmarking**. We compare the Composer with a recently implemented state-of-art model, the Agglomerator (Garau et al., 2022), which also aims to leverage the concept of neuronal coherence (islands of identical vectors) to group neuronal representations at different levels into a hierarchical structure. Using 1000 random samples and 5 random seeds, we evaluate both the Composer and the Agglomerator across the four datasets. The coherence-based metric is naturally extended to assess the perfor-

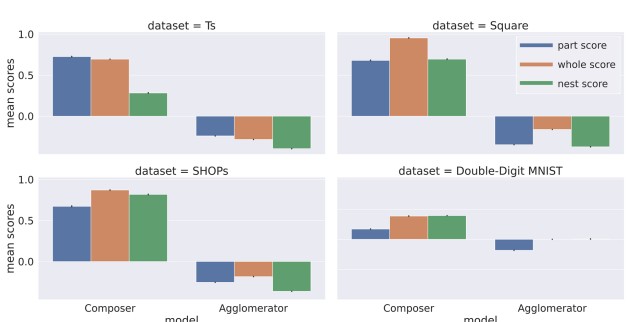

Figure 10: Benchmarking results on four datasets

mance of the Agglomerator. As depicted in Fig.10, the Composer consistently outperforms the Agglomerator on all datasets. Moreover, the Agglomerator did not successfully generate object-centric representations at each level, which are essential prerequisites. This raises concerns about the reliability of its parsing capabilities. For more details on the benchmarking, see Section A.8.

**Ablation 1: DAE vs Coherence**. Given that denoised feedback from DAE is a critical component in the Composer, it is useful to investigate the correlation between the denoising efficacy of DAE and the resulting scores. To this end, we pre-trained 100 pairs of DAEs on the SHOPs dataset–each consisting of a part-level and a whole-level DAE–using an identical architecture but varied learning rate, resulting in a range of denoising efficacies. These pre-trained DAE pairs were then integrated into the Composer for parsing tasks. Fig.9a illustrates the relationship between the denoising loss of the DAEs and the coherence scores. The figure indicate a strong positive correlation between lower denoising loss and higher scores across all metrics, suggesting that the effectiveness of denoising feedback is directly linked to the nested neuronal coherence achieved by the Composer.

**Ablation Study 2: Timescale Parameters**. A key component within the Composer is the bio-inspired constraints, primarily implemented as a series of time-scale parameters. To ascertain the contribution of these constraints to the Composer's performance, we conducted an ablation study on these parameters using SHOPs dataset, as shown in Fig.9b, where each parameter is individually ablated by setting it to zero. For instance, $g = 0$ indicates the elimination of the relative refractory period and $\tau_d = 0$ suggests the removal of inner-column feedback delay. In comparison to the original model, all ablated models exhibit reduced scores. Notably, the removal of the inner-column feedback delay $\tau_d$, the whole-level refractory period $\tau_{r2}$, and the cross-level feedback delay $\tau_{d'}$ results in particularly detrimental effects, as evidenced by the reversal of the Nest Score. The results suggest that the coupling delay, refractory period, and integration time window are all key factors that contribute to the performance of the Composer. Further details on the ablation study can be found in A.5.3.

# 7 RELATED WORK

Slot-based models represent a significant research direction for object-centric representation using pre-defined discrete slots, as evidenced by works such as (Greff et al., 2015; 2016; 2017; 2019; 2020; Locatello et al., 2020). This approach has also been extended to generate hierarchical structures in graph neural networks (Han et al., 2022; Xu et al., 2017), generative models (Deng et al., 2021), attention-based models (Sun et al., 2021; Fisher & Rao, 2022) or capsule networks (Hinton et al., 2018). Nonetheless, the discrete and pre-defined nature of slots inherently restricts their capacity to infer representation structures continuously from uncertainty in a content-sensitive manner, such as determining the presence or number of objects. On the other hand, it is not clear how human brain implements the boundary across slots and the weighting sharing required among slots (Greff et al., 2020). Consequently, it poses a limitation in developing vision systems that closely mimic human perception.

In this paper, we investigate mechanisms that eschew discrete and predefined slots in favor of continuous, content-sensitive neuronal coherence. Although this concept has been sparingly explored in machine learning, there are several intriguing related studies that define coherence in various ways: (1) through the phase of complex values (Löwe et al., 2022), (2) by the timing of spike firings (Zheng et al., 2022), and (3) via the rotation of features (Lowe et al., 2023). However, these works primarily focus on single-level objects, neglecting the part-whole relationship. The Composer extends the concept of spike timing to accommodate the part-whole hierarchy, but it is not apparent how the complex-valued network (Löwe et al., 2022) and feature rotation (Lowe et al., 2023) could be adapted to represent different levels of objects. This is because both approaches heavily rely on a specific form of activation function as a binding mechanism, which lacks free parameters to differentiate between levels, unlike the time scale parameters in the Composer. Finally, the ideology of identical islands of vectors, as proposed by Hinton (2021) and implemented by Garau et al. (2022) holds promise for representing part-whole relationships through the coherent orientation of vectors. However, despite its ambitious goals, this approach currently lacks rigorous quantitative assessments to evaluate the hierarchical structures, which are crucial for validating or challenging the representation hypothesis.

# 8 CONCLUSION

In this paper, we look for solutions to continuously represent part-whole hierarchy by neuronal coherence. To achieve this, we merge insights from neuroscience with machine learning techniques to create the cortical-inspired Composer. This system dynamically build-up "neural syntax" of "neural words" through iterative bottom-up coincidence detection and top-down predictive attention. We anticipate that the Composer will pave the way for a new paradigm that unifies neuroscience theories with machine learning methods, aiming to develop a more human-like visual processing system in the future. For a discussion on limitations and potential future directions, See A.1.

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

## A  APPENDIX

In the main text, we leave out several details to make things self-contained within limited pages. These details are organized in the Appendix as followings:

### A.0.1  GENERAL DISCUSSIONS

We first discuss the **"Limitation"**, Broader Impact, and potential Future Work of this paper in Section.A.1 and Section.A.2. Followed by showing the **"Code Availability"**, Section.A.3, where the detailed realization of the model and main results can be found.

### A.0.2  METRICS

Developing metrics to evaluate hierarchical relationship is a major challenge for this work. In the main text, we mainly provide the intuitions of what is the metric and what each score measures. In Section.A.4 (**"Metric"**), we provide a step-by-step introduction to how metrics are developed and how can these metrics be generalized to account for broader works in the future.

### A.0.3  HOW THE COMPOSER WORKS

While the intuition of the mechanism and the detailed architecture are introduced in the main text, several additional information might be helpful to further capture the picture of how the Composer works: (1) how the Composer is initialized? (2) What is the detailed configuration of time scale parameters and the robustness of the Composer to varied time scale settings (ablation study and sensitivity test). What is the contribution of time scale parameters to the Composer? (3) The training details of DAE and the contribution of DAE to the Composer. (4) To what extent the Composer is bio-plausible?

To resolve these potential questions, we show model details in Section.A.5 $\sim$ Section.A.7, including "Initialization" in Section.A.5.1 and the **"Time Scale"** in Section.A.5.2. After that, we provide **"Training"** details in Section.A.6. Additional ablation studies of times scale parameters and DAE are provided in Section.A.5.3∼Section.A.5.4, and Section.A.6.5 respectively. Then we also enumerate all the **"Biological Motivations"** and neural correlates of the Composer in Section.A.7.

### A.0.4  BENCHMARK

We provide a brief introduction of the "Benmark" model and how the evaluation scores are naturally generalized to this "non-spiking" model to evaluate the part-whole hierarchy in Section.A.8.

### A.0.5  ADDITIONAL DISCUSSION OF THE MAIN-TEXT EXPERIMENTS AND ADDITIONAL RESULTS

If the reader has confusions on the detailed setting about the main text experimental figures. We provide supplementary discussions in Section.A.9.

### A.0.6  MORE VISUALIZATIONS

We illustrate more detailed visualization results in Section.A.9.3 on different datasets (Fig.22 to Fig.29). Failure cases are included.

### A.0.7  ADDITIONAL RESULTS

We lastly provide two additional results to provide a preliminary answer to questions that may arise when reading the main text: (1) Can composer generate different part-whole relationships given an ambiguous image like Fig.1 and (2) Can the architecture in Fig.3 be extended into more levels? Results on multiple solution can be found in A.9.4 and Fig.30. Results on more levels can be found in A.9.5 and Fig.31.

### A.0.8 ANIMATION VIDEO

Since the dynamical process is best illustrated by dynamical visualizations, we provide a zip file containing videos visualizing the dataflow of the Composer in SI , about 60MB to be downloaded.

## A.1 LIMITATION AND FUTURE WORK

In this section, we highlight several limitations that could be addressed in future works.

**(1) Pixel-wise SCS as low-level feature layer and the Scalability**

In this work, the part-whole hierarchy is represented and evaluated based on the pixel-wise spike coding space (SCS), which has a one-to-one mapping to the image's pixel space (topographical mapping). Shared pixel-wise spatial map of $s_1, s_2, \mathrm{GT}_1, \mathrm{GT}_2, x, \gamma_1, \gamma_2, \Gamma$ to the input image $x$ makes the spiking representation and the top-down attention much more interpretable so that the part-whole hierarchy could be explicitly evaluated and visualized as Fig.6, Fig.7. Also different object-level spiking pattern can be compared to compute the Nest Score. While topographical mapping (a one-to-one spatial relation between internal representation to the external physical world) is a common feature of the cortical representation, the representation (at each spatial point) could be more abstract. For example, instead of a binary neuron associated with the "presence" of an object occupying the location, a population of binary neurons can be assigned to each location, so that different aspects of features (associated with the object at the corresponding location) could be accounted for by the population vector. Assigning a population of neurons with locations is similar to the capsule idea in Capsule Network (Hinton et al., 2018) and the mini-column organization in Hinton (2021). This suggests a future direction to combine the neuronal synchrony (temporal coherence) with identical islands of neurons (spatial coherence) in the GLOM architecture.

Besides, the object could be represented in the latent layer of the DAE instead of the pixel-level SCS in each column, similar to the Slot Attention(Locatello et al., 2020). Representing objects in the latent layer could enable transforms between levels as in GLOM, by replacing pixel-wise gating in this paper into a neural network that potentially parameterizes a coordinate transformation. It is notable that the underlining hypothesis is that all feature-levels and object-levels share a "topographical" spatial map. The difference is what is being represented at each location (along feature level) and how representations at different locations are correlated (along object-level). This is a natural generalization of the representation hypothesis in Fig.1 in the Main Text.

Besides, the input to the SCS of the lowest level is not restricted to be the pixel-level image but could be the output feature map of an encoder, which is called tokenization in Agglomerator (Garau et al., 2022). This kind of generalization has also been discussed in Hinton (2021). **Notably, recent work shows that the gap between object-centric representation models (previously focused on toy datasets) and real-world images can be bridged by pre-trained DINO model**(Seitzer et al., 2022; Lowe et al., 2023). By applying the pretrained DINO model (Caron et al., 2021) as front-end, original model only need to deal with much lower-dimensional latent representations, which have more similar structure as synthetic datasets. Seitzer et al. (2022) and Lowe et al. (2023) provides the insights that seemingly toy models can be scaled to account for real-world dataset by applying proper front-end encoders. For Composer, a general-front end is also applicable and how to scale the model to account for neural syntax of more complicated scenes is a line of promising future works.

**In sum**, the limitation of the part-whole hierarchy as a pixel-level relationship in pixel-level SCS could potentially be generalized in three directions: (1) To allocate each location a column of spiking neurons to form a "representation column" at each location. (2) replacing the simplified cross-level interaction between pixel-wise SCSs as a proper neural network between latent layers. (3) the input to the model could be generalized to the tokenized embeddings from the front-end encoder (e.g. DINO (Seitzer et al., 2022; Lowe et al., 2023; Caron et al., 2021)). Notably, all these generalizations are compatible with the Composer and could be explored as future works.

**(2) Coordination transformation**

As originally motivated in a cognitive science view point (Hinton, 1979) and restated in machine learning literature (Hinton, 2021), part-whole hierarchy contains two challenges: (1) the dynamical emergence of the part-whole tree structure and (2) the implementation of a part-whole coordinate transformation. The insight behind this paper is that the first challenge is the core challenge of the problem while the latter one could be solved by implementing the transformation as a neural network. In other words, the flexible forming of a symbolic tree structure (capable of capturing the basic nested part-whole relationship) within a pure neural network is the hard problem that challenges the neural network models. The second problem, implementing a coordinate transformation, is more compatible with the neural networks: such transformation could be realized as a (feedforward) neural network.

In this work, we focus on how to represent the part-whole hierarchy within a pure neural network model through emergent nested neuronal coherence. The coordinate of objects is assumed to be already aligned in a pixel-wise manner between the whole-level and part-level so that the coordination transformation is reduced to the inclusion mapping (similar to identity mapping). However, since the parsing tree is realized within a pure neural network, the mechanism is compatible with more general coordination transformation: it could be realized by replacing the pixel-wise gating with a trainable neural network, which parameterizes the coordinate transformation.

**(3) How many level are there?**

In the representation hypothesis, we separate the entire representation space into discrete number of object levels. Therefore, different levels are explicitly distinguished before-hand. However, different from "discrete slots to represent object", the discretization of representation into levels is a reasonable setting. The underlining hypothesis is that while object representation can be diverse and uncertain, human-vision only accounts for very limited finite number of levels at each instant. Although there are many object levels in the external world (from galaxy to atom), only a limited fraction of these levels are mapped to the internal representation space to form perception. For example, Hinton (2021) argued that 5 level is enough to account for human vision. The part-level and whole-level is a relative concept, so that a given object can be represented in either level based on the context. In other word, the internal finite number of object levels can be flexibly reused to account for different external object levels. This argument is the motivation to discretize the representation space explicitly into object-levels. But at each object level, objects are represented in a distributed manner, by neuronal coherence, instead of slots. As a result, the hierarchical relationship is also represented in a distributed manner.

In this paper, we show the part-whole hierarchy of two levels: whole and part. However, this minimal structure could be naturally extended to account for more levels, since the form of interaction between levels is mostly irrelevant to how many levels are there or which level it is in. All levels could share a unified form of cross-level interaction and within-level interaction. Therefore, by stacking the columns along the hierarchy, more levels are accounted for. As discussed in Hinton (2021), up to five levels are sufficient to realize human-like vision.

**(4) Synthetic image**

In this paper, we use synthetic images to demonstrate how to represent the part-whole hierarchy. The benefit of using the synthetic image is that: (1) a common sense reasonable part-whole relationship is known beforehand as ground truth, therefore it is more convenient to explicitly evaluate the representation and test the capability. (2) The ground truth assignment of objects (part/whole) is known, which could be utilized to evaluate the neuronal coherence. The weak side of a ground truth is that such explicit assignment of part-whole ignores the ambiguity of parsing the scene: the parsing could depend on many factors like prior knowledge, attention, goal, internal state, and so on. Besides, parsing a real-world image without explicit part-whole hierarchy might be challenging for other reasons (overlap, background, etc.). However, recent models (Hinton et al., 2018; Sun et al., 2021; Garau et al., 2022) that claim to solve the part-whole problem actually resemble performing hierarchical feature extractions. Such confusion is partly due to the ambiguity of the part-whole relation, object definition and the complexity of features in the real-world images, which confuse the symbolic structure. Therefore, taking the present status of the problem[1] and the challenges the problem implicates[2] into account, one desirable roadmap is to focus on explicit evaluation based on synthetic data first (so that it is easier to interpret whether the mechanism works) and then gradually generalize the outcome to increasingly complex datasets in the future.

**(5) Learning scheme**

In this work, we treat the "sense" of what the object should look like as prior knowledge embedding in the parameters of DAE's weight, which in turn determines the dynamical property of the Composer. Indeed, such prior of prototype is needed for humans to parse a visual scene as well. For example, given a visual scene of a face (Fig.1a), a human observer should have already had the concept of

---

[1] Representing the part-whole hierarchy in a pure neural network is still an unsolved problem (Hinton, 2021)

[2] In essence, the part-whole problem requires a general solution to the sub-neuro-symbolic architecture and to realize hierarchical split of computational problem in a divide-and-conquer way. This paper explores the temporal aspect of the solution.

the eye, the nose, and the mouth in their mind, so that a face is parsed into wholes and parts. Here, "what each object looks like" are treated as priors of content and the emergence of parsing structure is treated as the inference of the representation structure. Therefore, in this work, we consider how a hierarchical structure could emerge as hierarchical neuronal coherence in neural networks given the prior knowledge of the content of objects. On the one hand, some of the priors are indeed hard-wired in the brain through a long period of evolution (related to Gelstalt psychology (Wagemans et al., 2012), like proximity, similarity, enclosure, continuation, closure, symmetry, common fate, etc.); on the other hand, some of the others may be gradually learned during evolution. Therefore, the learning scheme could be updated to capture how the part-whole hierarchy could emerge during the unsupervised perception of the multi-object world.

One recent work to generalize DASBE shows that the pre-training (Zheng et al., 2022) in DASBE can be naturally generalized to end-to-end unsupervised learning (Zheng et al., 2023) with an unsupervised loss function to predict the external input (predictive coding). The oscillatory cell assemblies also emerges, without any prior knowledge even for content! The insight is that: the model architecture in this paper (delay-coupled column organization, time scale hierarchy) could be regarded as inductive bias to encourage a hierarchically factorized representation during the unsupervised training of the whole model as a recurrent spiking neural network to reconstruct what it sees on average during a temporal period. More details are discussed in Section.A.6.4. **It is promising future work to figure out whether the "neural syntax" can be learned by simply predicting the visual input**. This picture is consistent with the neuroscientific hypothesis of "Cognition from Action"(Buzsáki et al., 2014). Therefore, we could treat the Composer as a constructed solution of the "neural syntax" by combining machine learning and neuroscience, and this solution provides the insights on how to learn the solution from data via unsupervised training, similar to from constructing the "neural words" in Zheng et al. (2022) to learning the "neural words" in Zheng et al. (2023). Taken together, we hope the Composer provide a new paradigm to combine machine learning models[3] and neuroscientific hypothesis (cell assemblies and cortical dynamics) to (1) solve part-whole hierarchy in machine learning field (in a bio-plausible way) and (2) understand how neural syntax emerges by predicting the world (action or attention).

## A.2 BROADER IMPACT

On the positive side, the model parses objects with neuronal coherence in the spike coding space composed of spiking neurons without explicit supervision. The mechanism by principle is not limited to a certain modality or certain object type. Thus, it may help develop human-like perception systems. Besides, with biological relevant features (eg. delayed coupling) and phenomena (eg. synchrony / neuronal coherence), the model may also act as a data-driven biological model to understand the perception process in the brain.

On the negative side, since the model is not supervised, it is harder to control what it learns. The current model is only trained on simple synthetic datasets and learns to group at the superficial pixel level, therefore the representation is highly explainable. However, grouping in latent space on real-world datasets requires to develop evaluation and visualization methods to make the representation in latent space more understandable. We believe this may serve as a step toward more transparent and interpretable predictions.

## A.3 CODE AVAILABILITY

The source code for results in the paper and a video demonstrating the whole simulation process can be found at:

https://drive.google.com/drive/folders/1GTHhpdafze6rExjD9NtV8beLfruMCMJR?usp=sharing.

---

[3]DAE can be generalized into more advanced models like BERT, Diffusion model, etc, which also perform denoising.

## A.4 METRIC

In the main text, we introduced that the cell assemblies are treated as clusters of spike trains and therefore, the neuronal coherence is measured as the inner-cluster coherence of clusters, based on ground truth assignments. In this section, we further unfold how the metrics for quantitative evaluations are defined based on the Silhouette Score, including the Part Score, Whole Score, and Nest Score. Since the Silhouette Score is based on the similarity measure among samples, we first introduce how the similarity among spike trains is measured, where the Victor-Purpura metric shows up (Section.A.4.1). Then we introduce the Silhouette Score and how it could be extended to account for varied aspects of the part-whole representation (Section.A.4.2~Section.A.4.4). Detailed perturbation study of metrics is included in Section.A.4.5. Finally, we discuss how the metrics can be generalized to evaluate other models with similar attempts to group distributed representation into objects by certain similarity measures (identical islands of vectors in Hinton (2021)), in Section.A.4.7. Hyper-parameters for evaluation is shown in Table.1.

### A.4.1 HOW TO MEASURE THE DISTANCE BETWEEN SPIKE TRAINS: VICTOR-PURPURA METRIC

The Victor-Purpura metric (VP-metric) is a classical non-Euclidean metric to measure the distance between arbitrary spike trains for evaluating the temporal coding in the visual cortex (Victor & Purpura, 1996). The motivation is that spike trains can have varied length and of binary value, so that the distance measure (like Euclidean metric) defined on fixed-length real-valued vector does not apply, especially to capture the precise temporal structure of spikes. The idea is that spike train can be treated as (binary) "Strings" and the distance from one spike train to the other can be defined as the number (cost) of "operations" to transform one spike train to the other ("Edit distance" of Strings).

In Victor & Purpura (1996), three types of operations are identified (Fig.11): 1. add a spike (cost=1); 2. delete a spike (cost=1); 3. shift a spike for length $\Delta t$ (cost=$\Delta t/\tau$), where $\tau$ is a parameter to control the temporal precision of the spiking code (or the temporal sensitivity of the metric). By sequentially applying the three operations ($T(u)$ in Fig.11), a spike train can be transformed to the other. The Victor-Purpura distance is defined as the minimal cost to transform a spike train to the other (Fig.11):

$$D_{VP}(s_i, s_j; \tau^{-1}) = min_T(\sum_{u=1}^{|T|} cost(T(u))) \tag{10}$$

$$T(u) \in \{\text{delete}, \text{add}, \text{shift}\} \tag{11}$$

$$cost(\text{delete}) = cost(\text{add}) = 1 \tag{12}$$

$$cost(\text{shift}, \Delta t; \tau) = \Delta t/\tau \tag{13}$$

where $T(u), u = 1...|T|$ is a sequence of basic transformations to transform $s_i$ to $s_j$ (or vice versa). The costs of the three basic transformations are different. The most special one is the shift operation: there is a time scale parameter to control the punishment of shifting the spike to its neighbourhood.

Notably, it is proved that the definition satisfies the three principles of a metric: positivity, symmetry, and triangle inequality (Victor & Purpura, 1996). Thus, it induces a metric space of arbitrary spike trains, even if not embedded in a vector space of specified dimension. Since spike trains are non-Euclidean in nature, the VP-metric provides a more direct measure of these entities. The minimal cost is computed through a dynamic programming method.

A desirable feature of VP-metric is that the parameter $\tau$ explicitly controls the temporal sensitivity of the metric and the expected temporal precision to be considered. If the $\tau$ is chosen to be $\infty$, then shifting a spike will cause no cost ($1/\infty \sim 0$). Thus the distance is exclusively due to spike count[4], therefore spiking rate is measured. If the $\tau$ is chosen to be 0, then it measures the number of spikes

---

[4]All transforming cost comes from adding/deleting spikes

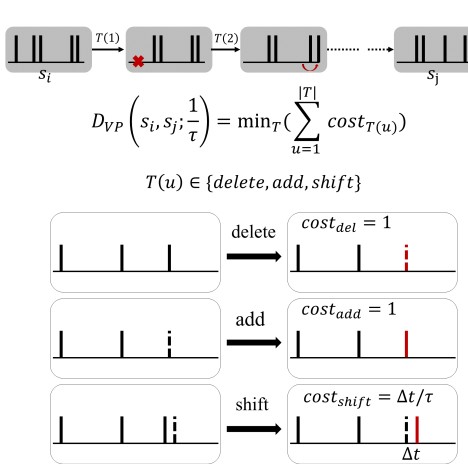

Figure 11: The Victor-Purpura metric. Top: a sequence of transformations; Middle: the distance between $s_i$ and $s_j$ is defined as minimum cost to transform one to the other; Bottom: three basic operations and their cost. Dashed bars are imaginary spikes which has been removed or has not been added. Red bars highlight the results due to the transformation.

that are not in absolute synchrony [5]. So small $\tau$ measures the spike train distance based on the very precise temporal synchrony structure. By varying the $\tau$, it is plausible to find the optimal coding scheme of the visual cortex (Victor & Purpura, 1996). In sum, $\tau$ is treated as a timescale parameter, to control the precision of temporal coding. In this paper, the part level is evaluated with smaller $\tau$ (part-level time scale) while whole level are evaluated with larger $\tau$ (whole-level time scale). The nestedness between the levels is evaluated with the time scale of the whole level (the child node should stay within the time window of their parents.)

### A.4.2 General Coherence Measure of Clusters: Silhouette Score

The Silhouette coefficient (Rousseeuw, 1987) is a score to evaluate the quality of clustering by measuring the inner-cluster coherence. Given a clustering assignment, the score is calculated using average intra-cluster distance (a) and average nearest-cluster distance (b). The score is computed as $(b - a)/\max(a, b)$. The document can be found at https://scikit-learn.org/stable/modules/generated/sklearn.metrics.silhouette_score.html. The best score is 1 and the worst score is -1. The values near 0 indicate overlapping clusters.

### A.4.3 Victor-Purpura Metric + Silhouette Score

How could the neuronal coherence be measured? Given ground truth assignments of clusters (each neuron belongs to which part or whole object), neuronal coherence (synchrony) is measured as the inner-cluster coherence of spike trains: the "inner-cluster" similarity and "inter-cluster" separability.

Specifically, if we take (1) each neuron as a **sample**, (2) the spike train of each neuron as **features**, (3) the ground truth assignment as the **clustering assignment** (each pixel on ground truth correspond to a neuron in SCS), (4) the VP-metric as the **distance measure**, then, the inner-cluster coherence (Silhouette Score) of the ground-truth-induced cluster assignment is exactly the coherence measure of neuronal coherence. In other words, the high Silhouette Score indicates that the spike trains of neurons of the same cell assembly are closer to each other in terms of VP-metric, which can be interpreted as neurons in the same cell assembly synchronizing better. Therefore, the VP-induced Silhouette Score sufficiently measures the grouping quality. The VP-induced Silhouette Score is also from -1 to 1. The best value is 1 (perfect grouping) and values near 0 indicate overlapping clusters (purely random firing without any temporal structure). Negative values generally indicate that a sample has been assigned to the wrong cluster, as a different cluster is more similar (neurons systematically synchronize to incorrect groups).

---

[5]only total synchronous spike trains have 0 distance while the slight shift of spikes has cost 1

During the whole simulation, only a segment of simulation (after the convergence) is used for evaluating the Silhouette Score, See Table.1. The length of segment can be flexibly selected as long as it covers at least one oscillation period. In this work, we simply select one large enough length value.

### A.4.4 SCORES TO MEASURE THE PART-WHOLE HIERARCHY

Coherence scores are all defined based on the VP-Silhouette Score and Ground Truth assignment.

**Part Score** is defined as the VP-Silhouette score with respect to the part-level spiking pattern and the part-level ground truth assignment:

$$\text{Part Score} = \text{Silhouette}(VP(spk_1, spk_1; \tau_p), \text{label}_1) \tag{14}$$

where $spk_1 \in \{0,1\}^{(N,\tau_l)}$ means the total spike trains in level 1 (part-level). $\tau_l$ is the length of each spike train for evaluation and $N$ is the number of (activated) neurons at part level. $VP(spk_1, spk_1)$ is the distance matrix whose elements $(i,j)$ are the VP-distance between the i-th spike train and the j-th spike train in the part level; The $\text{label}_1$ means the ground truth assignment of neurons in level 1 (part-level). $Silhouette$ is the Silhouette Score. $\tau_p$ is close to the (integration) time constant of part-level ($\tau_1$) (Table.1), controlling the temporal sensitivity of VP-metric (eq.10). Therefore, the Part Score measures the coherence level of the part level exclusively, independent of the activity in the whole level. Part Score indicates the quality of the grouping of tree nodes in the part level (Fig.14). For example, $Part - Score = 1$ indicates that neurons are synchronized perfectly into separated groups corresponding to the part objects. On the other hand, $\text{Part Score} = 0$ indicates that the neurons fire randomly and no temporal structure emerges (Fig.13∼Fig.15, bottom). In rare cases, $\text{Part Score} < 0$ indicates that (on average) neurons with different assignments are synchronized and neurons with the same assignments are not synchronized (Fig.16, bottom). Equation.14 is equivalent to the definition of Part Score in the main text, but eq.14 reveals how the Part Score is practically implemented. Same for other cases.

**Whole Score** are similarly defined as:

$$\text{Whole Score} = \text{Silhouette}(VP(spk_2, spk_2; \tau_w), \text{label}_2) \tag{15}$$

where $spk_2 \in \{0,1\}^{(N,\tau_l)}$ means the total spike trains of level 2 (whole-level). $VP(spk_2, spk_2)$ is the distance matrix whose elements $(i,j)$ are the VP-distance between the i-th spike train and the j-th spike train in the whole level; The $\text{label}_2$ means the ground truth assignment of neurons in level 2 (whole-level). $\tau_w$ is close to the time constant of the whole-level $\tau_2$: $\tau_w > \tau_p$ (Table.1). The Whole Score measures the grouping quality of tree nodes at whole-level (Fig.15).

On the one hand, the forming of tree nodes is a necessary condition to form the entire tree, so Part Score and Whole Score are important measures of the representation. On the other hand, since the Part Score and Whole Score measures the grouping in part/whole level independently (Fg.5cd in the main text or Fig.14 ∼ Fig.15), they do not reveal the correlation between levels. For example, the emergent cell assemblies can be arbitrarily permuted or translated (together) without affecting the scores (Fg.5e in the main text or Fig.16). Obviously, such arbitrary operations are serious enough to destroy a well-defined tree structure (Fig.16).

Therefore, we provide the additional score to capture the cross-level coordination: the Nest Score.

**Nest Score** is defined as the coherence score between part-level spiking patterns and whole-level spiking patterns based on whole level assignments:

$$\text{Nest Score} = (4/3) \cdot \text{Silhouette}(VP(spk_1, spk_2; \tau_w), \text{label}_2) \tag{16}$$

where $spk_1, spk_2 \in \{0,1\}^{(N,\tau_l)}$ means the total spike trains of level 1 (part-level) and level 2 (whole level) respectively. $VP(spk_1, spk_2)$ is the distance matrix whose elements $(i,j)$ are the VP-distance between i-th spike train in the part level and j-th spike train in the whole level. $(4/3)$ is a normalization factor, introduced below.

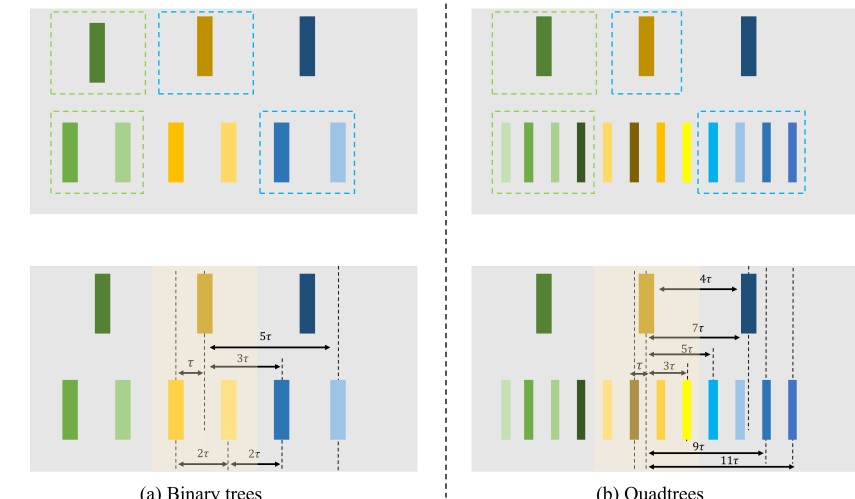

(a) Binary trees $\qquad$ (b) Quadtrees

Figure 12: Illustration of the Nest Score. Each colored bar stands for a population of synchronized neurons (cell assemblies).

Intuitively, the nestedness means that synchronized cell assemblies at the part-level are coordinated within the lifetime of whole-level cell assemblies (Fig.1 in the main text). Therefore, it could be formalized as the coherence measure between part-level spike trains and whole-level spike trains: $(b - a)/\,max(a, b)$. Here, $a$ is the average distance between part-level spike trains and whole-level spike trains that share the same whole-level assignment (Fig.12 top, green dashed box). $b$ is the mean distance between a whole-level spike train and the nearest part-level cell assemblies that the spike train is not a part of (the averaged distance between sets, Fig.12 top, blue dashed box). This formulation is similar to the Silhouette, but replaces the $VP(spk_1, spk_1), VP(spk_2, spk_2)$ to the $VP(spk_1, spk_2)$. If the Nest Score is high, it means the part-level neurons $s_1$ for the same whole-level objects ($\text{label}_2$) is correlated to whole-level neurons $s_2$. This spatial-temporal structure can be translated into the nestedness of cell assemblies across levels.

However, since part-level and whole-level should not be exactly the same, the derived $(b - a)/\,max(a, b)$ do not reach the 1 in best cases. To normalize the score into the range of $(-1, 1)$, we compute a compensatory factor — the "magic number": $3/4$. To simplify the problem, we assume that in the ideal parsing case, both part-level and whole-level spikes are synchronized perfectly and arranged uniformly along the time dimension (Fig.12 bottom). Assume the nearest time interval between part-level and whole-level spikes is $\tau$ (Fig.12 bottom), then for a "perfect" binary tree (Fig.12 a):

$$a = (\tau + \tau)/2 = \tau \tag{17}$$

$$b = (3\tau + 5\tau)/2 = 4\tau \tag{18}$$

$$\text{Nest Score} = (b - a)/\,max(a, b) = 3/4 \tag{19}$$

and for a "perfect" quadtree (Fig.12 b):

$$a = (\tau + \tau + 3\tau + 3\tau)/4 = 2\tau \tag{20}$$

$$b = (5\tau + 7\tau + 9\tau + 11\tau)/4 = 8\tau \tag{21}$$

$$\text{Nest Score} = (b - a)/\,max(a, b) = 3/4 \tag{22}$$

Interestingly, in both ideal cases, the Nest Score is $3/4$. As a result, we normalize the derived Silhouette Score by a factor $(3/4)$, which is exactly the eq.16. Here binary tree accounts for SHOPs, Ts, and double-digit MNIST while quadtree accounts for the Squares. The validity of the normalization is confirmed in Fig.5 or Fig.16, where Nest Score achieves 1 in ideal cases.

Lastly, the reason why it is valid to use whole-level assignment to ground both part-level and whole-level neurons is that we have assumed a one-to-one spatial relation between part-level SCS and whole-level SCS (topographical mapping to the physical world), and the complete object has a 'copy' at each level along the hierarchy (main text). It is also a conventional assumption of the cortex (Hinton, 2021).

### A.4.5 PERTURBATION STUDY

In order to verify the proposed scores, we conduct a perturbation study in Fig.5 in the main text. Here, we provide more details on how the perturbation is made and more discussions about the experiment (Fig.13 $\sim$ Fig.16).

Given an input image and its ground truth as in Fig.5a, we firstly 'artificially' build up the "perfectly" nested spike pattern in two oscillation periods (whole period ($2 \cdot \tau_{total}$) is 54 time steps). More specifically, all cell assemblies are synchronized almost perfectly and arranged uniformly along the time axis as in Fig.12. Part-level cell assemblies are coordinated within the lifetime of whole-level cell assemblies (Fig.5a). It is the ideal case, with 0 perturbation level in Fig.5bcde or Fig.13 $\sim$ Fig.16.

Then, for Fig.5bcd or Fig.13 $\sim$ Fig.15, we randomly and independently perturb the timing of spikes into nearby time points:

$$t_i \longrightarrow t_i \pm \Delta t, \quad \Delta t \tau \tag{23}$$

where $t_i$ is the spike timing of $i_{th}$ neuron, $i \in N$. $N$ is the number of total considered neurons (part-level or whole-level or both levels). $\tau$ is the timescale controlling the perturbation level. If $\tau = 0$, perturbation is zero. If $\tau$ equals half the length of the oscillation period ($0.5 \cdot \tau_{total}$), the perturbation will lead to pure random firings like Fig.13 $\sim$ Fig.15, bottom. As a result, we define the perturbation level in Fig.5bcd as $\tau/(0.5 \cdot \tau_{total})$, ranging from $0\%$ to $100\%$. A more detailed perturbation process is shown in Fig.13 to Fig.15. In Fig.5b or Fig.13, the perturbation is applied to both part and whole level so that all scores smoothly decrease from 1 to near 0. In Fig.5c or Fig.14, the perturbation is only applied to the part level, so that the Whole Score is not affected but both Part Score and Nest Score smoothly decrease from 1 to near 0. In Fig.5d or Fig.15, the perturbation is only applied to the whole level, so that the Part Score is not affected but both Whole Score and Nest Score smoothly decrease from 1 to near 0. In a word, in Fig.5c and Fig.5d (Fig.14 and Fig.15), we isolatedly verify the property of Part / Whole Score, which shows that they are capable of capturing the quality of node-level representation of a tree structure. In Fig.5b or Fig.13, we provide more common cases where both part and whole level degrades, which shows that three scores consistently measure the coherence of neuronal representation.

For Fig.5e or Fig.16, to isolatedly verify the role of Nest Score. We firstly build up perfect synchronized cell assemblies as in the perfect nested case (Fig.5a), but then perturb the timing of each 'cell assembly' at different levels. All spikes within the same synchronized cell assembly are perturbed with the same $\Delta t$ and different cell assemblies are perturbed by independent $\Delta t$ (Fig.5e top or Fig.16). Similarly, we define perturbation level as $\tau/0.5 \cdot (\tau_{total})$, where $\tau$ is the timescale controlling the perturbation. A more detailed perturbation process is shown in Fig.16. As shown in Fig.5e or Fig.16, the Nest Score decreases smoothly while Part / Whole Scores remain constant. Notably, the perturbation can lead to wrong hierarchical coordination: whole-level cell assemblies are synchronized with part-level cell assemblies of different assignments (incoherence). Thus, the Nest Score can decrease into values even lower than 0.

### A.4.6 THE SHIFT: FROM SYNCHRONIZATION TO POLYCHRONIZATION

In the neural system of the brain, the synchronization matters since the coincident arrival of spike trains could have a much larger effect on the target neuron. Therefore, it is the "synchronization" in the viewpoint of the reader neuron that really matters (Buzsáki, 2010). However, due to the diverse axonal delay (tens of milliseconds) of different neurons, coincidently arrived spikes are usually fired

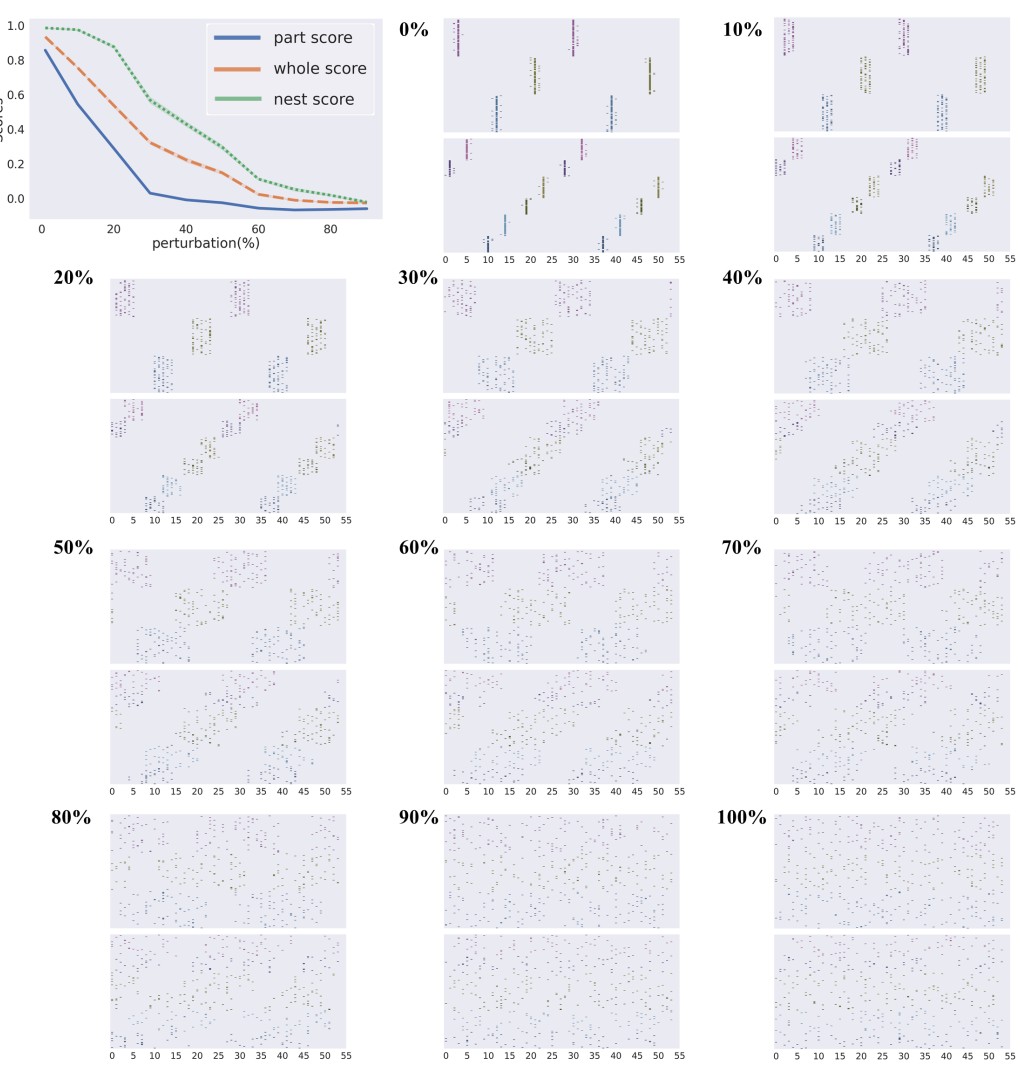

Figure 13: Perturbation study by adding increasing random noise to both part-level and whole-level spike patterns. Visualizations of the perturbed spiking pattern at different perturbation levels ($0\% \, to \, 100\%$) are shown, corresponding to the Fig.5b. Both part-level and whole-level gradually degrades into random firings.

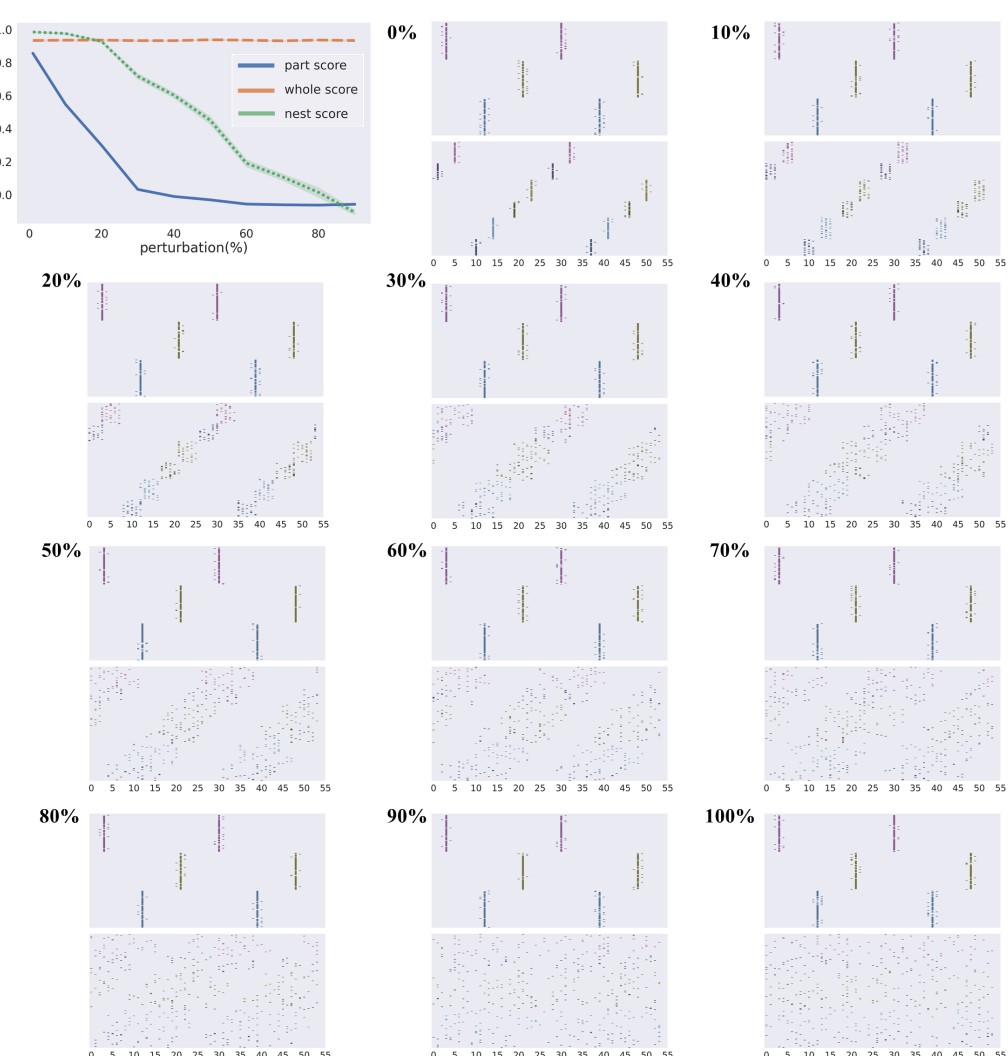

Figure 14: Perturbation study by adding increasing random noise to only part-level spike patterns. Visualizations of the perturbed spiking pattern at different perturbation levels ($0\%\ to\ 100\%$) are shown, corresponding to the Fig.5c. Part-level is degraded gradually while whole level remains unchanged.

Figure 15: Perturbation study by adding increasing random noise to only whole-level spike patterns. Visualizations of the perturbed spiking pattern at different perturbation levels ($0\%\ to\ 100\%$) are shown, corresponding to the Fig.5d. Whole-level is degraded gradually while part level remains unchanged.

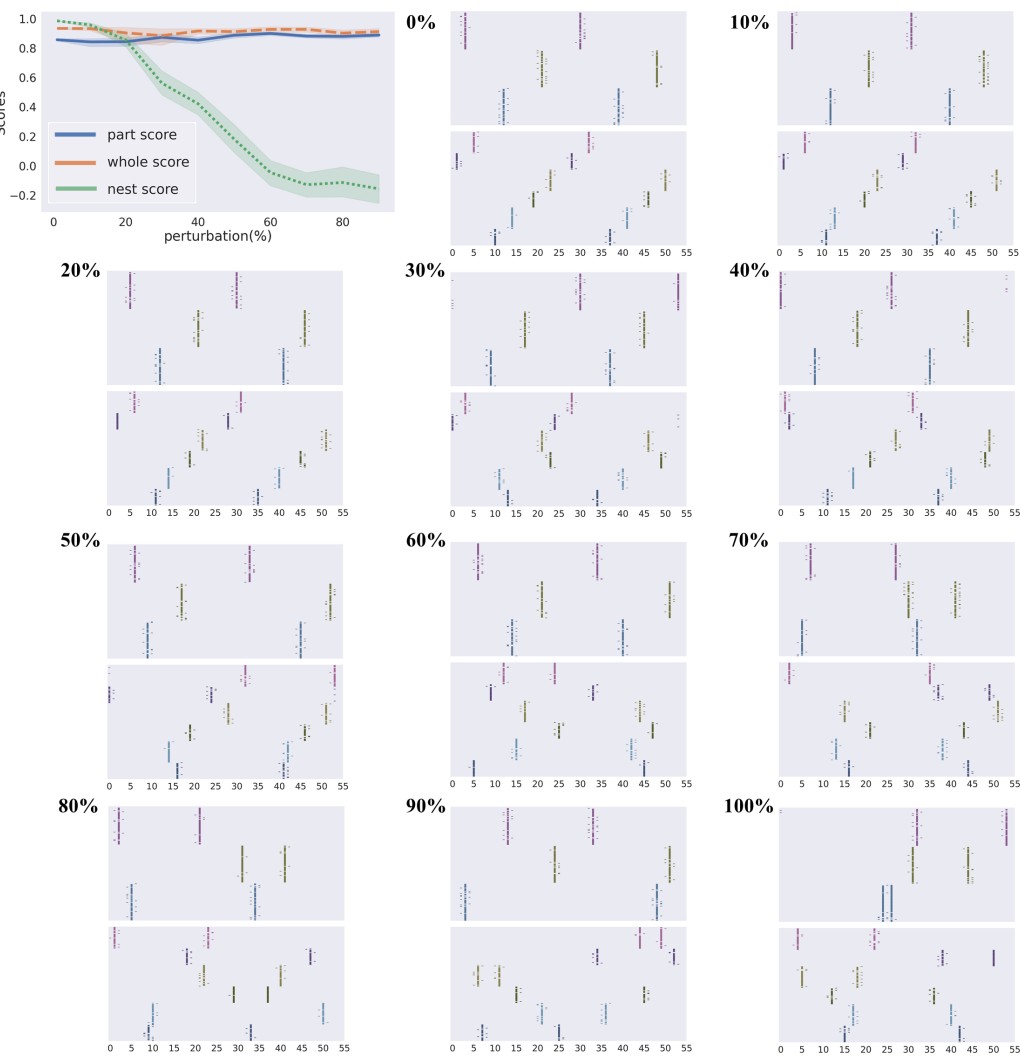

Figure 16: Perturbation study by adding increasing random noise to each assembly (both part-level and whole level). Visualizations of the perturbed spiking pattern at different perturbation levels ($0\%\ to\ 100\%$) are shown, corresponding to the Fig.5e. Relative coordination among cell assemblies is gradually changed while the synchronization of each cell assembly is unchanged.

Table 1: Parameters of the evaluation. $\tau_p$, $\tau_w$ is the time scale parameter for computing the VP-distance in each case. Shift is the fixed modified time steps for computing Nest Score and for visualization. The segment length of spike trains for (1) computing scores ($\tau_l$) and (2) for visualization is also shown.

| Dataset | Ts | Squares | SHOPs | Double MNIST |
|---|---|---|---|---|
| $\tau_p$ | 2 | 2 | 3 | 2 |
| $\tau_w$ | 6 | 7 | 6 | 4 |
| shift | 10 | 9 | 6 | 2 |
| segment length (score,$\tau_l$) | 160 | 75 | 42 | 32 |
| segment length (visualization) | 200 | 100 | 100 | 70 |

at different timings, yet with fixed temporal shifts. This phenomenon is called polychonization, or polychonized neuronal groups (PNG) (Izhikevich, 2006), which generalizes the concept of synchronization and is a more natural outcome of a real-world neural system, with potentially heterogeneous parameter settings. In other words, polychony and synchrony bear the same spirit of fixed temporal correlation, but polychonization could tolerate a fixed temporal shift among spike timings. While in an external observer's viewpoint, two things are different, in the viewpoint of the internal readout neuron, both can be the same thing.

In other words, if we shift all timing patterns with a fixed shift parameter, it is equivalent to the original pattern in the sense that the temporal shift can be compensated by the fixed axonal delay when being read out by a downstream module. Motivated by this fact, such slight fixed shifts are compensated (ignored) before computing the Nest Score[6]. In other words, in the Composer, whole-level and part-level cell assemblies are allowed to have a slight fixed temporal shift (translation slightly along the time axis). The representation is regarded as unaffected as long as the shift is a constant (Table.1).

### A.4.7 GENERALIZE TO EVALUATE OTHER MODELS

The metric proposed in this paper is also applicable to neural models that exploit similarity or coherence measures to group neural representations into part-whole hierarchies. GLOM (Hinton, 2021) is one interesting example and its implementation, Agglomerator (Garau et al., 2022) is compared as the benchmark. To measure the similarity among vectors, the Victor-Purpura metric is not needed anymore. Therefore, it is more direct to take each vector as a sample and the different dimensions of the vectors as the features in a clustering algorithm. In this way, three Silhouette-based coherence measures of spike trains could be naturally generalized to account for real-valued vectors. For example:

$$\text{Part Score} = \text{Silhouette}(D(l_p, l_p), \text{label}_1) \tag{24}$$

$$\text{Whole Score} = \text{Silhouette}(D(l_w, l_w), \text{label}_2) \tag{25}$$

$$\text{Nest Score} = \alpha \cdot Silhouette(D(l_p, l_w), \text{label}_2) \tag{26}$$

For GLOM (Hinton, 2021), $l_p$ and $l_w$ is the feature vector for each part-level column and whole-level column[7]. Besides, Euclidean metric for real-valued vector suffice for measuring the similarity: $D$.

---

[6]Let $s'_i$ denotes the shifted spiking patterns of the original spiking pattern $s_i$, then: $s'_2(t) = s_2(t + \text{shift})$ while $s'_1(t) = s_1(t)$. In other words, we slightly shift whole-level assemblies "backwards" relative to part-level spiking patterns for computing the Nest Score and for visualization. In Fig.6, we shift the $\gamma_i$ and $\Gamma$ in the same way for the same reason. The shift time step is very small and is a constant (Table.1)

[7]An interesting point is that the islands of identical vectors in GLOM are parallel to the correlated cell assemblies in the Composer, as long as we take each temporally unfolded spike train as the (binary) vector in each GLOM's column.

$label_i$ is the ground-truth parsing[8]. $\alpha$ is a normalization factor that can be determined based on the "ideal" case.

For complex-valued network(Löwe et al., 2022), which could potentially form compositional structures by similarity among the phase of complex values[9]. $l_p$ and $l_w$ is the phase vector for part-level and whole-level. Similarly, Euclidean metric suffices for measuring the similarity among phase vectors: $D$.

Taken together, the proposed metric can be generalized to evaluated related models that (1) represent part-whole structure by certain type of coherence measure and (2) being tested on synthetic datasets where ground truth is available.

---

[8]We are aware that the proposed coherence metric still requires the ground truth ($label_i$), which is a limitation to evaluate the performance on real-valued datasets. However, evaluating grouping or parsing without ground truth is a well-known big challenge for these fields and we leave this hard problem to future works.

[9]Currently, this line of work can not account for hierarchical object level since the grouping mechanism is realized as special activation function, which can not be flexibly modified to, for example, distinguish different object levels. In contrast, the time window of readout neurons in the Composer can be flexibly configured to shape different dynamics for different part-whole levels.

Table 2: Time scale constants of the Composer

| Dataset | Ts | Squares | SHOPs | Double MNIST |
|---|---|---|---|---|
| $T$ | 3000 | 3000 | 3000 | 3000 |
| $\tau_d$ | 80 | 75 | 42 | 16 |
| $\tau_{\delta 1}$ | 36 | 36 | 20 | 16 |
| $\tau_{\delta 2}$ | 35 | 35 | 20 | 15 |
| $\tau_{r1}$ | 15 | 24 | 12 | 16 |
| $\tau_{r2}$ | 14 | 24 | 12 | 15 |
| $\tau_1$ | 2 | 2 | 3 | 2 |
| $\tau_2$ | 6 | 12 | 6 | 8 |
| $\tau_D$ | 18 | 30 | 10 | 8 |
| $\tau_\Gamma$ | 15 | 16 | 8 | 8 |
| $\tau_{d'}$ | 80 | 75 | 42 | 16 |

### A.5 HOW THE COMPOSER WORKS

In the main text, we introduced the architecture of the Composer, the neuroscientific motivation of the architecture and the intuition of how the architecture generates the nested dynamics. However, while in the main text, we briefly mentioned that time scale parameters and the priors in DAE contribute to the emergence of part-whole structure, we leave out the details on parameter setting and training scheme into this Section and the following Section. Besides, the ablation study in the main text (for time scale parameter and for the DAE in Fig.9) is exemplified on one of the datasets for clarity, leaving out similar results on other datasets into this Section and the following Section.

In this section, we provide more details about practical implementations on how the model works. We first clarify the initialization of the dynamics in Section A.5.1 and Table.3. Then, we discuss the time scale parameter settings (Table.2) and ablation of time scale parameters in Section.A.5.2 and Section.A.5.4. In next Section.A.6, we provide training details of the DAE and more ablations about DAE (Section.A.6.5). Taken together, a more complete picture of how different modules contribute to the Composer is unfolded.

To have a intuitive impression of the dataflows in the Composer, a zip file containing videos visualizing the dynamics of the Composer is provided in SI (60MB).

#### A.5.1 INITIALIZATION OF THE DYNAMICS

The dynamics of Composer is gated by the top-down feedback (output from DAE or integrated spikes from higher-level SCS). However, at the initial phase, what should be the value of DAE output or higher-level feedback? One simple solution is to initialize the feedback as (uniform) random noise.

To formulate, eq.1 and eq.4 in main text are slightly extended to clarify the initialization process:

$$\rho_1(t) = x \cdot (\gamma_1 \cdot \Gamma_1 + r_1(t) \cdot \epsilon_1) \tag{27}$$

$$\rho_2 = (\lambda \cdot x + (1 - \lambda) \cdot D) \cdot (\gamma_2 + r_2(t) \cdot \epsilon_2) \tag{28}$$

where, the term $r_i(t) \cdot \epsilon_i$ is only for random initialization (See Table.3). $\epsilon_i$ is sampled from uniform distribution $U[0, 1]$ and $r_i(t)$ is the temporary amplitude of the noise, which is decayed rapidly along the simulation (decay rate $\sim 0.8$, Table.3). In other words,

$$r_i(t) = r_i \cdot (0.8)^{-t/\tau_d}, i = 1, 2 \tag{29}$$

where $r_i$ is the initial amplitude of the noise. During simulation, the noise is decayed every $\tau_d$ time steps for simplicity.

Table 3: hyper-parameters of the Composer (besides time scale constants). g (eq.2) is the (inhibitory) gating effect of relative refractory period ($\tau_\delta - \tau_r$), same for whole-level and part-level for simplicity. $\lambda$ in eq.4 describes the skip connection. $r_1, r_2$ describes the initialization (eq.29).

| Dataset | Ts | Squares | SHOPs | Double MNIST |
|---|---|---|---|---|
| $g$ | 0.5 | 0.3 | 0.3 | 0 |
| $\lambda$ | 0.3 | 0.4 | 0.4 | 0.4 |
| noise decay | 0.8 | 0.8 | 0.8 | 0.8 |
| $r_1$ | $\frac{1}{40}$ | $\frac{2}{3}$ | $\frac{1}{9}$ | $\frac{1}{8}$ |
| $r_2$ | $\frac{1}{40}$ | $\frac{2}{3}$ | $\frac{1}{9}$ | $\frac{1}{8}$ |

### A.5.2 TIME SCALES

The Composer is inspired by the neural syntax hypothesis(Buzsáki, 2010), which argues that the hierarchical organization of cell assemblies should be readout by hierarchical organization of time windows. In fact, it is likely in the Composer that the hierarchical organization of time window also encourages the emergence of hierarchical cell assemblies. Motivated by the hypothesis, spikes in SCS are all integrated within a narrow time window (implemented as a integration function $I_i$ of certain time constant $\tau_i$, $i \in \{1, 2, D, \Gamma\}$) before the downstream processing. The time scale parameters related to the model are shown in Table.2, which has appeared in eq.1 to eq.7 in the main text. $T$ is the entire simulation length. $\tau_d$ is the coupling delay of the top-down feedback inside the column, shared for both part-level and whole-level. $\tau_{\delta 1}$ is the total refractory period of part-level spiking neurons and $\tau_{\delta 2}$ is that of the whole-level spiking neurons. $\tau_{r1}$ is the absolute refractory period of part-level spiking neurons and $\tau_{r2}$ is that of the whole-level spiking neurons. $\tau_1$ is the integrative time window from part-level SCS to part-level DAE (eq.3) and $\tau_2$ is that from the whole-level SCS to whole-level DAE (eq.6). $\tau_D$ is the integrative time window from the part-level SCS to the whole-level SCS (eq.7, right). $\tau_\Gamma$ is the time window of the top-down feedback from the whole-level SCS to the part-level SCS (eq.7, left). $\tau_{d'}$ is the coupling delay of the cross-level top-down feedback from the whole-level SCS to the part-level SCS (eq.7, left). In this work, we set $\tau_{d'} = \tau_d$ for simplicity. Roughly speaking, we have:

$$\tau_d = \tau_{d'} > \tau_{\delta 1} \sim \tau_{\delta 2} > \tau_{r1} \sim \tau_{r2} > \tau_2 \sim \tau_D \sim \tau_\Gamma > \tau_1 \qquad (30)$$

Notably, part-level and whole-level columns are characterized by two timescale parameters (readout time window of SCS): $\tau_1 < \tau_2$, which softly shape the timescale (fast or slow) of the intra-column dynamics, which is inspired by the timescale hierarchy along the cortical hierarchy (Mahjoory et al., 2019) and the neural syntax hypothesis (Buzsáki, 2010).

If we take each time step as 1 millisecond in the brain, then the refractory period $\tau_\delta$ is around tens of milliseconds and the absolute refractory period $\tau_r$ is around ten milliseconds. The coupling delay is around 50 millisecond (Singer, 2021). The integrative time window matches that of the coincidence detector (several millisecond (König et al., 1996)). The frequency of oscillatory activity is around ten of milliseconds, within the Gamma band (Tallon-Baudry & Bertrand, 1999).

### A.5.3 ABLATION STUDY OF THE TIMESCALE PARAMETERS

In Fig.9b in the main text, we provide the ablation study of the timescale parameters. Here we provide more discussions (Fig.17).

Firstly, the coupling delay $\tau_d = \tau_{d'}$ is most essential for the capability of the Composer. As shown in Fig.9 and Fig.17, once removed, the parsing representation fails directly. As motivated by the neuroscientific studies on cortical dynamics (Singer, 2021), the delay coupling is also essential for generating the proper dynamical states in the Composer. How to understand this? As shown in Fig.2c,h, we need a chain of transient attractors to form a sequence oscillatory cell assemblies. Therefore, the positive feedback (the top-down feedback from DAE or from higher-level SCS) is desirable to wait for the modulated SCS neurons to recover from the refractory period. So that the cell assemblies sequence becomes a stable attractive states of the column dynamics. This idea is proved in (Zheng et al., 2022) for a single column. In general, the feedback delay provide a "time

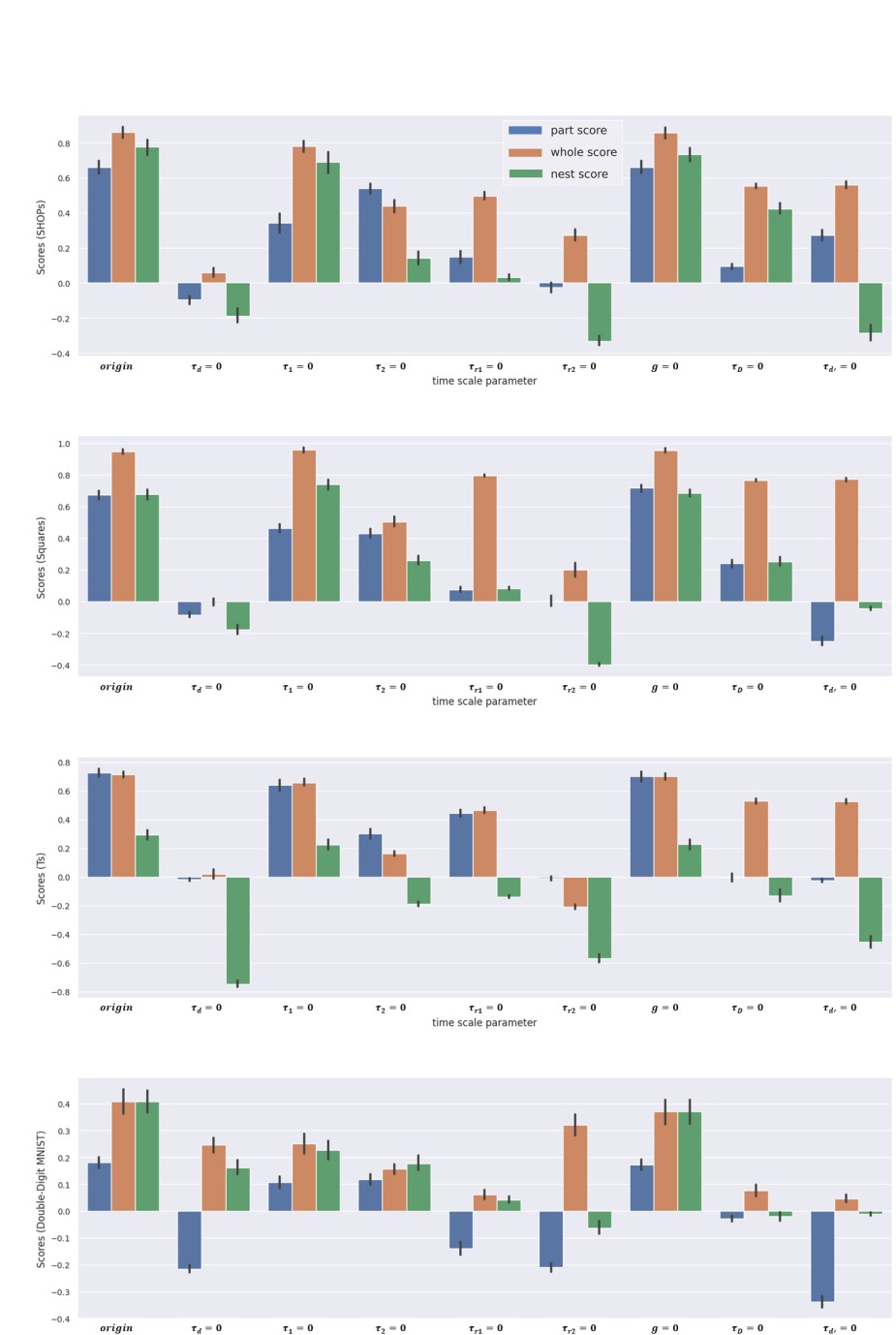

Figure 17: Ablation study on all datasets, providing additional results for Fig.9b in main text.

window" for different metastable states to compete to emerge and disappear. It is notable that the effect of delay is soft and the Composer is robust to the shift of delay within a reasonable range (See Sensitivity Test).

Secondly, the refractory period ($\tau_{r1}, \tau_{r2}$) has a secondary effect on the Composer. As shown in Fig.2c,h, the refractory period contributes to destroying the attractor dynamics (Fig.2b) to be transient (Fig.2c). If the refractory period is removed, the nested oscillatory states are not likely to be stable states any more. Therefore, all scores degrade for different degree.

Thirdly, the integration timescales ($\tau_1, \tau_2, \tau_D$) also matters significantly. But the effect is relatively soft. As argued in (Buzsáki, 2010), the time window of readout neurons determines what they can "see", which in turn determines what downstream modules (DAE) predict as top-down feedback. If the time window is very narrow, it is less plausible for the readout to "see" larger spatial-temporal scale coherence (or assemblies), which encodes higher-level objects. As a result, it is less likely to provide top-down predictive feedback for high-level objects. Instead, low-level objects are more likely to be captured by readout and top-down feedback. Taken together, the integration time window shapes the spatial-temporal scale of the cell assembly sequences and the time scale of the network dynamics (frequency of the oscillation).

Fourthly, the removal of the relative refractory period slightly degrades the nestedness of the Composer. The explanation is that: Representing the part-whole hierarchy is a combinatorial problem in nature, which needs to be iteratively "searched". For example, when the object number increases as in the Ts dataset, the number of assemblies increases. Given larger number of assemblies as tree nodes, the possible configurations of the parse tree (correct or incorrect) gets exponentially larger. Therefore, the searching becomes harder because the searching space enlarges exponentially. However, while hard refractory period forces the system to switch among different states (spike fires at wrong timings), the 'hardness' could prevent efficient self-correcting once the system gets into a wrong state (because the hard refractory period constraints the available next firing timing). Thus, introducing a relative refractory period can help the system jump out of the local minimum, once it 'finds' much better states. It is likely that for this reason, enforcing $g = 0$ in Fig.9 in the main text slightly degrades the Nest Score.

### A.5.4 PARAMETER SENSITIVITY TEST

We provide a sensitivity test of parameters on SHOPs dataset in Fig.18. The score is relatively robust with the perturbation of the parameters, as long as the parameter is within the suitable range described by eq.30. Therefore, although including so many time scale hyper-parameters may be undesirable for machine learning models, these hyper-parameters only "softly" influence the performance of the Composer. Actually, we do not perform precise parameter tuning and these time scale parameters are "minimally" required for a brain-inspired model that takes time-scale interactions into account. In other words, we somehow need a time window to readout the spike in SCS, and that is all the Composer requires. Since the time scale parameters have a relatively wide effective range, the Composer is potentially generalizable to broadly bridge neuroscience and machine learning to study human-like vision in the future.

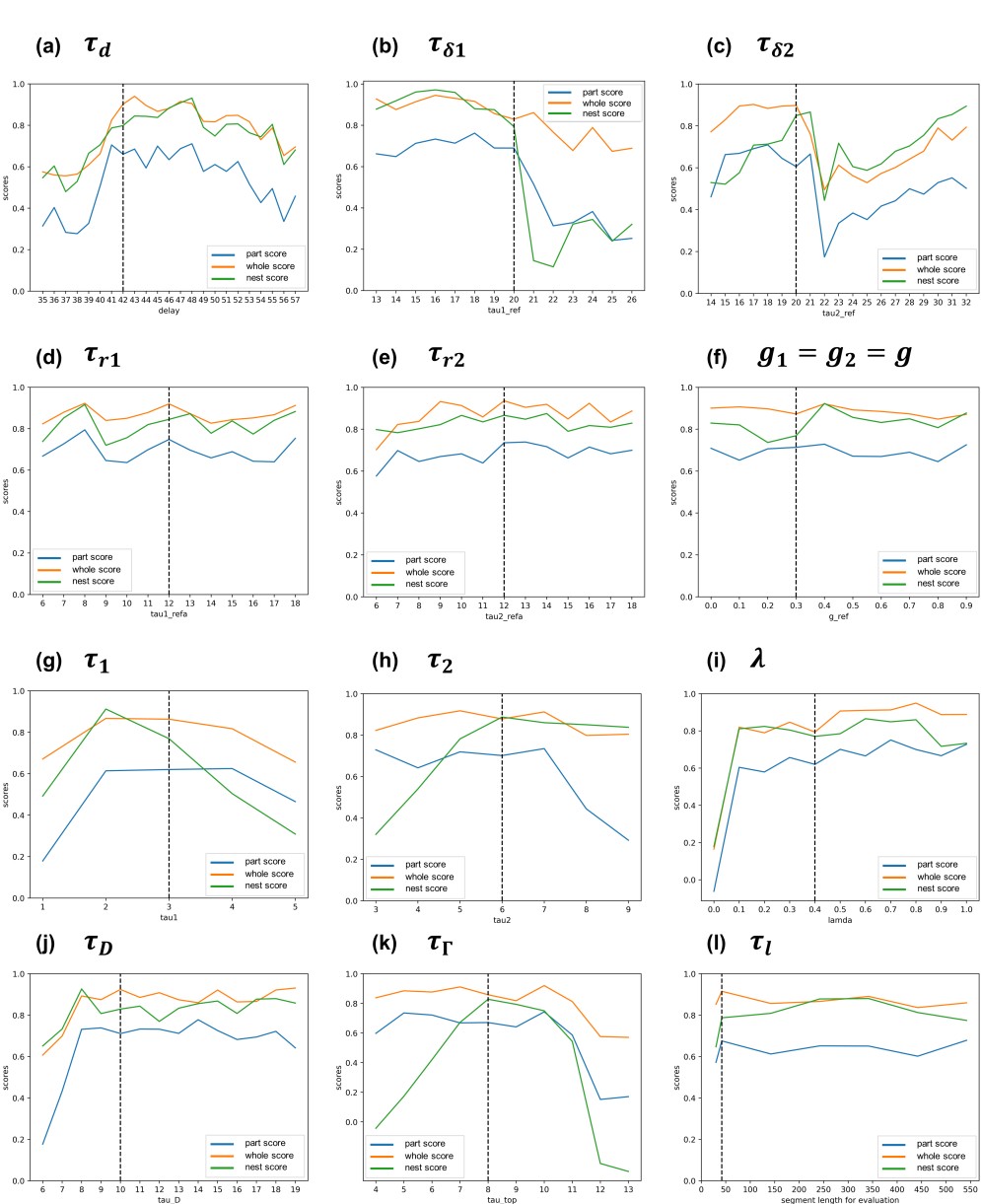

Figure 18: The sensitivity test of time scale parameters on SHOPs dataset. Black-dashed line indicates the value used for the Composer in the main text (Table.2). The value of parameters are perturbed to show how parsing degrades w.r.t parameter change. (a) The delay parameter of DAE feedback, same for part / whole level; (b) entire refractory period for part-level; (c) entire refractory period for whole level; (d)absolute refractory for part level; (e) absolute refractory for whole level; (f) the inhibitory effect of the relative refractory function; (g) the integration time window for part-level spiking neurons; (h) the integration window for whole-level spiking neurons; (i) the factor of the partial influence from skip connection; (j) the integration time window from part-level to whole level; (k) the integration time window from whole-level to part level; (l) the length of (spike train) segment used for evaluating the parsing quality (Table.1).

## A.6 TRAINING DETAILS

### A.6.1 RESOURCES

Our experiments have been performed on ubuntu 16.04.12 with devices: CPU (Intel(R) Xeon(R) CPU E5-2640 v4 @ 2.4GHz) and 4×GeForce RTX 2080 Ti. The python version is 3.6.3.

### A.6.2 NETWORK ARCHITECTURE AND TRAINING HYPERPARAMETERS

The details of training neural networks are shown in Table.4. All networks are trained with stochastic gradient descent (SGD).

Table 4: Details of training DAE

| **Dataset** | **encoder** | **decoder** | [c]**learning rate** | **noise** | [c]**minibatch size** | [c]**epoch num** |
|---|---|---|---|---|---|---|
| [l]Ts (part) | [l]FC(1600, 1000) Sigmoid() | [l]FC(1000, 1600) Sigmoid() | 1e-3 | 0.5 | 16 | 200 |
| [l]Ts (whole) | [l]FC(1600, 1000) Sigmoid() | [l]FC(1000, 1600) Sigmoid() | 1e-3 | 0.5 | 16 | 200 |
| [l]Squares (part) | [l]FC(3600, 400) Sigmoid() | [l]FC(400, 3600) Sigmoid() | 1e-3 | 0.8 | 16 | 200 |
| [l]Squares (whole) | [l]FC(3600, 400) Sigmoid() | [l]FC(400, 3600) Sigmoid() | 1e-3 | 0.6 | 16 | 200 |
| [l]SHOPs (part) | [l]FC(3600, 400) Sigmoid() | [l]FC(400, 3600) Sigmoid() | 1e-3 | 0.7 | 16 | 200 |
| [l]SHOPs (whole) | [l]FC(3600, 400) Sigmoid() | [l]FC(400, 3600) Sigmoid() | 1e-3 | 0.7 | 16 | 200 |
| [l]Double-MNIST (part) | [l]FC(6400, 2000) Sigmoid() | [l]FC(2000, 6400) Sigmoid() | 1e-3 | 0.5 | 16 | 200 |
| [l]Double-MNIST (whole) | [l]FC(6400, 2000) Sigmoid() | [l]FC(2000, 6400) Sigmoid() | 1e-3 | 0.5 | 16 | 200 |

### A.6.3 DATASET FOR TRAINING DAE

The details of training dataset are shown in Table.5. Examples of the training data are visualized in Fig.19. The setting of DAE training dataset follows the convention in Zheng et al. (2022).

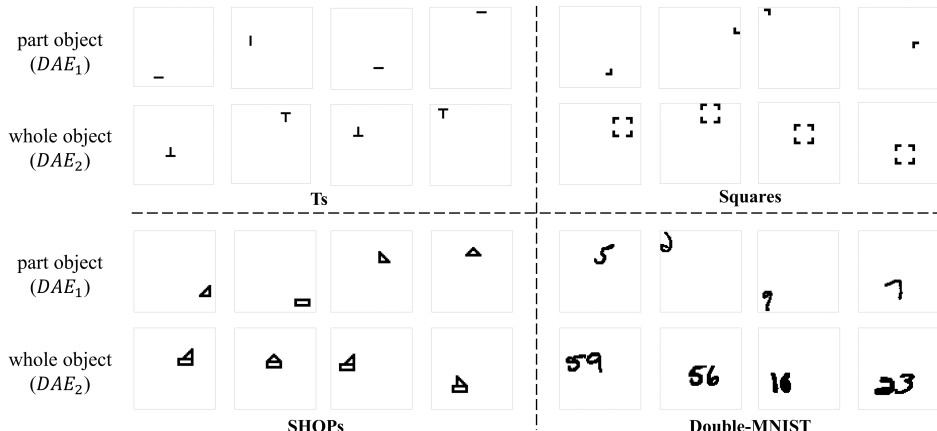

Figure 19: Examples of the training data to train part/whole level DAE.

Table 5: Training dataset details

| Dataset number | Training size | Input dimension | [c]Object |
|---|---|---|---|
| [l]Ts (part) | 60000 | $40 \times 40$ | 1 |
| [l]Ts (whole) | 60000 | $40 \times 40$ | 1 |
| [l]Squares (part) | 60000 | $60 \times 60$ | 1 |
| [l]Squares (whole) | 60000 | $60 \times 60$ | 1 |
| [l]SHOPs (part) | 20000 | $60 \times 60$ | 1 |
| [l]SHOPs (whole) | 20000 | $60 \times 60$ | 1 |
| [l]Double-MNIST (part) | 60000 | $80 \times 80$ | 1 |
| [l]Double-MNIST (whole) | 60000 | $80 \times 80$ | 1 |

### A.6.4 Loss function

The DAE (either part or whole) are trained to minimize the MSE loss between the output of DAE and original image:

$$loss(x) = (x - DAE_i(\tilde{x}))^2, \quad i = 1, 2 \tag{31}$$

where $x$ is the original single-object image in Fig.19. $\tilde{x}$ is the corrupted version of $x$. Notably, the training of DAE has an unsupervised form and does not provide any explicit information on how to bind distributed features into the tree nodes or to form the parsing tree (coordination of multi-level tree nodes)[10]. These compositional structure all emerged during the simulation dynamics. All the training does is to provide the minimal prior about what (on average) the object (part/whole) looks like, so that the model could make sense of the multi-object scene at all. It is plausible that such priors of "object prototype" also exist in the brain to help parse the scene. For example, before parsing the face (Fig.1a in the main text), a person should have a prior about the eye, nose, and mouth. Such prior of content should also influence the outcome of the parsing structure.

In this paper, we treated these senses of the object (part or whole) as prior knowledge and explored how the parsing structure emerges on the condition of the prior knowledge. While some of the prior may be hard-wired in the brain through evolution, others may also be learned during development. The learning aspect of these priors is not discussed in this preliminary model, but recent work by Zheng et al. (2023) provides insight into how it could potentially be achieved: the general architecture

---

[10]In other words, the pre-training is only for representation content instead of representation structure.

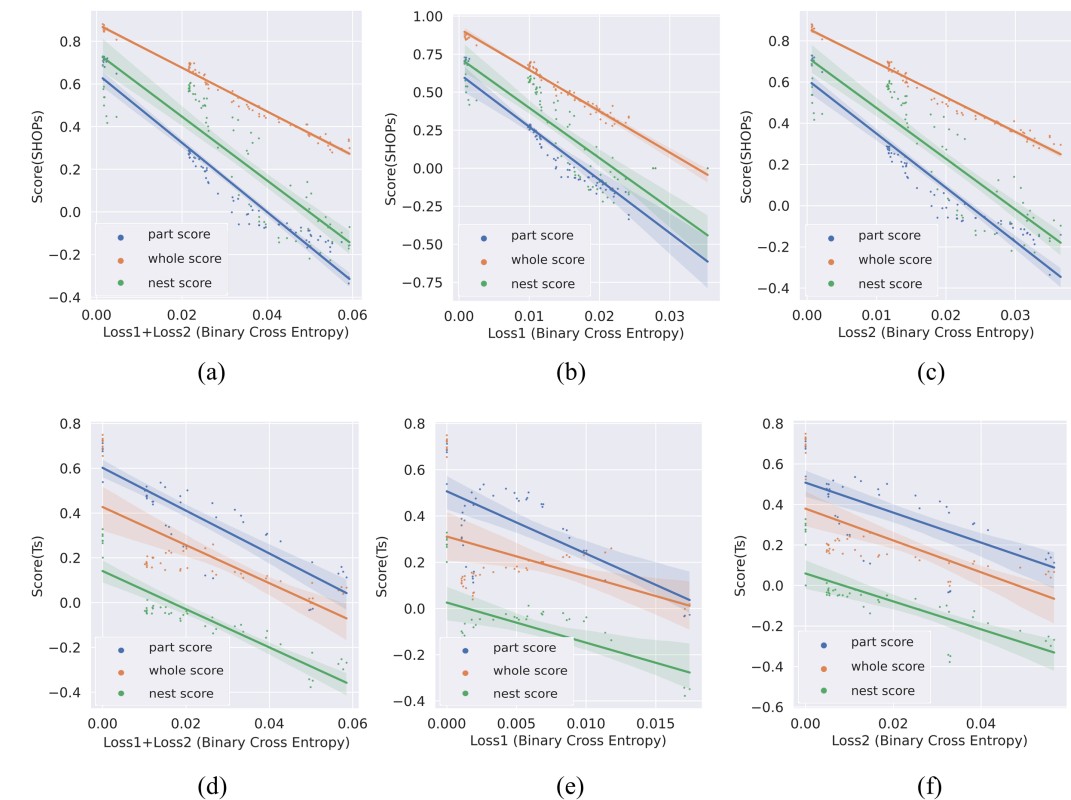

Figure 20: Loss vs Score (additional results). (a)(b)(c) results on SHOPs dataset; (d)(e)(f) results on Ts dataset. (a)(d) Relations between scores and total loss of part / whole level DAE ($loss_1 + loss_2$); (b)(e) Relations between scores and total loss of part-level DAE ($loss_1$); (c)(f) Relations between scores and loss of whole-level DAE ($loss_2$); All results are consistent.

in this work, including the explicit separation of columns levels, the delay-coupled-bottom-up-top-down column architecture and hierarchical organization of time-scale constants, could be treated as the **"inductive bias"** of the end-to-end training. On the one hand, these biological constraints guarantee the nested states to be stable states of the network dynamics, given that the DAE is selective for different level objects (part/whole). On the other hand, the limited time window may also bias the DAE to "learn" to predict objects consistent with the scales of the time-window, because the time window softly determines what DAE can see "clearly" (Zheng et al., 2023). Taken together, instead of training DAE separately, we could treat the entire model as a recurrent spiking neural network and train the model by back-propagation through time (BPTT) (Wu et al., 2018). The loss function is only need to be modified minimally: the MSE loss between the entire scene (containing multiple whole-level objects) and the **averaged** top-down feedback (eg.$\gamma_1(t)$, or $\Gamma(t)$). Due to the constraint of hierarchical temporal structure of the model (inductive bias), it is more efficient to learn a part-whole hierarchy representation to predict the whole image. Then, the single-object prior is possible to be learned in a fully unsupervised manner. We leave it as a promising future work: how neural syntax can be learned from predicting the external world, consistent with neuroscientific hypothesis (Buzsáki et al., 2014).

### A.6.5 ABLATION STUDY OF THE DAE

In Fig.9a, we conduct ablation studies to the DAE module in order to find out the relation between the parsing scores and the total training loss of part-level and whole-level DAEs. We randomly selected 100 learning rates from $(10^{-3}, 1)$ and for each selected learning rate we trained one part-level DAE and one whole-level DAE. So there are 100 part-level DAEs and 100 whole-level DAEs for each dataset (100 DAE pairs). Then we evaluate the parsing score of the Composer equipped with each of

the 100 DAE pair. Specifically, the x-axis in Fig.9a is the summed loss of both $DAE_1$ and $DAE_2$ that are trained with the same randomly selected learning rate. During the evaluation, exceptional data points where loss gets unreasonably large due to sick learning rate are removed.

Lastly, we find that the overall relationship between the DAEs and Scores is consistent across datasets and not closely dependent on which DAE is used for comparison. For example, we show more results in Fig.20. The relationship is consistent across different cases. For this reason, we show one of the results (Fig.20)a in the main text without losing generality.

Taken together, the positive relationship between lower denoising loss and higher scores indicates that there are direct interplays between the DAE and the parsing ability of the Composer. It make sense since the quality of DAE affects the quality of associatvive memory in Fig.2b and in turn affects the quality of coherence structure in Fig.2c,h,i.

## A.7 BIO-PLAUSIBILITY

In this section, we list and provide detailed discussion about the biological correlates of the design of the Composer in hope of inspiring future innovations.

1. **Delay-coupled oscillatory neural network**: In a recent work, Singer (2021) describes the cerebral cortex as a delay-coupled recurrent oscillator network, which is very different from the architecture in the deep learning field. In the Composer, such architecture is captured, integrated within the deep learning framework, and acting as an essential ingredient of the mechanism (Section.A.5.3). The coupling delay provide a time window for alternative cell assemblies compete to emerge and disappear in order . In other words, the coupling delay makes the system non-Markovian and of infinite dimension (approximately, see Izhikevich (2006)), so that the coding scheme and the capability of associative memory is much enlarged[11]

2. **Feed-forward and feed-back pathway along the cortical hierarchy**: In general, the cortex is organized into similar columns (Douglas & Martin, 2004), which is composed of six layers from layer I to layer VI (Fig.2d,g). Cortex are spatially organized corresponding to the spatial structure of the external physical world and hierarchically organized into levels. These basic features are captured in our model and act as essential elements for representation hypothesis: the representation of the part-whole hierarchy depends on such spatial and hierarchical organization. Notably, in a neurology paper (Markov et al., 2013), the author also specifies the organization of the feedforward and feedback hierarchy. In detail, there are recognizable feedforward and feedback pathways between layer II/III of higher and lower level columns. This corresponds to the cross-level interaction between the part-level SCS and whole-level SCS in our model. Lastly, Markov et al. (2013) also shows that a long-distance feedforward path from lower level to high level exists in layer IIIb. These are realized as the skip connections from external driving input ($x$) to the whole level SCS layer in the Composer (eq.4). More generally, the feedback from higher levels contains signals originating from both layer II/III and layer V/VI (corresponding to the latent space in the Composer). It is left to future work to study the "cross-level" interaction between pixel-level SCS and the latent space of DAEs.

3. **Time scale hierarchy**: Along the cortical hierarchy, there is a gradient of timescale hierarchy (Mahjoory et al., 2019)—'*We found that the dominant peak frequency in a brain area decreases significantly, gradually and robustly along the posterior-anterior axis, following the global cortical hierarchy from early sensory to higher order areas*'. Such time scale hierarchy is exploited in our model as the basis for representing hierarchical inclusion relationships among part-level cell assemblies and whole-level cell assemblies. However, since the frequency spectra are not unlimited, the capability of the part-whole representation may be limited by the range of the total frequency bands. Here, we treat such limitation as a shared weakness of our model and the brain, since the temporal resolution of cross-frequency coupling has shown to be a constraint for the capability of working memory of humans ($7 \pm 2$) (Nicola & Clopath, 2019). Indeed, Hinton (2021) argued that human also has a limited range of the hierarchy depth to represent instantaneously ($\sim 5$ levels). At least three frequency bands could be explored in the future: gamma band, alpha band, and theta band.

4. **Topographical mapping**: As mentioned above, the spatial organization of the cortical column has a topographical correspondence to the physical world, called the topographical mapping (Eickhoff et al., 2017). Such location-wise representation is exploited in Hinton (2021) as a core basis to represent the part-whole hierarchy and is similarly essential for our model. The topographical relationship enables the representation of objects as a set of (grouped) spatial regions with correlated activation patterns (grouped pixels with similar spike trains in the Composer or grouped columns with identical vectors, so called "identical islands of vectors" in GLOM). Besides, the location-wise representation also helps to clarify the inclusion relationship between whole and part across the hierarchy, both in our model and in GLOM. For example, the spatial region for part objects should also be spatially covered by their whole objects. However, spatial organization itself is not sufficient to represent part-whole hierarchy, because multiple co-activated features within the same level

---

[11]Each memory is realized as a trajectory instead of a single fixed point. The trajectory is the combination of transient fixed points so that the attractive states "in-principle" expand combinatorially or exponentially.

can lead to ambiguity (the binding problem see Malsburg (1994)). As a result, we need similarity measure or coherence measure to specify "which is which". This insight motivates the representation hypothesis of both spatial hierarchy in terms of topographical mapping and temporal hierarchy in terms of neuronal coherence.

5. **Abstract away the cortical BU/TD processing as autoencoder**: Predictive coding (Rao & Ballard, 1999) was first proposed by Dana H. Ballard and Rajesh P. N. Rao to explain the extra-classical receptive-field effects in primary visual cortex. Then, the predictive coding theory was mapped to the canonical circuit of cortical circuit (Bastos et al., 2012) and served as a unified theory of brain function (Friston, 2010). In the predictive coding model, the bottom-up and top-down feedback attention (BU/TD) is formalized as the autoencoder architecture, and the reconstruction error should be minimized to achieve minimal "prediction error" or "surprise". Such architecture is exploited in our model to realize the inner-column bottom-up / top-down pathways (BU/TD) and reconstruction error is minimized as the objective function of training. Interestingly, such predictive feedback is also related to the temporal coherence both in the cortex (Engel et al., 2001) and in the Composer. Interestingly, "abstracting away certain part of a dynamical system in neuroscience as a learnable neural network" has the advantage of building-up more flexible dynamical models by training on large datasets. Therefore, it is plausible to generate new hypothesis beyond the traditional ones (Eckstein et al., 2023; Peterson et al., 2021).

6. **Sparse code and dense code**: The dual coding scheme in the cortical circuits has been recognized when representing features: ultra-sparse coding in the superficial layer (layer II/III) and dense coding in deeper layer (layer V/VI) (Tang et al., 2018; Wang, 2018). While the latter encodes the statistical aspects of features, the former might additionally encodes the relationships. In this work, the dual coding scheme is realized as the sparse spike code in SCS and real-valued dense vector code in the DAE's latent space. The synchrony in the SCS additionally encodes the relationship among objects.

7. **Relative refractory period**: Strictly speaking, the absolute refractory period (ARP) refers to the phase immediately after a spike initialization ($\sim 2$ ms). The later phase where a spike is harder to be triggered (though not impossible) is referred to as the relative refractory period (RRP) (Dayan & Abbott, 2001). If we take 1-time step as 1 millisecond in real-world time, then the absolute refractory period is around 10 milliseconds (Table.2) in the Composer, which is much longer than the strict absolute refractory period. Therefore, the picture should be clarified as follows: the excitability of spiking neurons after a spike increases gradually, in the form of $1 - e^{-t/\tau}$. At the beginning phase, the excitability is low enough to prevent the neuron from firing a second spike given the conventional stimulus strength, but since the excitability increases rapidly during this phase, the relative period length is small compared to the whole refractory period. This beginning phase where the excitability is low enough compared to the stimulus strength but of fast increasing rate is treated as the absolute refractory period. In contrast, during the rest of the period, the excitability has recovered to the extent where neurons might generate a second spike but with a much lower probability. Since the recovery is much slower during the second phase, the temporal range is much longer than that of the first phase. This slow recovery phase is modeled as the relative refractory period in this work. The total refractory period can expand from tens of milliseconds to much longer, depending on the channel type on the axon of the neuron (Gerstner et al., 2014). On the other hand, it is also conventional in numerical modeling that the absolute refractory period is modeled no less than 5 ms. In sum, the time scale of refractoriness fits the conventional setting of biological systems.

8. **Dentritic computation of pyramidal cell**: The driving signal and modulatory signal are distinguished in the cortical circuit (Lee & Sherman, 2010), where the driving signal acts on the proximal site of dendrites (near to the soma) and the modulatory signal acts on the distal sites (far from soma) (Spruston, 2008). The two types of inputs interact in a non-linear way. Such non-linear interaction between driving input and modulatory input is captured as the multiplication between the bottom-up integration and top-down modulation, realized as the pyramidal cells in the SCS (Fig.3ab in the main text). Such a gating effect inside the column is essential for the emergence of cell assemblies and the gating effect across levels is essential for the coordination of cell assemblies into nested temporal structure, so called nested neuronal coherence.

9. **Coincidence detector**: Abeles (1982) argued that cortical neurons in superficial layers are coincidence detectors, which detect sparse synchronous events within a narrow time window. In our model, the time constant of the integrative time window is small ($\tau_1 \sim 2\text{ms}, \tau_2 \sim 5\text{ms}$). As a result, the inner-level bottom-up integration of spiking activity in the superficial layer (pixel-wise SCS) is modeled as coincidence detectors. Such a narrow time window enables two things: (1) stochastic spikes fired at extremely adjacent time steps should be detected as a single event; (2) the temporal resolution of the synchronous event is kept within a small time-scale ($\sim \tau_1, \tau_2$). Both are important to form a high-quality parse tree. Interestingly, a similar concept has also been developed in Hinton (2021), named as 'coincidence filtering'.

10. **Meta-stability of cortical network**: *"...Single-trial analyses of ensemble activity in alert animals demonstrate that cortical circuit dynamics evolve through temporal sequences of metastable states. Metastability has been studied for its potential role in sensory coding, memory, and decision-making. Yet, very little is known about the network mechanisms responsible for its genesis..."* (Mazzucato et al., 2015). In this work, we build such a system of metastable states (Fig.2) by integrating the spiking neural network (SNN) and artificial neural network (DAE as ANN) and further demonstrates its computational role in vision.

11. **Neuronal assembly as code words**: *"A widely discussed hypothesis in neuroscience is that transiently active ensembles of neurons, known as "cell assemblies," underlie numerous operations of the brain, from encoding memories to reasoning. However, the mechanisms responsible for the formation and disbanding of cell assemblies and the temporal evolution of cell assembly sequences are not well understood...I suggest that the hierarchical organization of cell assemblies may be regarded as a neural syntax..."*(Buzsáki, 2010). Besides, assemblies are shown to be able to realize arbitrary computation function (Papadimitriou et al., 2019). By combining machine learning models and neuroscientific constraints, we show how cell assembly can be transiently formed and disbanded, and be organized into a sequence at each level, and be hierarchically organized into spatial-temporal nested structure to express the neural syntax. More generally, various features, even of a continuous nature are represented as neuronal assemblies in the brain (population binary code), this coding scheme provides the basis to enable the Composer to deal with continuous features (RGB color) in the future (Stockman, 2019). The reservoir of neuronal assemblies could be more efficiently realized in neuromorphic devices (Pei et al., 2019) to account for larger range of features. From the viewpoint of a recent paper on assembly formation (Miehl et al., 2022), our model generate the assemblies by DAE-induced symmetry-breaking, which is one of the mechanism for assembly-generation.

12. **Temporal binding theory and feature integration theory**: Temporal binding theory (Engel & Singer, 2001; Malsburg, 1994) and feature integration theory (Wolfe, 2020) are two mainstream theories to solve the binding problem: how distributed information is bound together to form the whole. The former is based on time coding and neuronal synchrony while the latter is based on top-down attention searching on a spatial map. The temporal synchrony, temporal coding, top-down attention, and spatial map are all captured in this model. Thus it is promising to explore whether it could serve as a canonical model to unify the two theories.

13. **The role of sequential / spatial attention**: The binding of distributed features or the segmentation of objects are also argued to be related to spatially-shift attentions (Roelfsema, 2023). In the Composer, such spatial / sequential attention co-emerges during the simulation (Fig.6e). Beyond theories that only highlight the role of attention, we further treat sequential attention and neuronal coherence as a "symbiotic couple", and can be unified in a bigger picture of associative memory: both attention and neuronal coherence are partial description of certain-type associative memory (Fig.2). This bigger picture is consistent with recent findings (Ramsauer et al., 2020), which suggests that associative memory and attention is actually the same thing, but viewed from different angles.

14. **Temporal-spatial theory of consciousness**: *"We postulate four different neuronal mechanisms accounting for the different dimensions of consciousness: (i) "temporospatial nestedness" of the spontaneous activity accounts for the level/state of consciousness as the neural predisposition of consciousness (NPC); (ii) "temporospatial alignment" of the pre-stimulus activity accounts for the content/form of consciousness as the neural prerequisite of con-*

*sciousness (preNCC); (iii) "temporo-spatial expansion" of early stimulus-induced activity accounts for phenomenal consciousness as neural correlates of consciousness (NCC); (iv) "temporo-spatial globalization" of late stimulus-induced activity accounts for the cognitive features of consciousness as the neural consequence of consciousness (NCCcon)."* (Northoff & Huang, 2017). In this work, the temporospatial nestedness emerges and indicates the perceptual awareness (eg. recognizing the part-whole relationship), and the temporospatial alignment clarifies the content/form of the scene. The underlining assumption is that: the perceptual awareness emerges only if a part-whole relationship (of a visual scene or of a cognitive concept or of a plan or of a solution of certain problems) is well-organized in the internal representation of the brain (neural syntax). The quality of the representation reflect the "level of awareness" (eg. clear sight or dreamy sight). This assumption might be verified or falsified in future works.

15. **Gamma oscillation and perceptual awareness**: "*...One theory suggests that rhythmic synchronization of neural discharges in the gamma band (around 40 Hz) may provide the necessary spatial and temporal links that bind together the processing in different brain areas to build a coherent percept. In this article we propose that this mechanism could also be used more generally for the construction of object representations that are driven by sensory input or internal, top-down processes...*" (Tallon-Baudry & Bertrand, 1999). In this work, the spiking activity in the SCS approximately oscillates at the gamma band (tens of milliseconds if each time-step is regarded as 1 millisecond.) The gamma oscillation dynamically groups neurons into object representations (the representation hypothesis in the main text).

16. **Preconfigured brain**: In a recent "inside-out" conceptual framework of the brain, as Gyorgy Buzaki put it—"*...This is the organization I call the preformed or preconfigured brain: a preexisting dictionary of nonsense words combined with internally generated syntactical rules. The neuronal syntax with its hierarchically organized rhythms determines the lengths of neuronal messages and shapes their combinations. Thus, brain syntax preexists prior to meaningful content..."Preconfigured" usually means experience-independent. The backbone of brain connectivity and its emerging dynamics are genetically defined. In a broader sense, the term "preconfigured" or "preexisting" is also often used to refer to a brain with an existing knowledge base, ....In the preconfigured brain model, learning is a matching process, in which preexisting neuronal patterns, initially nonsensical to the organism, acquire meaning with the help of experience...*" (Buzsáki, 2019). Thus, the well-trained DAE in this paper could be treated as an essential preconfigured structure due to genetic codes or the life-long calibration of the sensory-action loop, which captures a range of object prototypes. Plasticity may only provide a secondary role to increase the precision of the 'good-enough' model (Buzsáki, 2019). Overall, the Composer is highly motivated by the Buzaki's insights.

17. **Plasticity**: One of the designs that may depart from biology is that the connection weights are trained based on a gradient-based method instead of a correlation-based method, like Hebbian rule or spike timing plasticity (Gerstner et al., 2014). However, this could be explained from two points of view. First, as argued above, the well-trained DAE could be regarded as the preconfigured structure which is gradually searched from evolution (amount to stochastic gradient-based search). Second, since the DAE structure in this model is relatively simple, the training objective (minimizing reconstruction error, the difference between input and feedback) could be interpreted as increasing the correlation between sensory neurons and modulatory neurons, so that the gradient-based training equals correlation-based plasticity. Indeed, Melchior & Wiskott (2019) shows that gradient-based learning and Hebbian plasticity can be unified in case of a simple autoencoder. Further, we could imagine that there is a two-stage learning algorithm, like the wake-sleep cycle: during the day, the system infers entities based on learned weight, during the night, the learned objects replay and the system efficiently updates the weight by association, which corresponds to the training phase of the DAE. Similar treatment has also been discussed in Hinton (2021).

18. **Inner-layer recurrent connection**: Another design feature that may depart from the biological brain is that the spike coding space (SCS) itself is not recurrent in our model. However, this could also be explained from at least two points of view. First, the feedforward and recurrent connection usually have different functional roles in the cortical circuit,

and have different levels of domination. For example, layer IV in the visual cortex are mainly feedforward and the recurrent effect are relatively weak. As a result, the inner-layer recurrence of SCS are treated as secondary compared to the recurrence of inner-column top-down feedback or inter-level top-down feedback. So that it is temporally ignored for simplicity. Further, the localized inner-level recurrence may play a secondary role (different from that of top-down feedback) to speed up the convergence by forming a grid frame (by spatially-organized connection) to encode the prior of the proximity property of objects (Gestalt principles(Wagemans et al., 2012)). Secondly, the entire two-layer column could be recognized as a single layer, with DAE parameterizing the recurrent connection weight among spiking neurons, similar to Dmitry Krotov's idea in Kozachkov et al. (2023). And the general mechanism still works. In other words, there is no restriction to view the two-level system as a column or a layer. In either case, the models maintain their bio-plausibility.

19. **Polychronization** refers to the generalization of absolute synchronization into structured asynchrony. As argued in Izhikevich (2006), due to the heterogeneity and conduction delay of the neural system, polychrony is more plausible than absolute synchrony. While the externally observed spike firing time is asynchronous, the arriving time of asynchronous spikes to downstream readout neurons is (internally) synchronous. In other words, the shift in spiking time is balanced out by the shift in conduction delay. According to the Buzsaki's idea (Buzsáki, 2010), the more rigorous definition of cell assemblies should be based on internal observation (downstream readout neurons) instead of external observation (human observer). Therefore, polychronous representation is in essence also synchronous representation. In Composer, due to the cross-level coupling-delay the part-level cell assemblies and whole-level cell assemblies seems to have a fixed temporal shift (Table.1). However, from a readout neuron perspective (eg. SCS spiking neurons), the spike arrival of whole level feedback ($\gamma$) is just in time to enslave the spike firing of part-level SCS neurons. For this reason, we ignored the slight temporal shift when computing the score (Section.A.4.6). Besides, to make the visualization more intuitive, we compensate the temporal shift for visualization in Fig.6, Fig.7 in the main text: synchronization is more intuitive to capture the nestedness than polychronization, although they all indicates the coherence state.

Table 6: Scores of the Composer

| Dataset | Ts | Squares | SHOPs | Double MNIST |
|---|---|---|---|---|
| Part Score | $0.73 \pm 0.005$ | $0.67 \pm 0.007$ | $0.67 \pm 0.008$ | $0.17 \pm 0.003$ |
| Whole Score | $0.69 \pm 0.001$ | $0.87 \pm 0.003$ | $0.87 \pm 0.003$ | $0.39 \pm 0.005$ |
| Nest Score | $0.28 \pm 0.004$ | $0.81 \pm 0.003$ | $0.82 \pm 0.004$ | $0.39 \pm 0.002$ |

Table 7: Scores of the Agglomerator

| Dataset | Ts | Squares | SHOPs | Double MNIST |
|---|---|---|---|---|
| Part Score | $-0.24 \pm 0.002$ | $-0.35 \pm 0.002$ | $-0.25 \pm 0.001$ | $-0.18 \pm 0.003$ |
| Whole Score | $-0.28 \pm 0.002$ | $-0.16 \pm 0.001$ | $-00.18 \pm 0.002$ | $0.00 \pm 0.001$ |
| Nest Score | $0.40 \pm 0.003$ | $-0.37 \pm 0.003$ | $-0.36 \pm 0.001$ | $0.01 \pm 0.004$ |

## A.8 BENCHMARKING

### A.8.1 THE SCORES OF THE COMPOSER

Here we list the scores of the Composer corresponding to Fig.10 in the main text. The value in Table.6 is the mean averaged scores on 1000 randomly selected samples and 5 random seeds are used. The error bar is very low.

### A.8.2 THE SCORES OF THE AGGLOMERATOR

Here we list the scores of the Agglomerator corresponding to Fig.10 in the main text. The value in Table.7 is the mean averaged scores on 1000 randomly selected samples and 5 random seeds are used. The error bar is very low.

### A.8.3 INTRODUCING THE AGGLOMERATOR

The Agglomerator (Garau et al., 2022) is a GLOM (Hinton, 2021) inspired implementation to deal with the part-whole hierarchy. The basic idea is to use similarity measures among vectors, which are called columns, to dynamically group neural representation into "identical islands of vectors". The spatial inclusion relationship among islands is interpreted as the part-whole hierarchy. We show the simplified architecture and representation scheme of the Agglomerator in Fig.21, in case the reader is not familiar with the model.

More specifically, columns are organized into different levels. At each level, columns are spatially located on a grid mesh, like the topographical mapping. Different levels have different spatial scales, reflected by different radii of horizontal connection among columns within the levels.

To put it in a nutshell, the Agglomerator also attempts to exploit (spatial) neuronal coherence to dynamically form the tree node at each level and to coordinate the nodes naturally by spatial nestedness. It is the comparability of the representation / architecture that we choose Agglomerator as an appropriate benchmarks and recent SOTA. However, as shown in Table.7, they failed to form the node-level representation at all in the four explicit datasets, indicated by the Part Score and Whole Score lower than 0. Since in GLOM, hierarchical coordination is not reflected by the cross-level similarity among vectors (but only by spatial relationship of islands), the Nest Score does not reveal more insight except that representations between levels are not as coherent as in the Composer.

### A.8.4 HOW COHERENCE IS MEASURED IN THE AGGLOMERATOR?

Actually, it is direct to generalize the coherence metrics to account for evaluating the identical islands:

$$\text{Part Score} = \text{Silhouette}(D_2(l_p, l_p), \text{label}_1) \tag{32}$$

$$\text{Whole Score} = \text{Silhouette}(D_2(l_w, l_w), \text{label}_2) \tag{33}$$

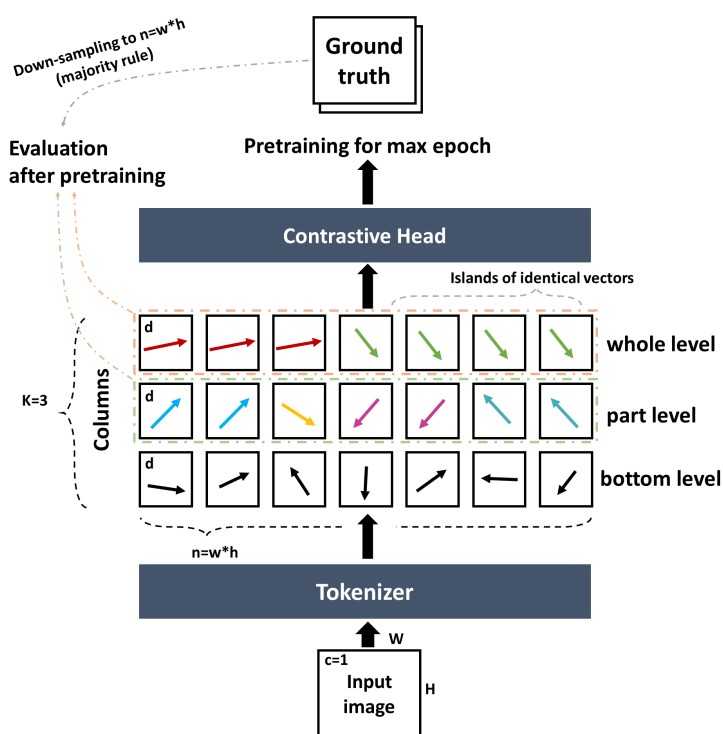

Figure 21: Illustration of the architecture of the Agglomerator: How part-whole hierarchy should be represented as spatially nested islands of identical vectors and how evaluation is achieved.

$$\text{Nest Score} = Silhouette(D_2(l_p, l_w), \text{label}_2) \tag{34}$$

where $D_2$ is the Euclidean distance metric, $l_p \in \mathbb{R}^{(w*h,d)}$ is the part-level column representation. $w$ and $h$ is the size of the grid mesh of columns, $d$ is the dimension of each column. Here, the dimension $d$ is parallel to the time dimension in the Composer. If identical islands are formed, Part / Whole Scores should reach values near 1.

### A.8.5 THE ARCHITECTURE DETAIL OF THE AGGLOMERATOR

We bear the most parameters and architecture settings from the original paper. The detailed parameter setting is shown in Table.8. An illustration of what these parameters refer to is shown in Fig.21.

Here, $K = 3$ is the number of levels, where the first level (bottom) is the output of the tokenizer and only extracted features are represented at the bottom level. Therefore, we consider the second level as the part-level and the third (top) level as the whole-level. $T$ is the number of iteration steps.

$d$ is the dimension of representation at each column and $w, h$ are the width of the grid mesh of columns (shared among levels). $n = w * h$ is the total number of columns. $c$ is the input channel number, since the datasets are all binary we choose the input channel number to be 1. $W, H$ are the size of the image. The max epoch = 100 is also consistent with the pre-training process in the original paper. The evaluation is conducted after 100 epochs of the contrastive pre-training.

In this paragraph, we present a detailed description of the key networks employed in Agglomerator. The network structure of the convolution tokenizer is outlined in Table 9. Within this table, the variable $e_d$ denotes the embedding dimension of Agglomerator, which is specifically set to 12, 12, 5, and 12 for SHOPs, Squares, Double-Digit-MNIST, and Ts datasets, respectively. The output of the convolution tokenizer network is subsequently rearranged (reshaped) and fed into the bottom-up and top-down column networks, which exhibit the structure presented in Table 10. Here, the variable $d_1$ in the bottom-up column network is set to 96, 96, 250, and 96 for the four datasets accordingly; $d_1$ in

Table 8: Parameter details for benchmarking

| Parameter | Ts | SHOPs | Squares | Double-Digit MNIST |
|---|---|---|---|---|
| K | 3 | 3 | 3 | 3 |
| d | 48 | 48 | 48 | 125 |
| T | 5 | 5 | 5 | 5 |
| w,h | 10 | 15 | 15 | 8 |
| c | 1 | 1 | 1 | 1 |
| max epoch | 100 | 100 | 100 | 300 |
| batch size | 32 | 32 | 32 | 128 |
| learning rate | 0.05 | 0.05 | 0.05 | 0.05 |
| W, H | 40 | 60 | 60 | 80 |
| [l]number of training objects | 6 | 3 | 3 | 2 |

the top-down column network is set to 48, 48, 125, and 48. $d_2$ in the bottom-up network is set to 384, 384, 1000, and 384 respectively; $d_2$ in the top-down network is set to 192, 192, 500, and 192. Additionally, $n_p$ signifies the number of patches (same as $n = w * h$ in Table.8) and takes the values 225, 225, 64, and 100, respectively. Lastly, the structure of the network utilized in contrastive learning is depicted in Table 11. In this context, $p_d$ represents the patch dimension and assumes the values 48, 48, 125, and 48 for SHOPs, Squares, Double-Digit-MNIST, and Ts datasets, correspondingly. It is worth noting that the $n_p$ values remain consistent with the settings in column networks.

Table 9: Parameter details for convolution network

| Network Structure |
|---|
| [l]Conv2d(1, $e_d$ // 2, kernel_size=(3, 3), stride=(2, 2), padding=(1, 1), bias=False) |
| BatchNorm2d($e_d$ // 2) |
| ReLU() |
| [l]Conv2d($e_d$ // 2, $e_d$ // 2, kernel_size=(3, 3), stride=(1, 1), padding=(1, 1), bias=False) |
| BatchNorm2d($e_d$ // 2) |
| ReLU() |
| [l]Conv2d($e_d$ // 2, $e\_d$, kernel_size=(3, 3), stride=(1, 1), padding=(1, 1), bias=False) |
| BatchNorm2d($e_d$ // 2) |
| ReLU() |
| [l]MaxPool2d(kernel_size=(3, 3), stride=(1, 1), padding=(1, 1), bias=False) |

Table 10: Parameter details for column network

| Network Structure |
|---|
| LayerNorm($n_p$) |
| [l]Conv1d($d_1$, $d_2$) |
| GELU (for bottom-up network) or Siren (for top-down network) |
| LayerNorm($n_p$) |
| Conv1d($d_2$, $d_1$) |

A.8.6 DOWN-SCALING OF THE GROUND TRUTH

In the Agglomerator, original $W * H$ images are firstly tokenized into $n = w * h$ patches, with $w = W/4$ and $h = H/4$. These tokenized embeddings are treated as bottom-level column representations.

Table 11: Parameter details for contrastive head

| Network Structure |
| --- |
| LayerNorm($p_d$) |
| Dropout(p=0.3) |
| Rearrange('b n d - b (n d)') |
| LayerNorm($n_p * p_d$) |
| Dropout(p=0.3) |
| Linear($n_p * p_d, n_p * p_d$) |
| LayerNorm($n_p * p_d$), |
| GELU() |
| LayerNorm($n_p * p_d$) |
| Dropout(p=0.3) |
| Linear($n_p * p_d$, 512) |

Due to the down-sampling effect of the tokenizer, the number of columns is smaller than the original pixels. Further, since the Agglomerator is super-computationally expensive, scaling as $O(w^4)$, reserving the original dimensionality of images ($w = W, h = W$) is not computationally plausible. Therefore, we impose down-sampling to the ground truth with the same reduction ratio, so that the dimension of column embedding ($w \times h \times d$) matches the down-sampled ground truth ($w \times h$). The down-sampling is based on majority rule.

### A.8.7 THE FAILURE OF THE AGGLOMERATOR

As shown in Table.7, the Agglomerator failed in all cases. As far as we know, in the original paper of the Agglomerator (Garau et al., 2022), the representation is not quantitatively evaluated in terms of the hierarchical structure but for classification accuracy and object detection. Also, in the visualization, the representation across levels is more likely to extract features at different scales, instead of forming an interpretable part-whole hierarchy. As far as we understand, although the motivation and basic idea of the representation hypothesis are very promising, the representation hypothesis is not explicitly verified or falsified by quantitative metric. As shown in Table.7, it is worth re-evaluating the parsing ability of the Agglomerator on images with more 'explicit' part-whole relationships and more appropriate 'quantitative metrics' as we do.

Another explanation is the symmetry in our dataset. In the original paper, the seeming parsing of different parts is likely due to the different colors associated with the parts. In other words, It is observed that the Agglomerator only groups locations of similar color into islands (like feature extraction), instead of parsing the object based on knowledge of the object-centric representation (e.g. same object can have different colors and different objects can share the same color). To put it in another way, the grouping in the Agglomerator is due to the external asymmetry in the scene, e.g. different colors. However, in our dataset, all objects have the same color (black), and the symmetry-breaking process for grouping needs to occur internally. Therefore, the symmetry (shared color among objects) can challenge the Agglomerator to parse the scene.

On the other hand, failure on our synthetic dataset and metrics neither excludes its potential validity in other cases nor excludes the possibility that it can be improved to solve the problems. Compared with our model, the Aggglomerator's architecture is more flexible in dealing with real-world images, and training on larger real-world datasets can be very different from training on small-scale synthetic datasets. Therefore, the limitation of our dataset and metric is also notable. Besides, the downsampling of the ground truth is likely to magnify the failure of the Agglomerator since the Silhouette Score is more sensitive to incoherence when the sample number is lower ($w \times h < W \times H$). It is partially the reason why the score tends to be lower than 0. However, the results indeed show that SOTA models can fail when explicitly evaluated.

To put it in a nutshell, we highlight that the problem of representing part-whole hierarchy needs to be more explicitly evaluated to verify or falsify the proposed model. As far as we know, representing the part-whole hierarchy is far from being solved and combining real-world data and synthetic data is essential to evaluate models in the future.

A.9  Details on experiments and additional results.

### A.9.1  Convergence

In Fig.8, we show the convergence of scores along the iteration. 100 randomly selected samples are used,and the score are evaluated every 100 time steps (so 3000/100=30 data point in total for each score).

Convergence on the SHOPs dataset achieves the best overall results. However, the potential overlap of part-level objects when composing the whole-level object imposes additional challenges on recognizing the part-level objects, indicated by the relatively lower Part Score in Fig.8a.

Convergence on the Squares dataset is very interesting. On the one hand, the Whole Score takes the lead all the time, indicating that global information is firstly recognized by the Composer, which is very similar to human vision (Lee & Nguyen, 2001) and is also consistent with Gestalt psychology. On the other hand, the Part Score and Nest Score undergo an initial descending period before going up. Here, we explain this phenomenon: Compared with the very starting phase, where spikes are randomly and densely fired, the emergence of whole-level squares around $500 \sim 1000$ time steps provide new conditions on the part level. While this has benefits in the long run, it could degrade the representation in the short run, because the Composer needs to rethink its representation and make modifications. For example, the part-level firing becomes sparser, and there are more incorrect synchronizations. This may degrade the part-level grouping and hierarchical coordination (nestedness). In other words, the fact that each whole object is composed of four parts complicates the self-correcting / searching process, after the whole-level objects are recognized. Fortunately, after a short period of self-correcting, the Scores go up again and gradually converge to expected neuronal coherence as in other cases.

Convergence on the Ts dataset is very challenging due to the object number is much larger. On the one hand, the Composer needs to distinguish 6 wholes and 12 parts. On the other hand, 6 whole objects and 12 part objects impose $6^12$ potential combinations of part-whole relationships (each part can choose to belong to one of the six wholes). Therefore, it takes time to search for / sample the optimal configuration. Even if the parts/wholes are grouped by neuronal coherence, there is a high possibility that the parts and wholes are not well coordinated to form proper nested structure. Since the neural computation in the brain can also be regarded as sampling (Buesing et al., 2011), these challenges may also cause problems in perception like the binding problem (Engel & Singer, 2001; Von der Malsburg, 1999). Therefore, on the Ts dataset, the Nest Score lags behind the other scores.

Convergence on Double-Digit MNIST is also challenging for the Composer because the objects are of much higher diversity. Therefore, it is harder for the Composer to clearly distinguish the objects and to form well-synchronized cell assemblies. Therefore, the Part Score is lower than other scores and the variance is higher than in other cases. However, it is surprising that the Composer still achieves good parsing, indicated by the convergent Nest Score, even though objects are less recognizable.

### A.9.2  Visualization

In Fig.6 in the main text, we visualize the spiking pattern, attention map, and local field potential along the convergent process. To better visualize the cell assemblies, we reorder the index of neurons on the y-axis (Fig.6c) so that neurons encoding the same object are close on the y-axis. Besides, in order to distinguish different cell assemblies more clearly, we color the spikes based on the ground truth assignment of the neurons in SCS (= pixels in the image), so that the color of the cell assembly indicates what object the cell assembly represents. In other words, the represented objects can be directly recognized by comparing the color of the cell assembly and ground truth. This fact can be verified by comparing the circled cell assemblies in Fig.6c, the circled zoomed-in spike patterns in Fig.6d, and the circled objects in the ground truth (Fig.6a). It is clear that the synchronized cell assemblies gradually emerges from randomness along the simulation. Each synchronized cell assembly represents the parts/wholes of the object. Cell assemblies at different levels are coordinated properly according to the part-whole relationship.

Surprisingly, the emergent parsing tree reform itself during the dynamics in Fig.6 c: From phase II to phase III, the part assemblies (colored by deep and light green) reversed their order! This observation demonstrates the multi-stable property of the part-whole solution in the Composer, which is a dynamical system in nature. This property is essential to account for the uncertainty and diversity

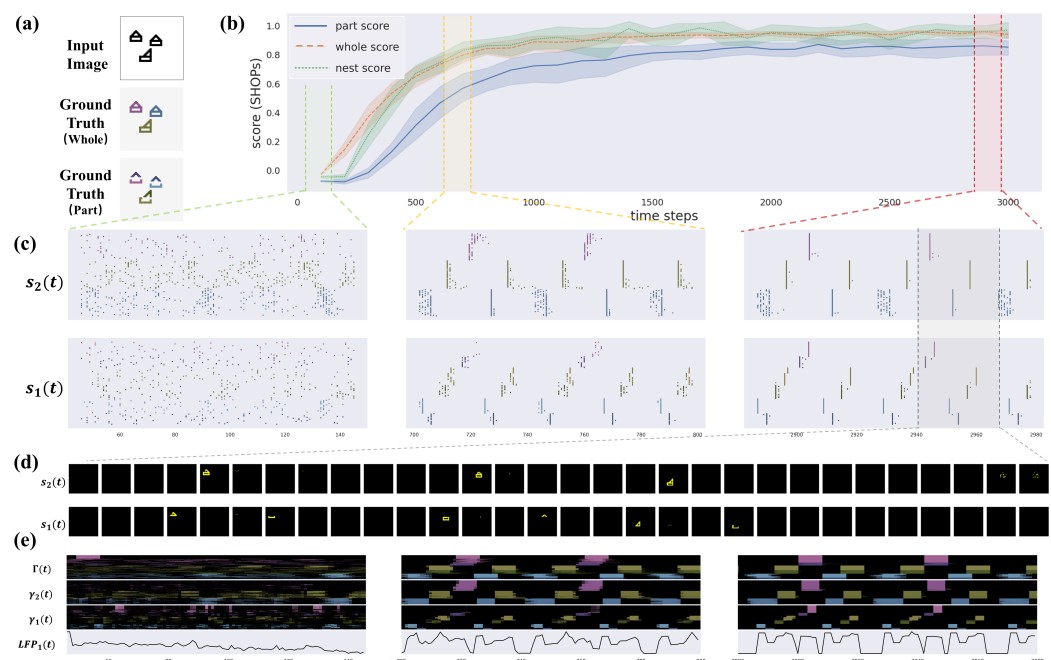

Figure 22: Additional visualization on SHOPs dataset.

of the part-whole relationship and challenges most traditional artificial neural networks with only feedforward connection or only by supervised training (Greff et al., 2020).

In Fig.6e, it is also observed that different types of top-down attention also emerge into structured patterns. To keep consistent with Fig.6c, the neuron indexes are also reordered and the attention map is also colored based on the ground truth. The depth of the color reflects the value of the attention map. It is observed that the structured pattern has the same order as the spiking pattern, yet of long timescales. This indicates that attention plays a role in modulating the spike timing in SCS. However, such modulation is not single-way, but a iterative interplay between bottom-up integration and top-down modulation. Therefore, both DAE and SCS play essential roles in solving the parsing problem.

In Fig.6, we also shows the emergence of the oscillatory LFP at the part level, which is the summed top-down feedback: $LFP_1(t) = \sum_{i=1}^{N} \gamma_{1i}(t)$, where $i$ is the neuron index in the part level.

### A.9.3 MORE VISUALIZATIONS

In Fig.7, we briefly show the visualization results on other datasets. Here we provide more detailed visualization results on the four datasets Fig.22 to Fig.29. Two cases are provided for each dataset, including one normal case and one fail case.

We also provide a zip file containing videos to visualize temporal evolution of neuronal activities in SI, about 60MB.

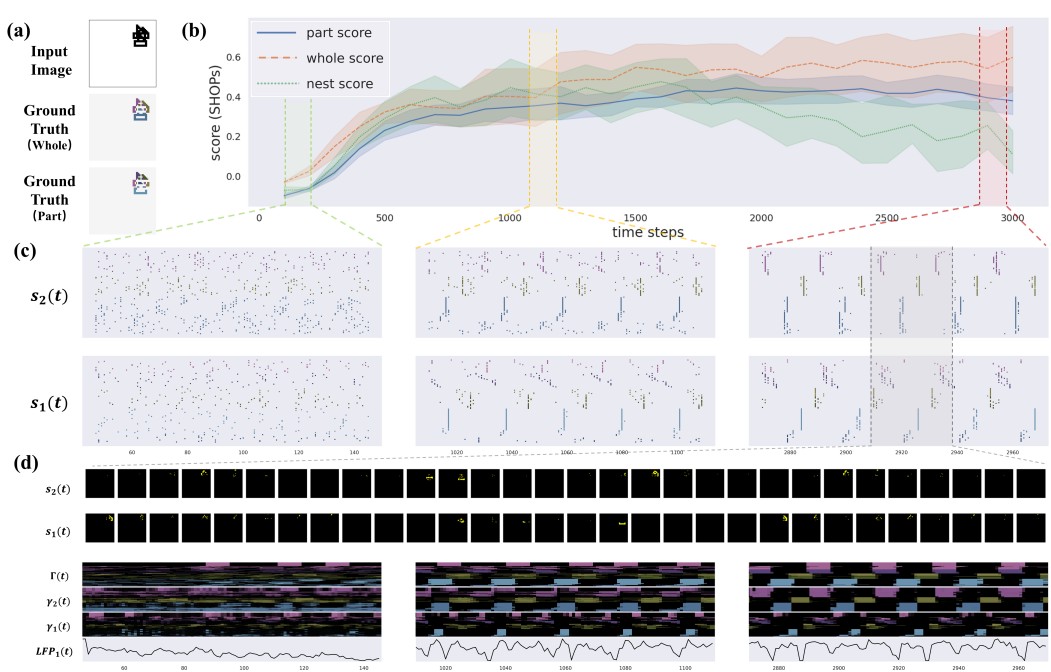

Figure 23: Additional visualization on SHOPs dataset: The fail case, when objects sickly overlap.

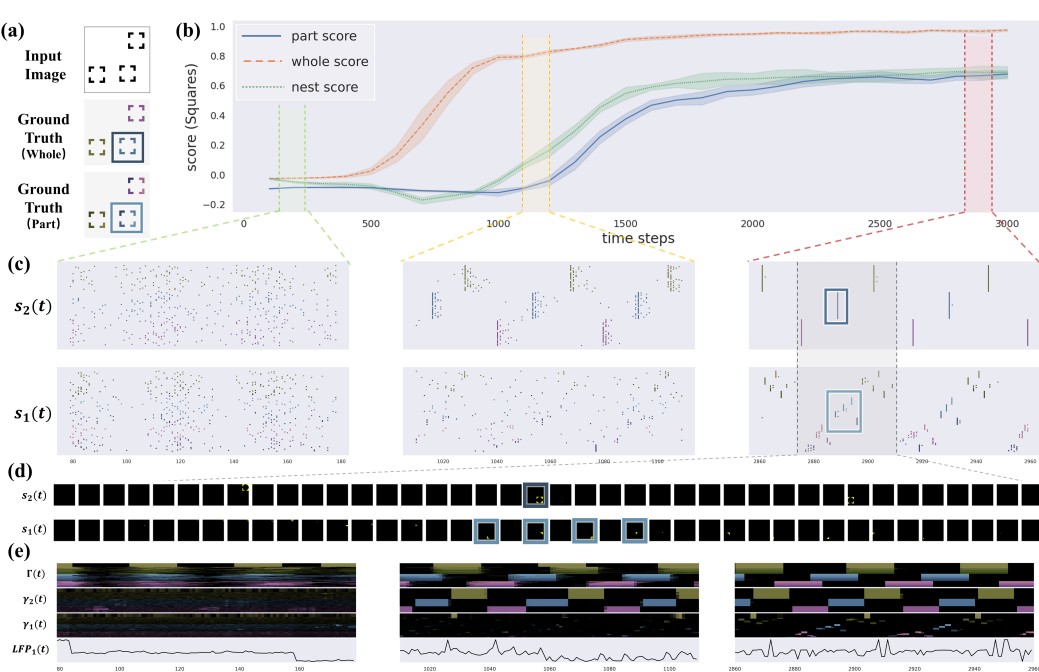

Figure 24: Additional visualization on Squares dataset: Squares are not overlap.

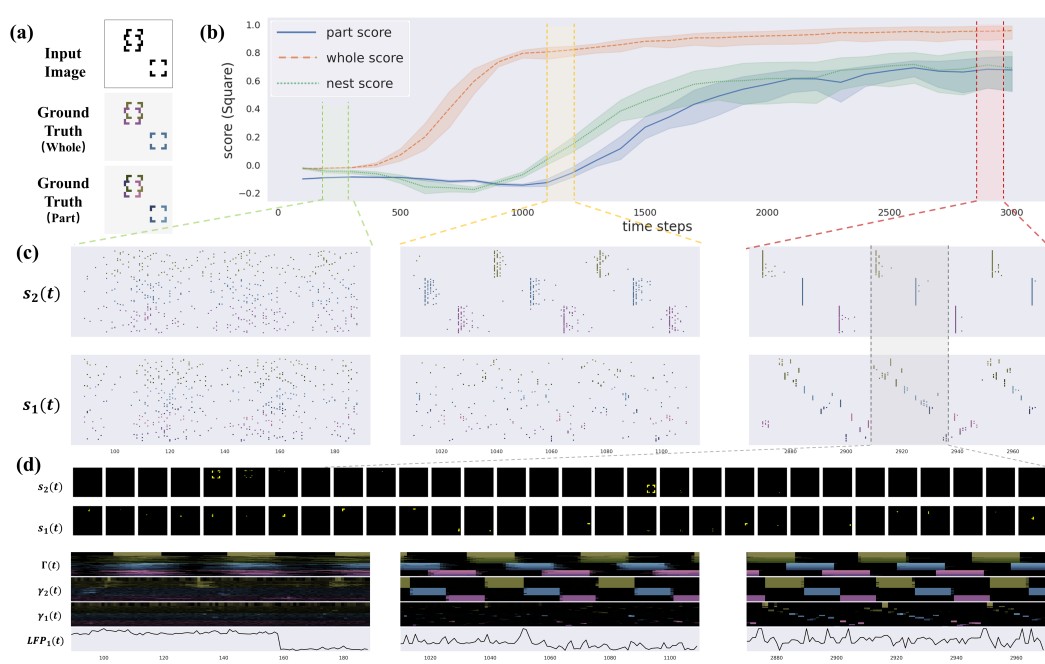

Figure 25: Additional visualization on Squares dataset: Two Squares heavily overlap but the nested structure remains.

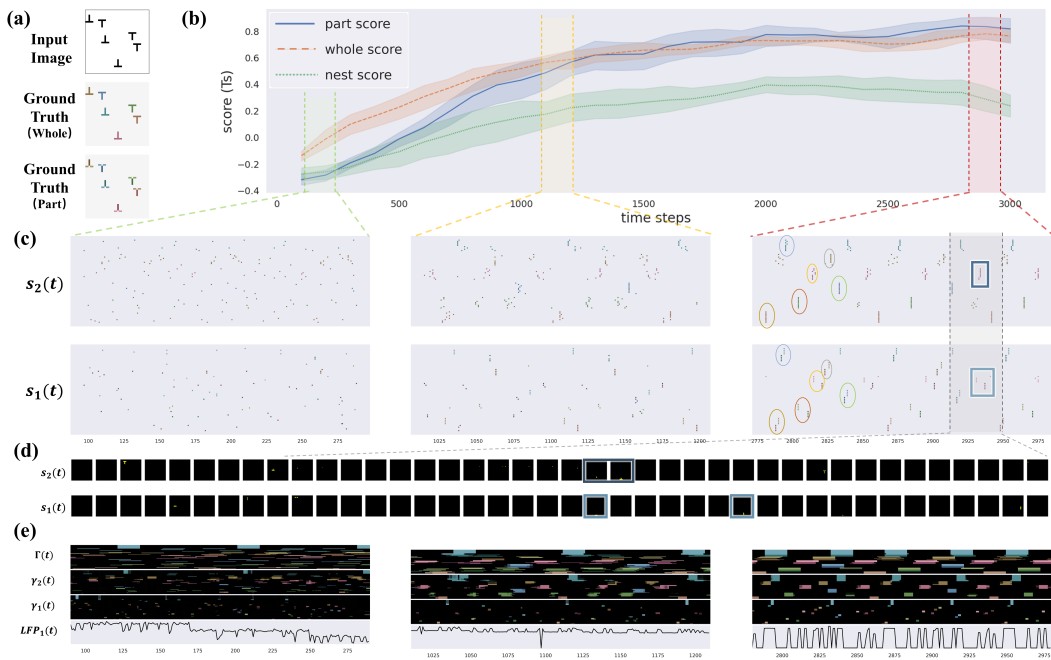

Figure 26: Additional visualization on Ts dataset. Colored circled indicates the coordinated cell assemblies. Same color indicates the part-whole relationship.

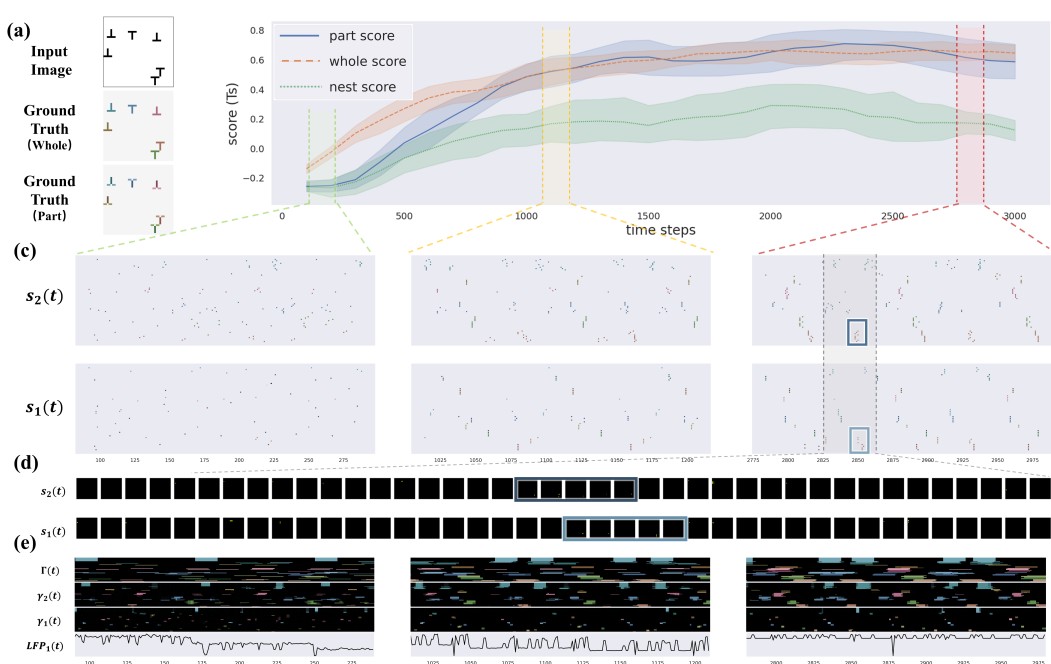

Figure 27: Additional visualization on Ts dataset. Nestedness is not as clear as fig.26.

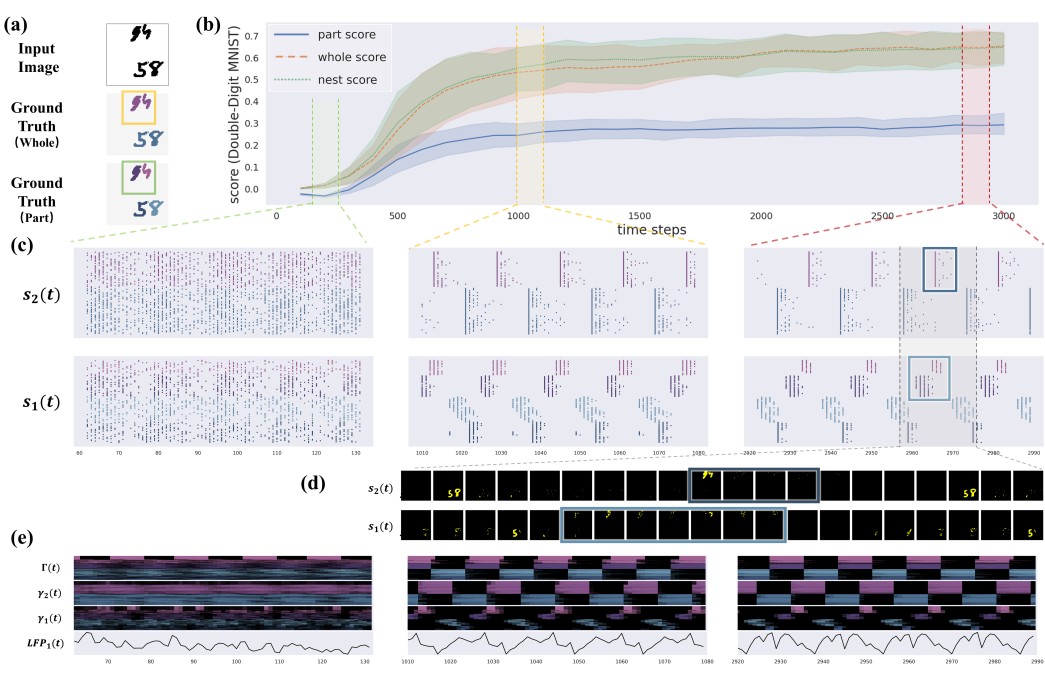

Figure 28: Additional visualization on Double-Digit-MNIST dataset.

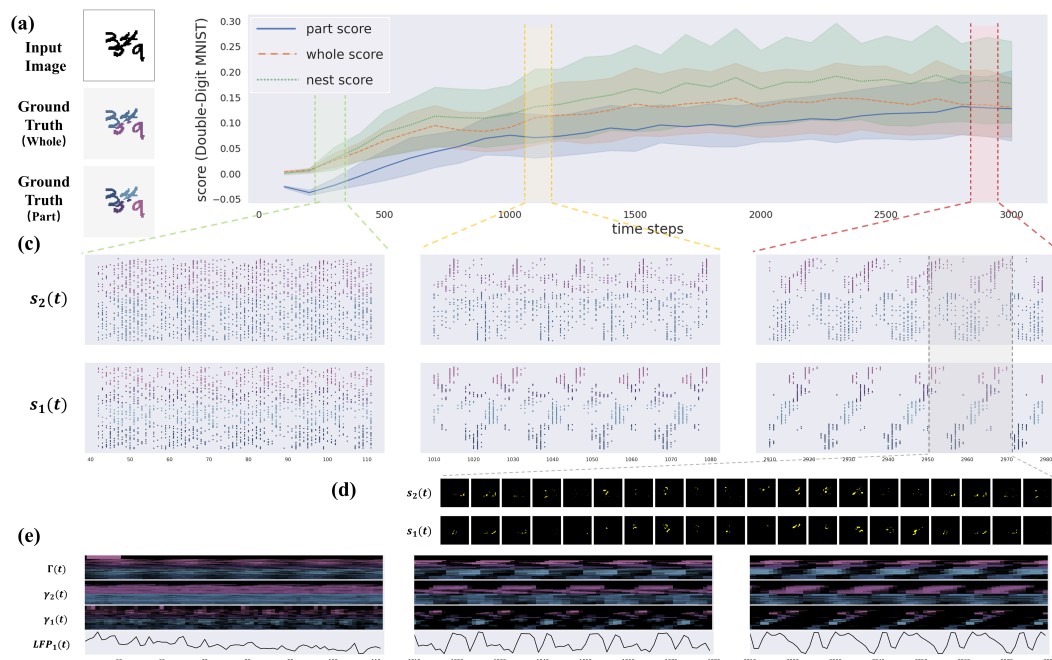

Figure 29: Additional visualization on Double-Digit-MNIST dataset: Digits are crowded and the coherence structure is weaker.

### A.9.4 MULTI-STABILITY

In Fig.1 of the main text, we demonstrated one of the challenge of representing part-whole hierarchy as the diverse interpretation of a given visual scene, based on self-consistency among contents. To showcase that Composer is capable of generating multiple part-whole relationship given an ambiguous visual scene, we uses a new dataset, Teris dataset. Part objects are squares, and whole objects are composed of 4 squares in various ways (Fig.30). So there are 15 possibilities.

We follow the same training protocol as introduced in A.6. Then we test the parsing result given an ambiguous image (Fig.30top-left), which is composed of two Teris objects. And there are alternative interpretations, shown in Fig.30 middle and right respectively.

To visualize the part-whole structure of spiking activity, we assign a color to each block in the image (Fig.30bottom-left), and color-code the neurons / spikes based on it. This process is similar to the main text. However, since the whole-level grouping is ambiguous, there is no "ground truth" at all and the coloring is only for visualization purpose.

Surprisingly, we observed that two different type of part-whole hierarchy emerges, corresponding to different interpretation of the ambiguous image. This addiction result showcase the content-sensitive structure inference ability of Composer.

### A.9.5 THREE-LEVELS

As argued in the main text, the proposed framework can be generalized to more than 2 levels in principle, because the columns and cross-level pathways can be reused for multiple levels. Here, we provide an implementation of such generalization, where a three-level Composer are realized to parse samples in the SHOPs dataset. For SHOPs, We take the whole scene as the additional level: scene level.

As shown in Fig.31, hierarchically organized cell assemblies and hierarchical attention sequence emerges. Since the top level is the whole scene, there is no need to switch. So the scene assemblies slowly oscillate collectively and the top-level attention sustains. Here, for simplicity, we use whole-level ground truth to color the neuron / spikes at the scene-level.

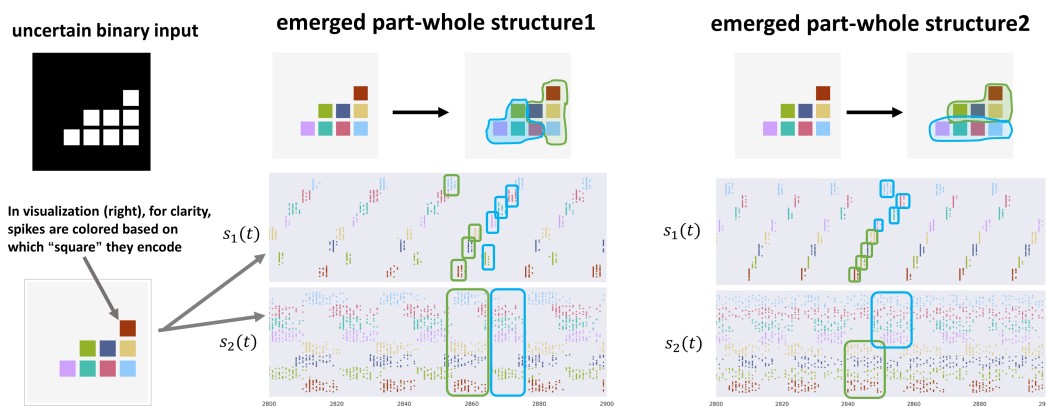

Figure 30: The composer can infer different reasonable part-whole structures when the input is "ambiguous" in terms of compositional structure. We train the model in a new Tetris dataset. Part objects are squares, and whole objects are composed of 4 squares in various ways (15 possibilities). Then when it tries to decompose the ambiguous input. Multiple part-whole structure emerges as mentioned in Fig1 in main paper.

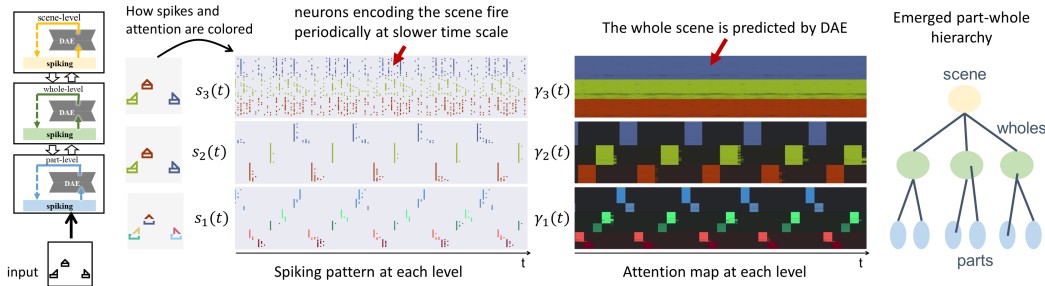

Figure 31: Extending the Composer to 3 levels to account for scene-level, the top-level neurons fires coherently at slow time scale ($s_3(t)$).

This additional result offers evidence that the Composer architecture is capable of being generalized to additional levels.