# OpenReview forum: "Representing Part-Whole Hierarchy with Nested Neuronal Coherence"
_ICLR.cc/2025/Conference — Submitted to ICLR 2025_

### Official Review · Reviewer_9gYW · 2024-11-04

**Soundness:** 2
**Presentation:** 3
**Contribution:** 2
**Rating:** 5
**Confidence:** 3

**Summary:**

This paper proposed a model called composer that can identify objects and their constituent parts in the pixel space. This model is based on two levels of interconnected spiking neurons and denoising auto-encoders (DAEs) representing whole/part levels. At each level, the DAE recovers the shapes remembered during pre-training from noisy states. Meanwhile, the coupling delay and refractory period of the spiking neurons make the neural activities move out of the steady state and transition to other potential steady states that the DAE could recover. The two levels representing the whole and parts are interconnected, enabling nesting, where the constituent parts of a whole object are identified. The author showed that the model can identify the objects, their constituent parts, and the nesting relationship in several synthetic datasets involving identifying simple shapes and their parts.

**Strengths:**

This paper studies and proposes a model that identifies and represents objects and their constituent parts. The model uses two levels of spiking neurons and denoising autoencoders to recover part-whole hierarchy in the dynamic neural states, a novel neural architecture for this problem. They showed with concrete evidence that this model could work and demonstrated it on four different synthetic datasets they developed. They also proposed clearly defined evaluation metrics -- the part/whole/nest score -- that measure how well spiking neurons represent the desired structure. According to these metrics, the composer model performs better than another method, Agglomerator, on their dataset.

**Weaknesses:**

While the theme of this paper is about representing the part-whole hierarchy, it doesn't explain why explicitly representing this hierarchy is an important question to solve. As also noted by the author, "real-world images typically lack explicit parsing structures." This makes me wonder whether it is a well-posed question that the brain or brain-like intelligent system needs to represent the part-whole hierarchy in an explicit manner. What problem in the real world may benefit from explicitly representing such a hierarchy? Maybe it is the ability to generalize out of distribution, the ability to reason, or something else. It would be helpful if the author could explain what makes
explicitly representing part-whole hierarchy in the proposed model a desired approach. What new abilities does the system have that other methods do not have, and what problem does this model solve that other models cannot solve?

This also connects to my concern about the benchmark results in this paper, which showed that their method outperformed the previous SOTA model Agglomerator in the four tasks they evaluated. According to their metrics, the previous SOTA model Agglomerator only performed randomly or even worse than randomly in these tasks. However, these tasks involving identifying pre-defined simple shapes and their parts seem like very simple problems that various baseline methods, such as template matching, convolutional networks, or Bayesian inference methods, could solve. It would be helpful if the author could also show the results with these baseline methods and clarify if performances are gained from the composer model or to further explain why these baseline methods are unsuitable for the tasks they studied.

Other minor points:
Figure 1 of the paper motivated that representing the part-whole hierarchy is challenging because the interpretation of the parts and wholes can be ambiguous. This paper has suggested multiple times that slot-based models are limited due to their inability to express uncertainty, and the composer model seems to resolve this problem. However, there isn't a metric in the paper that quantifies the uncertainty of their model, nor did they have a task to evaluate how the model resolves the ambiguous case motivated by Figure 1. It would be helpful if the author could explain how the model expresses uncertainty and how that could be validated in ambiguous cases motivated in Figure 1.

I hope this point could help. It is not a part of my decision assessment:
This paper presented a model inspired by the brain's structure and mechanisms, and motivated that this model could act as a data-driven biological model to understand the brain. However, I could not find any comparison to real behavioral or neural data in this paper. Along that direction, it would be great if future work could show more concrete, well-defined comparisons with real neural data in the brain or human/animal behavior if the model is intended as a model of the brain.

**Questions:**

1. The four tasks used in the benchmark seem like very simple problems that various baseline methods, such as template matching, convolutional networks, or Bayesian inference methods, could solve. How does this proposed model compare with these baseline models, or can the author clarify why these baseline models are unsuitable for these tasks?

2. What problem in the real world may benefit from explicitly representing part/whole hierarchy, and how does this proposed model make progress in solving those problems? What new abilities does the proposed model have that other existing methods do not have, and what problem does this model solve that other existing methods cannot solve?

3. How does this model represent uncertainty, and how do we quantify that? Why is representing uncertainty better than not representing it?

---

> ### Author Response · Authors · 2024-11-16
> **Discussion with The Reviewer**
>
> We thank the reviewer for taking the time to read the paper and providing very detailed comments. The questions and comments are very thoughtful and we consider them carefully. We would like to exchange our thoughts on it as below.
>
> # The motivation of representing part-whole hierarchy
>
> ## The importance of object-centric representation [1]
> The object-centric representation has argued to be essential for combinatorial generalization [1]: how to group distributed features into a symbol-like entity ? Such symbol-like representation is helpful to (be reused to) build-up compositional representation (more complex objects), and to infer relations among objects [1]. Here, object is not limited to visual objects, but can be generalized to words / sentence / concepts. For example, GNN's power on reasoning among objects largely attributes to its explicit representing each symbolic objects as "node vector", which is an instance slots [1] (but these slots has a predefined nature and are discrete, this implicate certain limitation of GNN.)
>
> [1] Klaus Greff, Sjoerd Van Steenkiste, and Jürgen Schmidhuber. On the binding problem in artificial
> neural networks. 2020.
>
> ## The hierarchical nature of objects
> As argued in [1] (sec 4.1), hierarchy is a fundamental nature of objects or at least how we understand / generate objects in the brain. e.g. word vs sentence; concept hierarchy, etc. For visual domain, Hinton finds psychological evidence for the part-whole relationship of visual objects when human perceive an object (different levels can interact) during his posdoctual period [3] and tried to figure out how distributed neural network represent such symbolic structure in a flexible manner later. This is the underliying motivation of a list of capsule networks he proposed: he imagined each capsule can represent an object so that different layers of capsules can represent object-hierarchy. But it turns out that no expected hierarchical relations emerged in those capsule networks. In his 2021 paper [2] and related talks and videos, he summarized his thoughts on this issue: why part-whole relationship is a very important but missing point in machine vision and why it challenges most current neural architectures.
>
> [2] Geoffrey E. Hinton. How to represent part-whole hierarchies in a neural network. Neural Computation, 2021.
>
> [3] Geoffrey E. Hinton. Some demonstrations of the effects of structural descriptions in mental imagery.
> Cogn. Sci., 1979.
>
> ## The importance of representation
> Is it necessay to represent all the hierarchy as neural activations, or is it just needed to somehow process the relevant information to give right answers to questions that related to the hierarchy. There is a reason to think in either way, and it might be a matter of belief or hypothesis. On the one side, human do not just answer reasoning questions: given a visual scene, we could "see" both parts and wholes almost at the same time (~500ms) and also recognize the hierarchical relationship among them without efforts. So it is likely that we represent all these things as neural activations. On the other side, forming such a hierarchical representation is a stronger hypothesis than answering questions related to hierarchy. If it can be achieved, it provides a much stronger guarentee for compositional representation (thus generalization) with high interpretability, and potentially high data efficiency, and a posibility to diagnose the issue when the outcome is not as expected (e.g. compositional error and hallucination in LLM). From the history, we have witnessed that how the rethinking in representation helps us to make computation / learning more efficient (it indicates whether there is a possible solution at all and how likely it is to find that solution,and what inductive bias we may need). e.g. binding problem[1]. part-whole relationship is another case.
>
> ## The importance of coherence-based representation
> The transition from symbolic system to neural network is mainly due to the fact that the continuous and distributed representation in neural network facilitate learning and inference in uncertainty; Learning in traditional symbolic system needs manually add / delete certain nodes or rules, due to its discrete nature [4]. And the discrete structure itself can not be continuously infered from the content [4]. In the main paper, we argued that the discrete and pre-defined nature of slots limits its capability to "infer the slots" or "learn the slots" from uncertain content similiarly. And generalizing slots into dynamically, distributed, continulus "neuronal coherence" is a important direction to  unlock these limitation (structure inference and learning the structure itself.)  We are not all there yet, but to showcase the feasibility of such a framework is the first step.
>
> [4] Timothy T Rogers and James L McClelland. Parallel distributed processing at 25: Further explorations
> in the microstructure of cognition. Cognitive science, 2014.

---

> ### Author Response · Authors · 2024-11-16
> **Discussion with The Reviewer (continued)**
>
> For the motivation, besides the general argument above, we provide a more direct answer to each comments:
> ##  "real-world images typically lack explicit parsing structures." This makes me wonder whether it is a well-posed question that the brain or brain-like intelligent system needs to represent the part-whole hierarchy in an explicit manner.
>
> What we mean here is that, there is no ground truth for objects and hierarchy: it depends. This poses the challenges for the models dependent on supervised learning by ground-truth labels. But like clustering problem or unsupervised representation learning, it doesn't mean that there should not be such a representation or there is no way to quantify such representation. The general idea is that, the representation is like unsupervise clustering and the quality of representation can be partially quantified as internal coherence (e.g. synchrony), like that used in clustering (e.g. Silhouette score). And this question is argued to be a cornerstone for human-like machine vision if ever solved [2]. For the relation to the brain, the representation hypothesis proposed in this paper is consistent to the neural syntax hypothsis proposed by Buzsaki [5]. Also, it is hard to imagine how we can make sense of objects with parts and wholes (we "see" them all), if there is not a representation realized by neural activation. If we acknowledge that it is a well-posed question, then, it is a very interesting to think about how "many"-symbol-like "parts" and "wholes", can flexibly emerge in the distributed representation space and are well organized into a structure.
>
> [5] György Buzsáki. Neural syntax: Cell assemblies, synapsembles, and readers. Neuron, 68:362–385,
> 2010
>
> ##  What problem in the real world may benefit from explicitly representing such a hierarchy? Maybe it is the ability to generalize out of distribution, the ability to reason, or something else. It would be helpful if the author could explain what makes explicitly representing part-whole hierarchy in the proposed model a desired approach. What new abilities does the system have that other methods do not have, and what problem does this model solve that other models cannot solve?
>
> Yes. The compositionality benefit from compositional representations. In [1], it has been argued that various shortcoming in ANNs, like tranferability, distribution shift, OOD generalization, are rooted in the representation (e.g. see fig1 in [1] and relevant arguments), which is focused in this paper. We suspect that the compositional error and halluciation in LLM may argubaly due to limitations of its object-centric representation (feature can interfere among different objects like that in the binding problem [1]) The core feature of the framework is (1) interpretability: if the part-whole hierarchy is represented in the network activation, the representation would be super-interpretable, and we know how distrubted features are organized. (2) coherence level implicitly indicate uncertainty, even in a level-wise manner; (3) dynamically representing part-whole relationship makes different level interact with each other to form a coherent solution, so that a change in one level can affect other levels, this could in turn make the object-based reasoning and inference more efficient; (4) Given many objects, all possible relation among them scales exponentially. And explicitly representing the part-whole hierarchy can reduce the inference complexity scale with number of levels. (5) The continuous nature of coherence make it feasible to learn the structure (how many objects at each level) based on the statistic of the dataset. It is a limitation of slot-based models, it is hard to add / delete discrete slots continuously by learning. Showing this possibility is our future work.
>
> In general, this paper focuses on laying a foundation of a new framework. A lot of future work can be done to exploit the potential benefit of such representation, to link that to reasoning tasks, or supervised learning, and to link that to neuroscience data.

---

> ### Author Response · Authors · 2024-11-16
> **Discussion with Reviewer (Continued2)**
>
> ## such as template matching, convolutional networks, or Bayesian inference methods, could solve. It would be helpful if the author could also show the results with these baseline methods and clarify if performances are gained from the composer model or to further explain why these baseline methods are unsuitable for the tasks they studied.
>
> In [1], it has been argued why traditional method like template matching or markov random field can not solve object-centric represnetation and it is basiclly due to the complexity of objects, see sec 4.3.1 in [1] for details. Briefly, the traditional approach has strong priors of objects and images, and are limited to only deal with superficial features (pixels) with expected spatial organization. This can not capture the complexity of objects. The method proposed in this paper, based on generative approach (e.g. DAE), while showcased in pixel-level grouping on toy datasets, are not fundamentally limited to pixel-level grouping and can be generalized to other domains. CNNs, as far as we know, do not have object-centric representation at all (no grouping), and suffer from this (fig1 in [1]). They have hierarchical layers, but do not have have part-whole hierarchy of "object-centric representation". For Bayesian approach, there are hierarchical bayesian models, but these models belongs to (instance) slot-based models, because they need to explicitly divide the latent space to represent a fixed number of objects, when they assign the latent variables: to see this, just remind that when we use GMM, we need to identify the "number" of individual Gaussian distributions (each Gaussian is pre-defined to be an object in these models, see Fig 16 in [1] for a demonstration). Here, we are looking for a fundamentally different roadmap. These models are undoubtly capable of dealing with the toy tasks in this paper, as the author suggested, but they are unsuitable in sense of the questions we raised, due to the conceptual limitation we argued above. On the other hand, our model do not have these limitations and in the appendix, we have discussed how our model can be technically extended into more general cases.
>
> ## However, there isn't a metric in the paper that quantifies the uncertainty of their model, nor did they have a task to evaluate how the model resolves the ambiguous case motivated by Figure 1. It would be helpful if the author could explain how the model expresses uncertainty and how that could be validated in ambiguous cases motivated in Figure 1.
>
> The three scores in the paper is a measure of synchrony and therefore a measure of the uncertainty (with respect to different aspects, uncertainty of node of different levels or uncertainty of edges). Alternatively, we could use any synchrony measure of spike trains to quantify the uncertainty, like that in [6]. The representation of uncertainty is beneficial because it could inform the downstream to what extent these information is reliable during the reasoning, to prevent amplifying the error. eg the emergence of the hierarchy itself is a process of such inference. For the ambiguity, we indeed provided a result to demonstrate this in Appendix, please see Fig 30 in Appendix. It is indeed an important feature of our model.
>
> [6] Hao Zheng, Hui Lin, Rong Zhao, and Luping Shi. Dance of snn and ann: Solving
> binding problem by combining spike timing and reconstructive attention. Advances in Neural Information Processing Systems,
> 2022
>
> ## This paper presented a model inspired by the brain's structure and mechanisms, and motivated that this model could act as a data-driven biological model to understand the brain. However, I could not find any comparison to real behavioral or neural data in this paper.
>
> We thank the reviewer for point it out. Yes, we are also looking for evidence for it in the neural data. But it is notable how challenging it is to find those correlated distributed cell assemblies in the brain. The longer-time-scale, larger-spatial-scale cell assemblies are much harder to identify and related to stimulus than shorter/smaller scale assemblies in vivo. They are not necessarily localized in a local region. However, the existence of such hierarchical organization of correlated cell assemblies and its functional role to represent part-whole relationship is a strong hypothesis in neuroscience [5]. Besides, we provides discussion on the bio-plausibility of various setting of our model in Appendix A.7. We are aware that it is still open question how brain represent objects or concepts and we do not attempt to over-claim that this model models brain. However, we believe that the overall framework is helpful to realize these hypothesis in a data-driven manner, and therefore generate interesting results that may have a predictive power for neuroscience studies. e.g. if combined with a downstream supervised task or LLM, we could ask how different task or cues, influence the neural code (e.g. neuronal coherence structure).

---

> ### Comment · Reviewer_9gYW · 2024-12-03
>
> Thank you for your detailed response.
>
> - My concerns about the motivation behind representing the part-whole hierarchy explicitly are partly addressed. Thanks for pointing me to these references. I think the benefits of interpretability, efficiency, and representation of uncertainty are very good points. I think perhaps it would be helpful if the author could add some similar discussions at the beginning of the paper. Part of the reason I raised this point was that the paper directly claims that explicitly representing the part-whole hierarchies in networks is a challenge at the beginning without motivating much about the reasons why this is a desirable approach. I think the rebuttal only partly addresses my concern: most of the arguments are still very conceptual. It can be made much more powerful if the author can point to a concrete benchmark, on which, perhaps, all of the leading models are models representing part-whole hierarchy explicitly, while models without this explicit representation fail.
>
> - My major concerns about the simplicity of the task used for evaluation and the lack of baseline models (which is also what I see as the major limitation of the paper) remain. The author argued that template matching and Markov random field can not capture the complexity of objects, while this limitation also applies to the model proposed in this paper. The argument that "CNNs do not have a part-whole hierarchy of object-centric representation" (therefore implying that CNNs are worse than models that represent a part-whole hierarchy) sounds cyclical and unconvincing. The author argued that Bayesian models need to represent a fixed number of latents. While GMMs may need a predefined number of latents in the model, non-parametric Bayesian models can have a dynamic number of latent factors (Gershman & Blei, 2012). In summary, many of these arguments are highly conceptual and not convincing to me due to a lack of grounding in real examples. These conceptual limitations are not the reason for not testing these baseline models. Instead, it is much more desirable and would make this paper much stronger if the author could identify a concrete task or benchmark and show the performance gain (or gains in other desired properties) of the proposed model compared with these baseline models.
>
> For the reasons given above, I will maintain my score for now.
>
> Reference:
> 1. Gershman, S.J. and Blei, D.M., 2012. A tutorial on Bayesian nonparametric models. Journal of Mathematical Psychology, 56(1), pp.1-12.

---

### Official Review · Reviewer_A4t8 · 2024-11-04

**Soundness:** 2
**Presentation:** 2
**Contribution:** 2
**Rating:** 5
**Confidence:** 4

**Summary:**

In this paper, the authors propose a novel neurally inspired architecture named COMPOSER that learns to perform hierarchical grouping of images into its constituent parts and sub parts. The authors develop new datasets and metrics, evaluating on which they show that COMPOSER is able to produce emergent hierarchical grouping of scenes via neural synchrony.

**Strengths:**

* Hierarchical grouping of images is a marker of biological intelligence. Training a neural model that mimics this ability of humans in an interpretable manner is a very interesting research direction.
* The paper's figures are of great quality and aid the understanding of COMPOSER, which is quite an intricate architecture with many moving parts.
* The authors present elaborate analyses on their proposed datasets highlighting how neural synchrony enables hierarchical grouping from images.

**Weaknesses:**

* **Lack of comparison to other comparable baselines**: The authors don't perform comparisons with other baselines, like Slot Attention, but they don't perform comparisons with these models (which have publicly available implementations) on the evaluation datasets. This is a major drawback as it is unclear how the proposed model is improving on existing art in neural perceptual grouping.
* **Overly simplistic evaluation datasets**: Several works exist which perform grouping on more complex naturalistic scenes (see [1]), yet the current submission evaluates models on very simple stimuli. It is possible the authors are evaluating on simple stimuli owing to the scalability issue of spiking neural networks, however, it is unclear how the current method is advancing over existing art.
* **On the compute efficiency of COMPOSER**: The authors must evaluate the compute and learning complexity of COMPOSER in comparison to prior art. Can there be a comparison on the number of FLOPS, model parameters or size between COMPOSER and other existing models?


References:
1. Ranasinghe, K., McKinzie, B., Ravi, S., Yang, Y., Toshev, A., & Shlens, J. (2023). Perceptual grouping in contrastive vision-language models. In Proceedings of the IEEE/CVF International Conference on Computer Vision (pp. 5571-5584).

**Questions:**

Please refer to my weaknesses section of the review.

---

> ### Author Response · Authors · 2024-11-17
> **Discussion with the reviewer**
>
> We thank the reviewer for acknowledging the value of our work and raising the very sharp question. We would like to try our best to resolve these concerns.
>
> ## Lack of comparison to other comparable baselines
> It is notable that the question we focus on is representing part-whole relationship, instead of single-level grouping / segmentation, which is mostly missing in current object-centric literatures. Either slot-attention or the paper the reviewer referenced is for single-level segmentation, and they are not capable of representing hierarchical structures. It is also notable that representing part-whole relationship is a much harder question than single-level object-centric representation: we need to represent the "relation" among objects instead of only representing the object itself. So it needs to rethink how this "relationship" should be represented in the network, especiilay if we want to represent it somehow as "neural activations" instead of in the connection weights. And how the representation of relation co-exist with many representaion of objects. And how the representation of relation interact with the representation of objects. And how they are disentangled so that we could figure out which is which. And how the relationship can be flexibly inferred and adapt in different cases. All possible hierarchical relationship scales exponentially with respect to the number of objects and how to infer this structure efficiently. And how to quantify distributed hierarchical structure if it is really realized. All these thoughts are beyond the single-level object-centric representation, and is an essential part to frame the problem in this paper.
>
> ## Overly simplistic evaluation datasets
> It is always desirable to showcase in complex datasets, but sometimes at the cost of interpretability. Here, for this problem, we really want to put the validation at the first place: to really make sure an expected part-whole relationship emerges like that in the representation hypothesis. It is quite a different taste from other works (e.g. Agglomerator [1]). We find that the work showcased in complex dataset may ignore such rigorous validation for the hierarchical relationship. For example, there is no explicit quatification of hierarchical relationship in [1] and it is not clear whether the proposed mechanism really works. Secondly, here, we focus on how to represent the part-whole relationship: the distributed representation of a symbolic tree structure. While the content in images can be very complex and fancy, the parsing structure behind is usually not so complex, especially for most single-object image dataset. Here, what we need is rich identifiable part-whole structure in the image. Even though the stimulus seems simple in the paper, the part-whole relationship is already very rich and complex. So the problem is not trivial at all. It is even a little suprising that a model can deal with these structrues in such a reliable manner. What is in our mind is that, given the basic mechanism of how to do this in general, and rigorous procedure to quantify it, scaling is a matter of time, network size and computing resources. Currently, we use super-lightweight realization to showcase the idea and there is a large room for scaling. On the other hand, we are aware that it would be super interesting if the results could really be scaled to real-word dataset, but as far as we know, there is no such work yet that can both have reliable structure representation and deal with real-world images. So we discussed that as future work in the Appendix A.1
>
> We are not sure what the reviewer means by "scalability issue of spiking neural networks". Actually, scalability is not necessarily an issue of SNN and there are large SNN models as well, and realizing on proper hardwares may further enhance the efficiency.
>
> [1] Nicola Garau, Niccoló Bisagno, Zeno Sambugaro, and Nicola Conci. Interpretable part-whole hierarchies and conceptual-semantic relationships in neural networks. 2022 IEEE/CVF Conference on Computer Vision and Pattern Recognition (CVPR), pp. 13679–13688, 2022.
>
> ## On the compute efficiency of COMPOSER: The authors must evaluate the compute and learning complexity of COMPOSER in comparison to prior art. Can there be a comparison on the number of FLOPS, model parameters or size between COMPOSER and other existing models?
>
> The current model is super-light weight, as shown in Table4 in appendix, the DAE for SHOPs is only a two-layer MLP with 400 hidden size (the number of parameter therefore ~ 60X60X400, where 60X60 is image size). The parameter can further be much reduced if realized as CNN. In contrast, [1] has a much larger size, 72 millian in total, even for a downsampled image (8X8 precision), even with CNN as backbone. Training complexity is a main limitation stated in [1].

---

> ### Comment · Reviewer_A4t8 · 2024-12-03
> **Response to rebuttal from authors**
>
> I thank the authors for engaging with my review and that of others. I still retain my major concern that the evaluation datasets are quite simplistic. On the scalability of SNNs, I meant that non spiking rate-based neural networks are better at learning expressive representations of large scale datasets unlike SNNs which haven't yet shown significant improvements over rate-based ANNs. Thanks to the authors for highlighting the compute requirements of COMPOSER. I am not changing my score, I recommend the authors to further demonstrate the proposed COMPOSER's ability on natural image datasets and make the contributions of the proposed work clearer.

---

### Official Review · Reviewer_YoA7 · 2024-11-04

**Soundness:** 3
**Presentation:** 3
**Contribution:** 3
**Rating:** 6
**Confidence:** 3

**Summary:**

The authors introduce a mechanism for representing the fact that objects have a kind of part hierarchy.   They implement this mechanism in a simple case and then evaluate it on some novel metrics for several simple synthetic datasets.

**Strengths:**

The idea is interesting.

The metrics evaluated are probably a rich approach to model understanding.  The neuronal analysis is creative.

**Weaknesses:**

It's a bit simplified, and it's not obvious how the approach will apply in real-world datasets.

**Questions:**

How will this model be applied to much more complex cases in the real world?

---

> ### Author Response · Authors · 2024-11-17
> **Discussion with Reviewer**
>
> We thank the reviewer for pointing out the proposed framework probably acts as a rich approach to model understanding, which is indeed our motivation behind.
>
> On the one hand, we would like to stress out the "non-simple" nature of the seemingly simple task. While the contents / features are minimal in these images, the underlying part-whole structure is rich. In contrast, such compositional structure may be quite simple for seemingly complex single-object images (even multi-object images). So, in the sense of the problem we focused on in this paper, the dataset is not as simple as it seems. We do not simply classify or reconstruct or segment these images, instead ,we aims to represent parts and wholes and their relationship as the distributed network activations in a flexible manner. That is quite a challenge for most ANNs, e.g. CNN, or Slot-Attention. So we want to stress that our dataset does not lose generality or validation power on the issue we focus on. And this issue is usually a missing one.
>
> On the other hand, the complexity of contents or features in the images can be reduced to similiar case as this paper if we have a powerful encoder to project high-dimensional raw image onto low-dimension manifold in latent layer (they have similiar "simple" appearance and we can start the procedure from those simpler latent layer), which is exactly the common practice in object-centric literactures [1,2]. So the rationale is that, once we have a rigorous framework of representation, prototypical model, and evaluation pipeline, scaling is a matter of time and computing resources. Therfore, we treat it as a future work and discussed how it can be done in the Appendix A.1. In general, scaling is a challenge for most, if not all, object-centric models if without a pre-processing to reduce the dimensionality.
>
> [1] Maximilian Seitzer, Max Horn, Andrii Zadaianchuk, Dominik Zietlow, Tianjun Xiao, Carl-Johann Simon-Gabriel, Tong He, Zheng Zhang, Bernhard Scholkopf, Thomas Brox, and Francesco Locatello. Bridging the gap to real-world object-centric learning. ArXiv, abs/2209.14860, 2022.
>
> [2] Sindy Lowe, Phillip Lippe, Francesco Locatello, and Max Welling. Rotating features for object
> discovery. ArXiv, abs/2306.00600, 2023.

---

### Official Review · Reviewer_Hzqi · 2024-11-04

**Soundness:** 2
**Presentation:** 2
**Contribution:** 2
**Rating:** 3
**Confidence:** 3

**Summary:**

The authors propose a biologically-inspired framework for representing part-whole hierarchies using neuronal coherence, implemented through a spiking neural network architecture called Composer. The system uses denoising autoencoders and hierarchical time scales to generate emergent oscillatory dynamics that encode part-whole relationships. This paper is very similar to a prior submission to ICLR2024 (submission 297), which I also reviewed.

**Strengths:**

- Addresses the fundamental binding problem in an interesting way
- Better organized presentation compared to previous versions

**Weaknesses:**

- The core technical contribution still feels unclear despite improved presentation
- The training methodology remains underspecified in the main text; crucial details are still relegated to a (very!) lengthy appendix
- The baseline comparison (Agglomerator) performs suspiciously poorly with little analysis of why
- The evaluation relies heavily on toy datasets with unclear path to real-world applications

**Questions:**

I commend the authors on improving this submission compared to last year's submission in terms of clarity and presentation. Yet the biggest flaw remains: it's a lot of prep work and dozens of pages of appendix to support a bespoke neural network that solves one very small, toy task. I would be more inclined to positively review this submission if it presented new results compared to last year's submission, especially presenting results on non-toy datasets.

---

> ### Author Response · Authors · 2024-11-17
> **Discussion with the Reviewer**
>
> We thank the reviewer for double reviewing our paper and acknowledging the improvement of our work.
>
> Indeed, we gave quite a lot of efforts to represent the idea in a self-contained manner in the main text, and to express it in an easy-to-follow manner, which is the main issue in the previous version.
>
> As the reviewer pointed out, the content of this paper is so rich that we have to move quite a lot of details into the Appendix, including the training methodology. The rationale behind is that, this paper has the duty to formulate the problem, representation-level hypothesis, novel insights of the solution, and novel evaluation pipeline, and put these issues at the first place. The training details or other technical details therefore have to be left to the Appendix, for readers who have interests. These are indeed not the main point of this paper and they can be realized in diverse ways, dependent on personal preference or computing resources. There is a large room or freedom to technically realize / extend the model, as long as one really understands the insights behind the general framework.
>
> We hope again that the reviewer review our work in a case-by-case manner, and understand our rationale to organize the main paper in this way. If we plug all the technical details into the main paper, it will occupy the room to clarify the more important conceptual issues, which will lead to more confusions in the end. Under our efforts, this version kept a subtle balance between technical details and overal picture.
>
> For the technical contribution, these are summarized in the introduction part of the main paper. To really understand these contributions, it might be necessary to rethink about the "where we really are" and core challenges on dealing with this issue: how to represent parts, wholes, and the relationship among them as distributed neural activations. Slot-model uses localized subspaces (discretely divided before-hand and fixed there-after) to deal with each objects, and it is unclear how to replace these slots as distributed and dynamically formed activations, which is called neuronal coherence in this paper, or alternatively identical islands of vectors in Hinton's imaginary paper [1] or feature rotation in one recent single-level object-centric paper [2]. But this issue is essential, not just interesting, to promote machine vision and to understand human vision [1]. Here, we need to think about how to formulate and quantify the hierarchical relationship and levels, even though we know how to represent each objects. And how to disentangle the representation of each objects from that of relationship in a neural representation space, while keeping both to be distributed. That is the contribution of representation hypothesis. Further, the idea of dealing with part-whole relationship in this way has a dynamical system nature [1], and it is mostly agnostic on whether it can work at all. That is partially why Hinton term his paper [1] as an imaginary picture, instead of solid realization. It is a big challenge and that is the contribution of our implementation of a prototype model: this line of idea can indeed work robustly after all. This is where we are on this issue.
>
> Also, we need to keep in mind that the nature of this work is to form discrete compositional structure in a distributed and continuous manner. The structure is at the first place. And it is necessary to verify such strcuture is really there. Such rigorous evaluation can not be  replaced by other familiar metrics in ANN like classification accuracy or reconstruction error or even single-level segmentation metrics. And this is the issue of most related works that claim they can work, even on real-world dataset [3]. If we look into how they evaluate the model or even the visualization, it is not clear whether there is such representation at all, or it is hard to distinguish these results from artifacts. That is why we stress the importance of quatitative measure on this issue, and start from relatively simple dataset with identifiable parts and wholes. Such rigorous evaluation is therefore an essential technical contribution to this field, considering where we are. Also, such dataset is not as simple as it seems considering the compositional structure behind: the part-whole structure is much richer than single-object (even real-world) dataset e.g. used in [3].
>
> [1] Geoffrey E. Hinton. How to represent part-whole hierarchies in a neural network. Neural Computation, 2021.
>
> [2] Sindy Lowe, Phillip Lippe, Francesco Locatello, and Max Welling. Rotating features for object discovery. Neurips 2023.
>
> [3] Nicola Garau, Niccoló Bisagno, Zeno Sambugaro, and Nicola Conci. Interpretable part-whole hierarchies and conceptual-semantic relationships in neural networks. 2022 (CVPR)

---

> ### Author Response · Authors · 2024-11-17
> **Discussion with the Reviewer (Continued)**
>
> For the performance of the baseline model [3], it is not a suprise at all. If one look into the original paper, there is no quatitative metric to guarentee they could solve this problem. And it is not clear what each layer is actually representing (there are five layers [3], but which layer is part? which layer is whole?). When looking into the visualization, which is demonstrated on single-object dataset without identifiable parts / wholes, it is not easy to distinguish these representations from artifacts. This is where we are on this issue. And what we aim to do here is to avoid such ambiguity and rigorously frame the problem and evaluate the outcome, even in simple cases. It is not easy to explain why they doesn't work in our simple dataset because it is euqally not obvious to explain why or whether they work at all. And it is where we are, it is the SOTA model, if we want to represent part-whole relationship as distributed neural activation. And this again implicate the technical and conceptual contribution of our work: we firstly show that there is an approach to really represent part-whole relationship in distributed manner (though for very simplified cases) in a robust manner, quantified by explicit metrics.
>
> For the concerns towards the real-world applications, we need to rethink: is it more important to showcase on real-world (even single-object) dataset, at the cost of agnostic on whether there are part-whole representation at all (are they artifacts?); or is it more important to firstly verify the idea in a quantifiable way and explain the framework (e.g. representation hypothesis) clearly. We strongly believe the latter is more important if we have to make a trade off. What's more, the framework is not necessarily limited on toy dataset, and there is a clear path to scale (prototypical models verified on toy-datasets) to more complex images [4], given enough GPUs. The idea in [4] is that, we can use a powerful encoder (e.g. DINO) to project the high-dim images onto a low-dim latent space, where the representation is much more compact and has a similiar appearance as the toy-datasets. Therefore, if we take these reduced latent representation as input to the object-centric models, it could still deal with those seemly complex images. The rationale behind is still that the “compositional structure” can be quite simple even for a seemingly complex image, and it is the former that we really care about. And we could leave the burden of dimension reduction to a powerful pre-processing encoder. Currently we realize the model on a single 2080 GPU, and the DAE in our model is just a two-layer MLP with 400 hidden size (SHOPs dataset). We do not use pre-processing to deal with images. So there is a large room to scale it. We have discussed possible paths on scaling in the Appendix A.1.
>
> We totally agree that scaling is an important direction to make the idea proposed in this paper more attractive and useful for real-world applications, and there is a long-way to go. However, at the same time, we hope to remind the reviewer that the value / logic of this paper still stands and self-contained even with seemingly toy dataset (they are not toy in terms of the compositional structure). And pluging more results (scaling and all the techinical details to achieve that) into the already very rich main text may be at the cost of making more basic conceptualization less clear. So we recommends to treat the scaling issue as a seperate problem, and it is indeed the case in related works [4]. And we hope the reviewer to evaluate the value / contribution and our response based on what this paper has already done, and where we actually are on the overall problem.
>
> Lastly, there are indeed several new interesting results, please see Appendix (Fig 30, 31). We use a new dataset (Teris) to demonstrate the "multistability" or "the capability to deal with ambiguity" of our model.
>
> [4] Maximilian Seitzer, Max Horn, Andrii Zadaianchuk, Dominik Zietlow, Tianjun Xiao, Carl-Johann Simon-Gabriel, Tong He, Zheng Zhang, Bernhard Scholkopf, Thomas Brox, and Francesco Locatello. Bridging the gap to real-world object-centric learning. ArXiv, abs/2209.14860, 2022.

---

### Meta-Review · Area_Chair_dz82 · 2024-12-20

**Metareview:**

This paper proposes a framework for representing hierarchical visual relationships called Composer. Composer consists of a set of denoising autoencoders which learns representations at multiple levels. To test this model, the authors construct 4 synthetic datasets with hierarchies, and demonstrates that Composer achieves better performance compared to a baseline model (Agglomerator) at finding part-level and whole-level structure. While reviewers appreciate the challenge of learning hierarchical structures and the connection to biological vision, concerns were raised on the unclear technical presentation, motivation of the problem itself and the experiments (e.g. relevance and experiments on real-world data), and choice of baselines (e.g. those brought up by reviewer 9gYW). After reviewing the rebuttal, I think the writing of the paper could be substantially clarified following reviewer comments (e.g. making the motivation more clear (why can't existing work understand Figure 1?; why is Composer unique in this?), adding experiments on real-world datasets would also definitely help). I highly encourage the authors to incorporate the suggestions from reviewers on the experiments and baselines to strengthen this paper.

**Additional Comments On Reviewer Discussion:**

Reviewers raised initial points on the baselines and experiments in the paper, as well as questions on the motivation of this work and the technical presentation. While some questions were answered by the authors during rebuttal, most reviewers still had remaining concerns about this paper (e.g. the baselines and dataset choices). This is confirmed during the AC-reviewer discussion period.

---

### Decision · Program_Chairs · 2025-01-22

Reject